# PAX3-FOXO1 dictates myogenic reprogramming and rhabdomyosarcoma identity in endothelial progenitors

Madeline B. Searcy[1,2,8], Randolph K. Larsen IV[1,2,8], Bradley T. Stevens[1,2], Yang Zhang[3], Hongjian Jin[4], Catherine J. Drummond[1], Casey G. Langdon [1], Katherine E. Gadek [1], Kyna Vuong[1], Kristin B. Reed [1], Matthew R. Garcia[1], Beisi Xu [4], Darden W. Kimbrough[1,5], Grace E. Adkins[1,2], Nadhir Djekidel [4], Shaina N. Porter[6], Patrick A. Schreiner [3], Shondra M. Pruett-Miller [6], Brian J. Abraham [3], Jerold E. Rehg[7] & Mark E. Hatley [1] ✉

Fusion-positive rhabdomyosarcoma (FP-RMS) driven by the expression of the PAX3-FOXO1 (P3F) fusion oncoprotein is an aggressive subtype of pediatric rhabdomyosarcoma. FP-RMS histologically resembles developing muscle yet occurs throughout the body in areas devoid of skeletal muscle highlighting that FP-RMS is not derived from an exclusively myogenic cell of origin. Here we demonstrate that P3F reprograms mouse and human endothelial progenitors to FP-RMS. We show that P3F expression in *aP2-Cre* expressing cells reprograms endothelial progenitors to functional myogenic stem cells capable of regenerating injured muscle fibers. Further, we describe a FP-RMS mouse model driven by P3F expression and *Cdkn2a* loss in endothelial cells. Additionally, we show that P3F expression in *TP53*-null human iPSCs blocks endothelial-directed differentiation and guides cells to become myogenic cells that form FP-RMS tumors in immunocompromised mice. Together these findings demonstrate that FP-RMS can originate from aberrant development of non-myogenic cells driven by P3F.

Rhabdomyosarcoma (RMS) is the most common pediatric soft tissue sarcoma[1]. Despite clinical trials and translational studies, the 3-year survival rate for children with high-risk RMS has remained at 20% for four decades[2]. RMS is histologically classified into two major subgroups, alveolar (ARMS) and embryonal (ERMS)[1]. Most ARMS tumors harbor chromosomal translocations t(2;13)(q35;q14) or t(1;13)(p36;q14) resulting in PAX3-FOXO1 (P3F) or PAX7-FOXO1 (P7F) fusion oncoproteins, joining the PAX3/7 DNA-binding domain with the transactivation domain of FOXO1[3–6]. The presence of P3/7F foretells a worse prognosis[2]. ARMS tumors lacking P3/7F both molecularly and

clinically resemble ERMS, thus classifying tumors based on fusion status as either fusion-positive (FP-RMS) or fusion-negative (FN-RMS) more accurately categorizes RMS patients[7]. RMS resembles undifferentiated, embryonic skeletal muscle and is believed to arise from skeletal muscle progenitor cells that fail to differentiate[8]. However, RMS occurs throughout the body in sites devoid of skeletal muscle including the prostate, urinary bladder, omentum, gallbladder, and pulmonary artery, highlighting potential alternative cells-of-origin[9–12]. Additionally, a subset of patients with high-risk FP-RMS present without an identifiable primary tumor with either widespread disseminated

[1]Department of Oncology, St. Jude Children's Research Hospital, Memphis, TN 38105, USA. [2]St. Jude Graduate School of Biomedical Sciences, Memphis, TN 38105, USA. [3]Department of Computational Biology, St. Jude Children's Research Hospital, Memphis, TN 38105, USA. [4]Center for Applied Bioinformatics, St. Jude Children's Research Hospital, Memphis, TN 38105, USA. [5]Rhodes College, Memphis, TN 38112, USA. [6]Department of Cell and Molecular Biology, St. Jude Children's Research Hospital, Memphis, TN 38105, USA. [7]Department of Pathology, St. Jude Children's Research Hospital, Memphis, TN 38105, USA. [8]These authors contributed equally: Madeline B. Searcy, Randolph K. Larsen IV. ✉e-mail: mark.hatley@stjude.org

disease or solely bone marrow involvement[13,14]. These tumors' cell of origin remains unknown.

P3F functions as a pioneer transcription factor for RMS[15]. Pioneer transcription factors are capable of binding and opening condensed chromatin thus altering DNA accessibility[16]. The altered accessibility to transcription factor binding sites could provide an opportunity for cell fate reprogramming. P3F activates myogenic super-enhancers that define RMS cell identity including *MYOD1, MYOGENIN* and *MYCN*[17]. The ability of P3F to both establish a new cell state through pioneering activity and drive myogenic transcriptional programs highlights that the cell of origin for FP-RMS may differ from a skeletal muscle cell. Often, transcriptional reprogramming by pioneer factors relates back to their normal tissue expression[18,19]. In the case of P3F, the regulatory regions driving *PAX3* and *FOXO1* expression give insight to potential cells-of-origin for FP-RMS. Specifically, cis-regulatory elements between +104 to +148 kb from the mouse *Foxo1* transcription start site drive FOXO1 expression in embryonic and adult vascular endothelial cells[20]. These regulatory elements are highly conserved across species and drive endothelial expression of FOXO1 in humans and mice[20]. Studies in human FP-RMS cell lines reveal that the t(2;13)(q35;q14) translocation generating P3F creates a novel topologically associated domain (TAD) by combining *PAX3* and *FOXO1* cis-regulatory regions[20]. This new TAD allows the interaction of the *PAX3* promoter with downstream *FOXO1* regulatory sequences driving expression of P3F in non-PAX3 expressing tissues specifically highlighting embryonic and adult vasculature as potential cells-of-origin for FP-RMS[20]. Thus, endothelial cell specific expression of P3F causing transdifferentiation of vasculature is a potential mechanism of transformation for FP-RMS.

Previously, our lab showed that FN-RMS can arise from non-skeletal muscle origins. Using genetically engineered mouse models, we illustrated constitutive activation of the Sonic Hedgehog pathway with conditional SMO^M2 expression in *aP2-Cre* expressing endothelial progenitors transdifferentiates cells to FN-RMS[21,22]. Whether FP-RMS has a similar cell of origin remains unknown. P3F and P7F expression in spinal cord progenitors triggers transdifferentiation of neural cells to FP-RMS-like cells with myogenic characteristics *in ovo*[23]. Experiments on the developing dermomyotome detail the balance between PAX3 and FOXC2 transcriptional activity that dictates myogenic versus vascular cell fate in differentiating somites[24]. Furthermore, P3F suppresses FOXC2, highlighting a potential mechanism for P3F to block endothelial differentiation by decreasing FOXC2 expression[24,25]. However, non-myogenic cells have not been shown as a cell of origin for FP-RMS tumors.

Here we show that P3F pioneering activity is sufficient to drive FP-RMS from endothelial cells in both genetically engineered mouse models and in human induced pluripotent stem cells (iPSCs) thus solidifying endothelial cells as a cell of origin for FP-RMS.

## Results

### P3F reprograms aP2-Cre expressing cells to functional muscle stem cells

Previously, we intercrossed *aP2-Cre* and *ROSA26-loxP-stop-loxP-tdTomato* (*R26-tdTom*) mice and showed *aP2-Cre* drove recombination resulting in Tomato-positive (Tom^+) endothelial cells within the skeletal muscle interstitium throughout the body despite tumors exclusively arising from the head and neck[22]. The Tom^+ cells were distinct from PAX7^+ skeletal muscle stem cells located beneath the lamina of the myofiber. The cells indelibly labeled by *aP2-Cre* do not participate in normal skeletal muscle development[22]. We had not determined whether *aP2-Cre* is expressed during muscle stem cell activation and whether *aP2-Cre* labelled interstitial cells participate in skeletal muscle repair. To explore this, we injected cardiotoxin into the gastrocnemius hindlimb skeletal muscle and the sternocleidomastoid (SCM) neck muscle of *aP2-Cre;R26-tdTom* compound mutant mice and showed that *aP2-Cre* labeled Tom^+ cells do not contribute to skeletal muscle regeneration following

injury (Fig. 1a). These data suggest that *aP2-Cre* is not expressed and does not drive Cre-mediated recombination in normal skeletal muscle development or muscle stem cell development or activation.

Next, we sought to explore whether the P3F would transform *aP2-Cre* endothelial progenitors. In *Pax3^P3Fm* mice, *Pax3* expression is unperturbed in the absence of Cre recombinase, and *Pax3-Foxo1* (P3F) is expressed following Cre-mediated recombination[26]. To generate the P3F oncofusion in mice, exons 1–7 of *Pax3* were fused to exons 2 and 3 of *Foxo1* along with 6.5 kb of the 3' untranslated region. Thus, P3F expression is driven by the *Pax3* promoter and influenced by the regulatory elements of *Foxo1* knocked into the *Pax3* locus; however, this model lacks downstream regulatory elements over 100 kb from the transcriptional start site required to drive FOXO1 expression in endothelial cells[26]. Enforced expression of P3F from the *Pax3^P3Fm* allele in combination with loss of either *Cdkn2a* or *Trp53* in myoblasts with *Myf6-Cre* induces tumors that resemble human ARMS[26]. To better recapitulate the mutational landscape seen in human FP-RMS tumors, we generated *aP2-Cre;Cdkn2a^Flox/Flox;Pax3^P3Fm/P3Fm; R26-tdTom* (ACP) compound mutant mice to drive P3F and tdTomato expression, and to delete *Cdkn2a* in *aP2-Cre* expressing cells[27,28]. Similar to *aP2-Cre;R26-tdTom* mice, Tom^+ cells in ACP mice were located between skeletal muscle fibers in the interstitium in muscle tissue throughout the body. We compared skeletal muscle from the hindlimb and the SCM given the locational specificity of FN-RMS tumors originating from *aP2-Cre* expressing cells in the head and neck[21]. Surprisingly, of 940 Tom^+ cells counted in the SCM of ACP mice, 30% co-localized with PAX7. None of the 66 PAX7^+ muscle stem cells counted beneath the basal lamina were Tom^+. Tom^+ cells in the hindlimb of ACP mice did not co-localize with PAX7 (Fig. 1b). PAX7 expression in Tom^+ interstitial cells suggests that P3F reprograms *aP2-Cre* labelled cells in the SCM to muscle stem cells.

To explore the myogenic potential of interstitial Tom^+PAX7^+ cells in ACP mice, we dissociated SCM and hindlimb, cultured all cells until confluence, and induced myogenic differentiation. We found that dissociated ACP cells from the SCM but not from the hindlimb formed myotubes evident by the co-localization of Tomato and myosin heavy chain (MyHC) staining in 51% of Tom^+ cells (Fig. 1c). To further define the identity of the Tom^+ SCM ACP cells, we used a previously reported fluorescence activated cell sorting (FACS) approach to isolate skeletal muscle stem cells[29]. We confirmed the fidelity of this method to quantitate PAX7^+ muscle stem cells by processing muscle from *Pax7^CreERT2;R26-tdTom* mice that express Tomato in muscle stem cells after tamoxifen administration (Fig. 1d). Using this method on ACP SCM and hindlimb muscle, 10% of isolated muscle stem cells from the SCM were Tom^+ while none from the hindlimb were Tom^+ (Fig. 1e)[22]. To determine whether these cells are functional muscle stem cells, we tested if they contribute to muscle regeneration after injury. Following cardiotoxin injury, Tom^+ myofibers were present in the SCM but not in the hindlimb of ACP mice and were comparable to injured *Pax7^CreERT2;R26-tdTom* muscle (Fig. 1f). These results demonstrate the capacity of P3F to reprogram endothelial cells in the SCM into cells with myogenic potential.

Given that P3F expression reprogrammed *aP2-Cre* labeled cells with muscle stem cell properties capable of muscle regeneration, we observed ACP mice to determine if P3F expression could drive transformation into FP-RMS. All 24 mice observed developed tumors with a median tumor-free survival of 251 days (Supplementary Fig. 1a). Although mice fail to develop FP-RMS, they manifest a variety of tumor types, including mononuclear phagocytic sarcoma and angiosarcoma (Supplementary Fig. 1b–f). We previously reported that *aP2-Cre;Cdkn2a^Flox* mice develop solely mononuclear phagocytic sarcoma suggesting that P3F expression drives diversity in the ACP tumor spectrum[30]. We validated *Pax3-Foxo1* expression in three ACP tumors, none of which expressed the myogenic regulatory transcription factors *Myod1* or *Myog* that are diagnostic for rhabdomyosarcoma (Supplementary Fig. 1g).

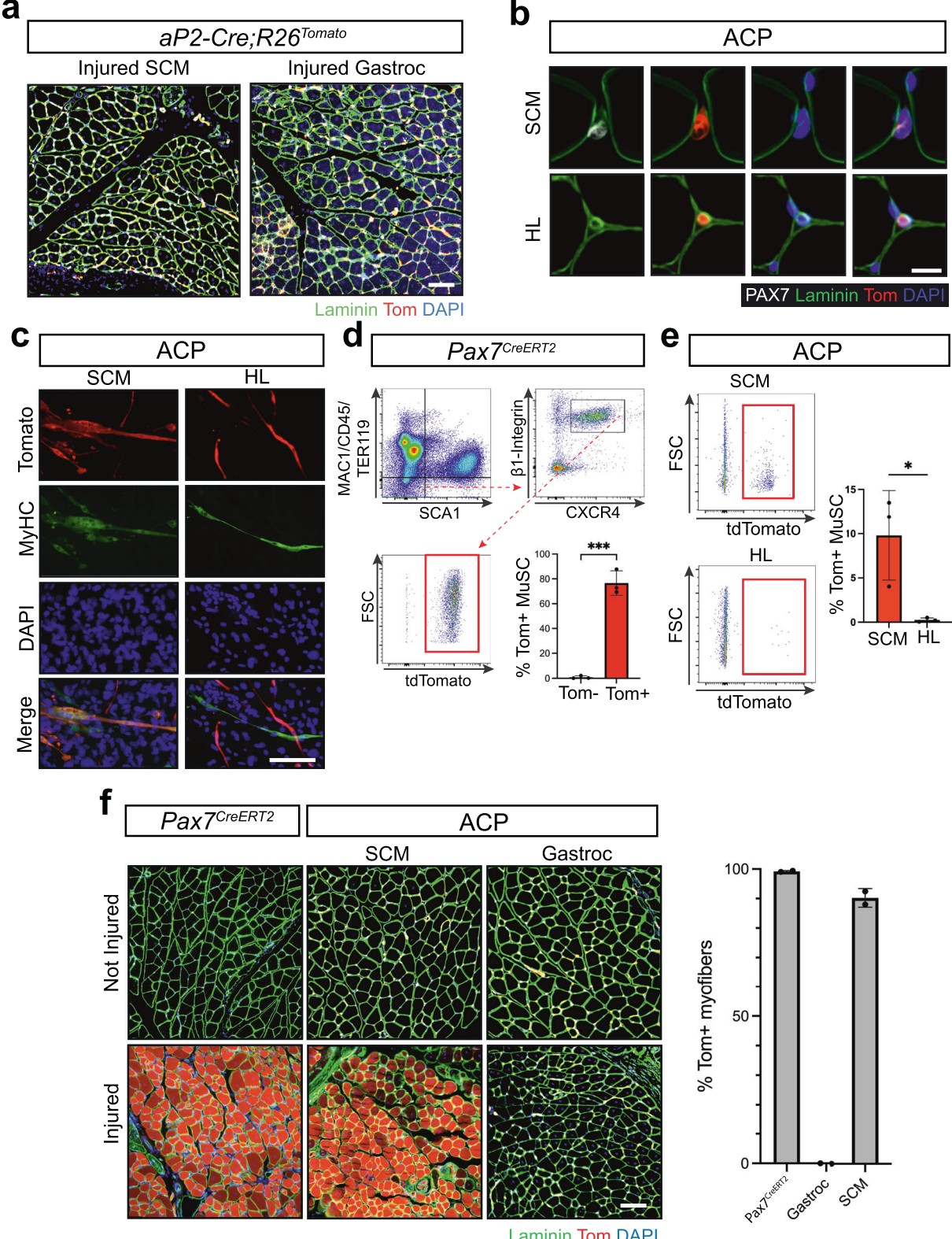

**Fig. 1 | P3F reprograms endothelial cells to functional muscle stem cells.**
**a** Representative immunofluorescence (IF) from *aP2-Cre;R26-tdTom* sternocleido-mastoid (SCM) and gastrocnemius (gastroc) following cardiotoxin (CTX) injury. Laminin (green), tdTomato(red), DAPI nuclear stain (blue) (*n* = 3). Scale bar = 100 μm. **b** Representative IF of SCM and quadricep femoralis hindlimb (HL). PAX7 (white), laminin (green), tdTomato (red), DAPI (blue). This experiment was repeated 3 times with similar results. Scale bar = 10 μm. **c** Representative IF of myogenic differentiation from ACP SCM and HL. Myosin heavy chain (MyHC, green), tdTomato (red), DAPI (blue)(n = 3, 100+ nuclei counted/replicate) Scale bar = 50 μm.

Flow cytometry analysis of muscle stem cells by (MAC1/CD45/TER119) and SCA1 negative, β1-Integrin and CXCR4-positive, and Tom⁺ cells from **d** *Pax7-CreERT2;R26-tdTom* muscle - (*n* = 3 biological replicates) (*P* = 0.0002) and **e** Flow cytometry from ACP SCM and HL (*n* = 3 biological replicates) (*P* = 0.0314). **f** Representative IF from CTX injured *Pax7-CreERT2;R26-tdTom* gastroc (*n* = 2), and ACP SCM (*n* = 2) and gastrc (*n* = 2). Laminin (green), tdTomato (red), DAPI (blue). Scale bar = 100 μm. *P* values determined by Student's *t*-test (unpaired, two-tailed); *P < 0.05 and ***P < 0.001. Data represented as mean+/- SEM. Source data are provided as a Source Data file.

Previously, we noted perinatal lethality using an independently derived *aP2-Cre* with the same 5.4 kilobase *aP2* (aka *Fabp4*) cis-regulatory region driving Cre to activate *SmoM2* expression[22]. *Fabp4-Cre;SmoM2* compound mutant mouse embryos had myogenic proliferations in the neck like *aP2-Cre;SmoM2* adult mice[22]. *aP2-Cre* leads to recombination in a variety of tissues in addition to endothelial cells including both brown and white adipose tissue, macrophages, and dorsal root ganglia[22,31,32]. We hypothesized the varied phenotype resulted from either variable Cre recombinase expression, altered cell specific expression, or positional effects from random integration. Allele- and age-dependent Cre-mediated recombination has been reported within the *aP2-Cre* lines[32]. To determine the difference in Cre-mediated recombination efficiency of *aP2-Cre* and *Fabp4-Cre*, we bred *aP2-Cre* and *Fabp4-Cre* mice to *R26-tdTom* reporter mice and performed real-time PCR and immunoblot. We found that *Fabp4-Cre;R26-tdTom* SCM and hindlimb has higher levels of Cre mediated recombination observed through increased Tomato expression (Supplementary Fig. 1h, i). To define the Cre expressing cell types in *aP2-Cre;R26-tdTom* and *Fabp4-Cre;R26-tdTom* mice we performed droplet-based single-cell transcriptional profiling (scRNAseq) on Tom$^+$ cells isolated by FACS from SCM. Unsupervised clustering with Uniform Manifold Approximation and Projection (UMAP)[33] identified 10 distinct clusters with very similar populations in Tom$^+$ cells from both *aP2-Cre;R26-tdTom* and *Fabp4-Cre;R26-tdTom* mice (Supplementary Fig. 1j, k)(Supplementary Data 2). *Fabp4-Cre;R26-tdTom* had significantly increased cell numbers in two distinct fibroblast populations (Supplementary Fig. 1l). The endothelial populations were not significantly different between *aP2-Cre;R26-tdTom* and *Fabp4-Cre;R26-tdTom* mice.

Surprisingly, scRNAseq in both Tom$^+$ cells from *aP2-Cre;R26-tdTom* and *Fabp4-Cre;R26-tdTom* mice revealed a myocyte population despite not observing Tomato expression in skeletal muscle cells in any previous experiments. Initial scRNAseq was performed to ensure all Tom$^+$ cells were part of the analysis. The liberal gating approach could have allowed Tom$^-$ cells to contaminate the Tom$^+$ population. To test this, we increased the FACS stringency to include the top 50% of Tom$^+$ cells from *aP2-Cre;R26-tdTom* SCM and repeated scRNAseq. Cluster analysis and UMAP confirmed that *aP2-Cre* expression occurred almost entirely in endothelial cells and not in myocytes (Supplementary Fig. 1m)(Supplementary Data 3). The myogenic genes *Acta1* and *Ckm* are not expressed by any of the clusters in this analysis (Supplementary Fig. 1n, o) and myogenic genes are not observed in the top expressed genes of any cluster (Supplementary Fig. 1p). These findings confirm that *aP2-Cre* is not expressed by myogenic cells. Next, we investigated whether increased recombination with *Fabp4-Cre* drives FP-RMS. *Fabp4-Cre;Cdkn2a$^{Flox/Flox}$;Pax3$^{P3Fm/P3Fm}$;R26-tdTom* (FCP) compound mutant mice developed tumors with 95% penetrance with a median onset of 188 days. FCP mice develop a spectrum of tumors including 3 of 20 mice with tumors consistent with solid-type alveolar FP-RMS (Supplementary Fig. 1q–u).

### Tek-Cre is expressed by endothelial cells labeled by aP2-Cre

Given the diverse tissue expression of the *aP2-Cre*, we sought to specifically direct P3F expression to explore an endothelial origin for FP-RMS. Tunica intima endothelial kinase (TEK) is expressed in endothelial cells and hematopoietic cells, and the 2.1 kb *Tek* enhancer and promoter region has been used to direct Cre recombinase expression in *Tek-Cre* (*Tie2-Cre*) mice to study the endothelial cell lineage[34–36]. Additionally, TEK expression has been reported in muscle stem cells suggesting that *Tek-Cre* could drive recombination in muscle stem cells[37]. To test this in our system, we interrogated whether Tom$^+$ cells from the SCM and hindlimb of *Tek-Cre;R26-tdTom* expressed muscle stem cell markers or could function as a muscle stem cell. Tom$^+$ cells from the SCM and hindlimb of *Tek-Cre;R26-tdTom* mice did not co-localize with PAX7, were not identified by a muscle stem cell FACS method, did not fuse with myoblasts in an in vitro myogenic

differentiation assay, and did not participate in muscle injury repair in vivo, demonstrating that *Tek-Cre* did not drive recombination in muscle stem cells in our mouse models (Supplementary Fig. 2a–d). To further identify *Tek-Cre* labeled cells, we performed RNA sequencing on Tom$^+$ and Tom$^-$ cells isolated by FACS from SCM of *Tek-Cre;R26-tdTom* mice (Supplementary Fig. 2e) (Supplementary Data 4). Tom$^+$ cells expressed genes consistent with hematopoietic and endothelial cells while Tom$^-$ cells expressed genes consistent with myogenic cells (Supplementary Fig. 2f–h) (Supplementary Data 5). Together these data suggest that the *Tek* cis-regulatory region in *Tek-Cre* mice lacks the myogenic regulatory element required for *Tek* expression in muscle stem cells and confirms that *Tek-Cre* is not expressed in muscle cells.

To compare specificity of *Tek-Cre* and *aP2-Cre*, we stained embryonic branchial arches and adult SCM from *Tek-Cre;R26-tdTom* mice for aP2. In both tissues, there were Tom$^+$ cells with and without aP2 staining; however, there were no aP2$^+$Tom$^-$ cells (Supplementary Fig. 2i, j). This showed that *Tek-Cre* labels cells distinct from those expressing aP2 both during development and in adult tissue. Next we determined the gene expression of Tom$^+$ and Tom$^-$ cells isolated by FACS from the SCM of *Tek-Cre;R26-tdTom* and *aP2-Cre;R26-tdTom* mice (Supplementary Fig. 2k) (Supplementary Data 6). Gene ontology (GO) analysis performed on 343 genes uniquely enriched in Tom$^+$ cells from *aP2-Cre* revealed terms associated with cell development and signal transduction, while GO terms from 1567 genes uniquely enriched in Tom$^+$ cells from *Tek-Cre* were mostly related to the innate immune response (Supplementary Fig. 2l, m) (Supplementary Data 7 and 8)[36]. GO terms of the 765 shared genes enriched in both *aP2-Cre* and *Tek-Cre* include vascular development and blood vessel morphogenesis, suggesting they share expression in similar endothelial populations (Supplementary Fig. 2n) (Supplementary Data 9). We further dissected the cell types in *aP2-Cre* and *Tek-Cre* that drive Cre-mediated recombination through scRNAseq of Tom$^+$ cells from SCM of *aP2-Cre;R26-tdTom* and *Tek-Cre;R26-tdTom* mice. Although there were some differences in frequency, both *aP2-Cre* and *Tek-Cre* mediate recombination in the same cell types in the SCM including endothelial and hematopoietic cells (Supplementary Fig. 2o-r)(Supplementary Data 10). These findings demonstrate that *Tek-Cre* is not expressed in skeletal muscle cells and can be leveraged to test P3F expression in endothelial cells.

### Endothelial cells from Tek-Cre;Cdkn2a$^{Flox/Flox}$;Pax3$^{P3Fm/P3Fm}$ mice are not muscle stem cells

Given P3F drives myogenic transdifferentiation of *aP2-Cre* cells in the SCM, we investigated the effect of P3F expression in *Tek-Cre* expressing cells in the SCM and hindlimb of *Tek-Cre;Cdkn2a$^{Flox/Flox}$; Pax3$^{P3Fm/P3Fm}$;R26-tdTom* (TCP) mice. FACS-based enrichment for muscle stem cells showed no Tom$^+$ cells within muscle stem cell populations (Supplementary Fig. 3a). Additionally, Tom$^+$ cells did not co-localize with PAX7 in the SCM or hindlimb (Supplementary Fig. 3b), did not fuse with myoblasts in an in vitro myogenic differentiation assay, or participate in muscle injury repair in vivo (Supplementary Fig. 3c, d). These data illustrate P3F does not reprogram Tom$^+$ muscle interstitial cells in TCP mice into functional muscle stem cells.

### Tek-Cre;Cdkn2a$^{Flox/Flox}$;Pax$^{P3Fm/P3Fm}$ mice develop FP-RMS

While Tom$^+$ cells in TCP mice are not functional muscle stem cells, we sought to determine if P3F expression would transform *Tek-Cre* expressing cells in TCP compound mutant mice. For comparison, we used a well-established genetically engineered mouse model of FP-RMS originating from skeletal muscle using *Myf6-Cre* to drive expression of P3F and conditional loss of *Cdkn2a*[26]. To determine the tumorigenic potential of *Cdkn2a* loss, we bred *Tek-Cre;Cdkn2a$^{Flox/Flox}$* mice lacking P3F and exclusively observed mononuclear phagocytic sarcomas (Supplementary Fig. 3e, f). Both TCP and *Myf6-Cre;Cdkn2a$^{Flox/Flox}$;Pax3$^{P3Fm/P3Fm}$* (MCP) mutant mice developed tumors with a median

tumor-free survival of 146 days and 138 days, respectively (Fig. 2a). MCP mice developed a variety of tumors at multiple locations including solid type alveolar FP-RMS (Supplementary Fig. 4a). TCP mice developed multiple tumor types with solid type alveolar FP-RMS being the most frequent that was anatomically restricted to the snout (Fig. 2b; Supplementary Fig. 4b). Cytologic analysis demonstrated diffuse sheets of round tumor cells with variable amounts of cohesive eosinophilic cytoplasm and rhabdomyoblasts, and immunohisto-chemical (IHC) staining for DESMIN, MYOD1, and MYOGENIN consistent with the human solid subtype of ARMS (Fig. 2c). TCP tumors have a higher proliferative index than MCP tumors (Fig. 2d). Both TCP and MCP display heterogenous expression of *Pax3-Foxo1, Myod1*, and *Myogenin* (Fig. 2e). TCP mice developed mononuclear phagocytic sarcomas like we observed in *Tek-Cre;Cdkn2a^Flox/Flox^, aP2-Cre;Cdkn2a^Flox/Flox^*, and ACP mice (Supplementary Fig. 1, Supplementary Fig. 4b, c)[30].

To elucidate the mechanistic underpinnings between TCP and MCP tumorigenesis, we performed global transcriptome analysis. Differentially expressed genes from bulk TCP and MCP tumors were associated with non-tumor stromal cells and lacked unifying characteristics by gene ontology (Fig. 2f) (Supplementary Data 11). Gene set enrichment analysis (GSEA) of TCP and MCP tumors compared to gastrocnemius ($n = 3$, GEO #'s GDM4331454-6) show that both groups of tumors are enriched for ARMS gene signatures demonstrating that they have similar gene expression profiles to human FP-RMS (Fig. 2g) (Supplementary Data 12 and 13). To further analyze the TCP and MCP tumors, we compared genes with >2-fold increased expression with a $P < 0.05$ in TCP and MCP tumors to gene sets of RMS core regulatory circuit transcription factors identified by St. Jude Children's Research Hospital[38] and the US National Institutes of Health (NIH)[27] and RMS dependencies identified by the Broad Institute[39] (Supplementary Data 14). TCP and MCP tumors express many RMS genes with MCP sharing *FOXO1* and *SOX8* expression and TCP sharing *FOXO1*, *SOX8*, and *MYOD1* expression with all three groups. (Fig. 2h). This highlights the similarity of the mouse models with human patient tumors and the importance of these transcription factors in FP-RMS.

One MCP tumor used for the analysis was distinct from the others and expressed lower levels of *Myogenin* and *MyoD1*, prompting us to investigate the heterogeneity present between MCP tumors. IHC staining for myogenic regulatory factors MYOD1 and MYOGENIN revealed diffuse, homogenous staining in all TCP tumors while the MCP tumors had focal, distinct areas of staining scattered throughout the tumor that were more consistent with tumors with mesenchymal and epithelial differentiation than with ARMS (Supplementary Fig. 4d). To better resolve cellular heterogeneity, we compared scRNAseq on two TCP tumors and five MCP tumors (Supplementary Fig. 4e–l). scRNAseq revealed that tumors from both genotypes have variable overlapping populations of cell types (Supplementary Fig. 4e–l)(Supplementary Data 15 and 16). TCP tumors were highly similar except for cluster 0 which is an ambiguous population lacking a clear distinguishing identity and is mainly present in TCP 83 (Supplementary Fig. 4e–h). MCP tumors harbor more variety than TCP tumors. MCP 395 entirely lacks cluster 0, a population like cluster 0 in TCP that lacks an obvious unifying identity. Myogenic tumor clusters 1 and 3 have varying representation across MCP samples. Additionally, MCP 46 and 48 have little to no expression of cluster 8, a myocyte population that expresses high levels of *Myod1* and *Myog* (Supplementary Fig. 4i–l). These findings highlight some heterogeneity in tumor cells from both TCP and MCP mice and confirms that FP-RMS can arise from both myogenic and non-myogenic cells of origin which is not distinguishable by tumor gene expression.

## FP-RMS from endothelial and myogenic cells-of-origin result in tumors comprised of similar cell populations

While FP-RMS tumors from both endothelial and myogenic cells had similar global gene expression profiles except for differences in stromal cells, we sought to uncover reflections of their distinct origins with single-cell analysis. We collected Tom+ cells by FACS from TCP and MCP tumors and from SCM skeletal muscle of *Tek-Cre;R26-tdTom* mice to identify *Tek-Cre* expressing, non-transformed cell populations in the TCP tumors. We performed scRNAseq on Tom+ cells and integrated these data with public datasets of mouse skeletal muscle cells including total cells from enzymatic digestion of hindlimb skeletal muscle (total muscle), tissue-resident skeletal muscle stem cells isolated by FACS (MuSCs), and committed primary myoblasts (MBs) from 3 month old C57BL/6J mice[40]. We performed unsupervised clustering and identified nine unique clusters present across the combined samples (Fig. 3a, b) (Supplementary Data 17). TCP and MCP are similar except for clusters 3 and 6 which are endothelial cells and B cells, respectively (Fig. 3a, b). These populations are also identified in *Tek-Cre;R26-tdTom* cells suggesting that in TCP samples these clusters are non-tumor cells expressing Tomato (Fig. 3b). Using scVelo to estimate RNA velocity and partition-based graph abstraction, we determined the sequence of transcriptomic events taking place across cell clusters to project cell trajectories reflecting progressive differentiation[41,42]. This analysis showed differentiation of endothelial cell cluster 3 to cluster 8, identified as mesenchymal stem cells by analyzing the differentially expressed genes (Fig. 3c, d) (Supplementary Data 18). Cluster 8 further differentiates to tumor cluster 0 suggesting that cluster 8 mesenchymal stem cells are a transition stage between the endothelial cell of origin and FP-RMS (Fig. 3c, d). The transition from endothelial to mesenchymal to tumor cells is reflected by the unique expression signatures of the top ten genes expressed in each cluster (Fig. 3e).

Next, we compared the overlapping clusters present in both TCP and MCP tumors along with primary myoblasts that share a similar gene expression profile to FP-RMS, and identified a tumor population comprised of clusters 0, 1, 2, 3, 7 and 8, marked by a dotted outline (Fig. 3a). While most endothelial cells in cluster 3 are distinct from the tumor, small groups of endothelial cells present in TCP and *Tek-Cre;R26-tdTom* are seen within the tumor population connected to mesenchymal stem cells and myogenic tumor cells, clusters 8 and 2, respectively. (Fig. 3a, b). To further refine the myogenic tumor population, we re-analyzed the designated tumor clusters and plotted them to compare the proportions of clusters present in TCP and MCP tumor cells (Fig. 3f) (Supplementary Data 19). We first looked at *Pax3, Foxo1, Myod1* and *Myogenin* expression and determined that all clusters except for 2 and 7 express *P3F*, and clusters 2, 3, 6, and 9 express both *Myod1* and *Myogenin* (Fig. 3g). While cells from both TCP and MCP tumors were present in all clusters, two clusters are statistically significantly different (Fig. 3h). Interestingly, cluster 0 was enriched in MCP tumors yet contains cells that express genes associated with endothelial cells such as *Epas1*, while cluster 6 was enriched in TCP and contains cells that express genes associated with myocytes such as *Myog* (Fig. 3g, h) (Supplementary Data 20 and 21). This unexpected result, that tumors arising from a myogenic Cre-driver (MCP) were enriched for a subpopulation expressing endothelial genes and tumors arising from an endothelial Cre-driver (TCP) were enriched for a subpopulation expressing skeletal muscle genes demonstrates how similar these tumors are and highlights the potential complications of assuming cell of origin based on tumor gene expression.

## TCP and MCP tumors recapitulate human RMS heterogeneity and gene expression

Recent work has highlighted how subpopulations akin to various stages of classical skeletal muscle development can be recurrently identified across human RMS tumors[43–45]. To determine if the murine TCP and MCP tumors shared the cell states identified in human RMS, we looked for expression of a subset of genes shown to be enriched in conserved cell states by Wei et al. (proliferation, muscle, hypoxia, interferon, mesenchymal-like, and apoptosis) in our integrated TCP,

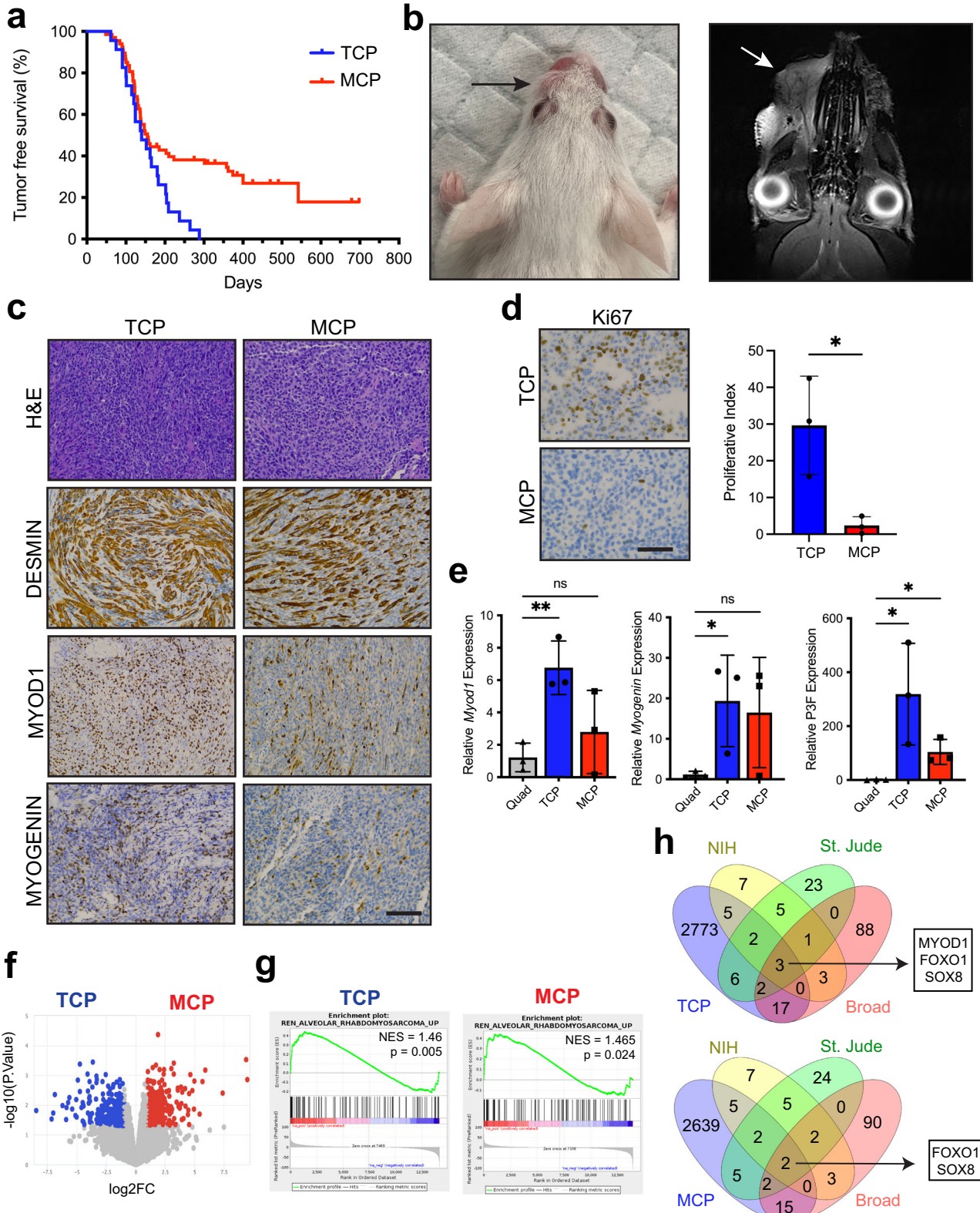

**Fig. 2 | FP-RMS in *Tek-Cre;Cdkn2a^{Flox/Flox};Pax^{P3Fm/P3Fm}* mice. a** Kaplan-Meier tumor-free survival. TCP (blue, *n* = 24) and MCP (red, *n* = 42). **b** Gross photo (left) and magnetic resonance imaging (right) of TCP tumor. **c** Representative histology of TCP (left) and MCP (right) tumors stained for H&E, DESMIN, MYOD1, and MYO-GENIN. *N* = 15 TCP and 29 MCP tumors. Scale bar = 100μm. **d** Representative Ki67 IHC and quantification in TCP (*n* = 3) and MCP (*n* = 3) tumors (three random fields of view per genotype) (*P* = 0.0256). Scale bar = 50 μm. **e** Gene expression by real-time PCR of *Myod1* (Quad−TCP, *P* = 0.0068), *Myogenin* (Quad − TCP, *P* = 0.0499) and *Pax3-Foxo1* (Quad−TCP, *P* = 0.0433) (Quad−MCP, *P* = 0.0187) in TCP (*n* = 3) and MCP (*n* = 3) tumors and quad skeletal muscle (*n* = 3). **f** Volcano plot showing differentially expressed genes with logFC greater than 1 and false discovery rate (FDR) less than 5% in TCP (blue) and MCP (red). **g** Enrichment plot of FP-RMS gene signatures (REN_ALVEOLAR_RHABDOMYOSARCOMA_UP) for TCP and MCP tumors compared to gastrocnemius. **h** Tulip plot overlaying genes two-fold increased in TCP and MCP tumors compared to gastrocnemius (*P* < 0.05), RMS genes identified by the NIH, SJCRH, and the pediatric dependency map (Broad). P values determined by Student's *t*-test (unpaired, two-tailed); *P < 0.05 and **P < 0.01. Data represented as mean+/- SEM. Source data are provided as a Source Data file.

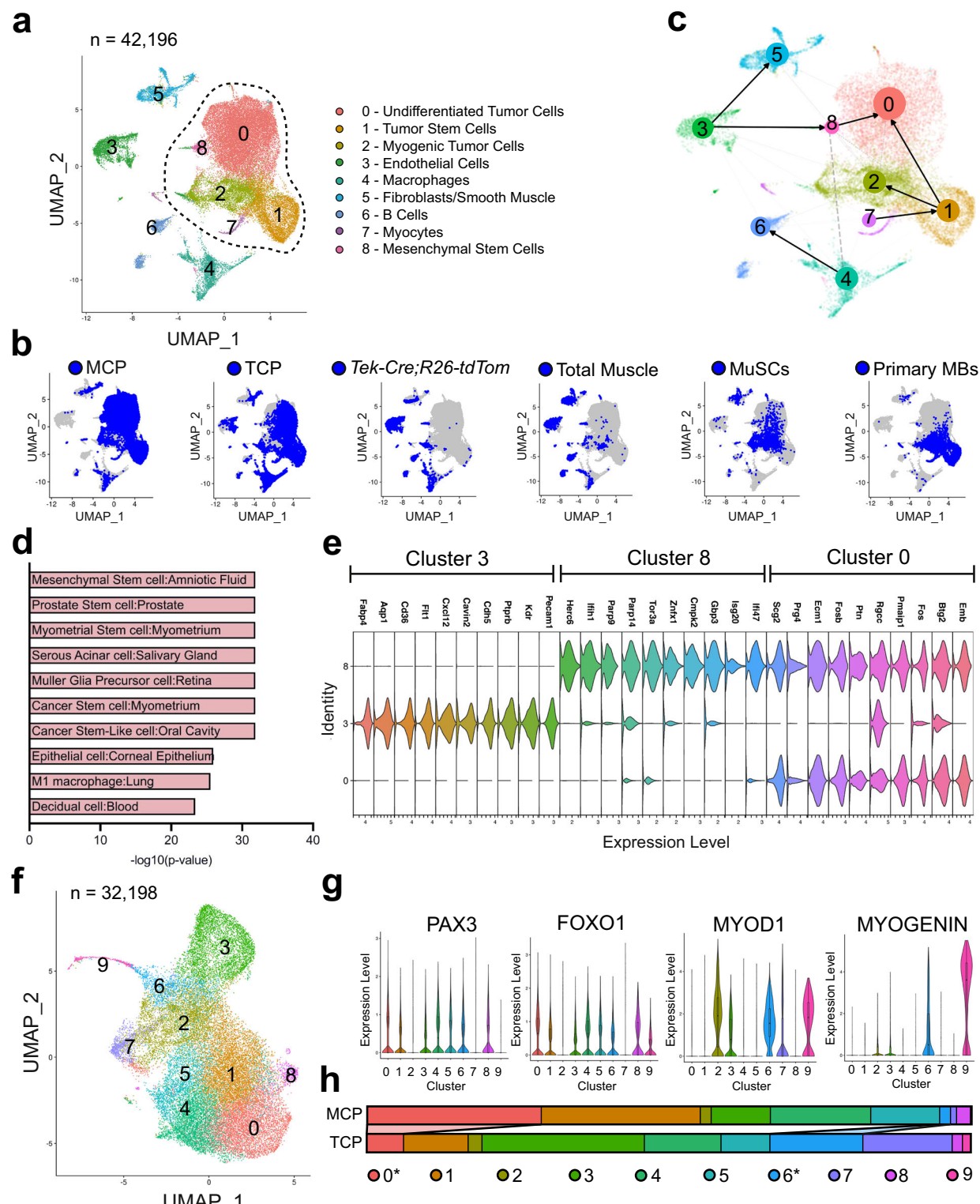

**Fig. 3 | FP-RMS tumors from TCP and MCP mice are indistinguishable on a single-cell level. a** UMAP and cluster analysis of single-cell transcriptional profiles of TCP (*n* = 2), MCP (*n* = 3), Tom⁺ cells from *Tek-Cre;Tomato* SCM skeletal muscle (*n* = 1), and mouse muscle datasets (*n* = 2 per group). Tumor cells and myogenic cells within dashed line. **b** UMAP highlighted by sample (MuSCs = tissue-resident skeletal muscle stem cells, Primary MBs = committed primary myoblasts). **c** scVelo analysis showing RNA velocity dynamical modeling of cellular dynamics. **d** Top ten terms by CellMarker Augmented 2021 from genes defining cluster 8. **e** Violin plot of top ten expressed genes in clusters 3, 8, and 0. **f** UMAP and re-clustered analysis of

single-cell transcriptional profiles of tumor and muscle population. **g** Violin plots showing *Pax3, Foxo1, Myod1* and *Myogenin* expression across clusters from **f** (*n* = 32,198). The median is defined as the 50% quantile. The upper whisker represents the largest observation ≤ to the upper hinge (75% quantile) +1.5*IQR, where the IQR is the interquartile range, or the distance between the first and third quartiles. The lower whisker represents the smallest observation ≥ to the lower hinge (25% quantile) −1.5*IQR. **h** Proportions of clusters present in TCP and MCP samples. *P* values determined by Student's t-test (unpaired, two-tailed); *$P < 0.05$.

MCP, MuSC, *Tek-Cre;R26td-Tom*, total muscle, and primary MB scRNAseq data[43]. Similar to the human scRNAseq data, genes enriched in the "Proliferation" and "Muscle" transcriptional states were only expressed in distinct clusters while genes enriched in the Hypoxia, Interferon, Mesenchymal-like, and Apoptosis cell states were expressed in multiple overlapping clusters (Fig. 4a). Additionally, Wei et al. assigned cells which did not express any conserved transcriptional modules to a "ground" state. This lack of a distinct gene expression pattern abrogates our ability to positively identify a similar population in our murine model. Feature plots show genes defining the "Muscle" state are strongly enriched in clusters 2 and 7, which we had previously defined as "Myocyte/Myogenic Tumor Cell", while the "Proliferation" state genes are strongly enriched in cluster 1, which we had previously defined as "Tumor Stem Cell" (Fig. 4b). This suggests that the genes driving cluster separation in the integrated TCP and MCP tumor data are similar to the cell state defining genes in the human RMS tumors. Importantly, these feature plots also illustrate that all the cell state expression modules, with exception of the ground state, in the human scRNAseq data are represented in both TCP and MCP tumors (Fig. 4b). Extending the analysis to all genes defining the gene modules within the conserved human RMS clusters demonstrated that all genes from cell state expression modules are robustly represented across both the TCP and MCP tumors (Supplementary Data 22) (Supplementary Fig. 5a, b). Together these findings indicate that TCP tumors develop similar heterogenous subpopulations to human RMS and further demonstrate the fidelity of TCP tumors as a model of human RMS.

## FP-RMS tumors from different cells-of-origin share similar chromatin landscapes

Having observed that scRNAseq did not uncover differences illustrating the distinct origins of TCP and MCP tumors, we aimed to uncover whether chromatin conformation could reflect differences in the cells of origin. High-throughput chromatin conformation capture (Hi-C) determined the genome organization of both TCP and MCP tumors were very similar at multiple resolutions with the most differences being sex related (Fig. 5a)[46]. Additionally, chromatin surrounding both skeletal muscle and endothelial genes is organized similarly in TCP and MCP (Fig. 5a). Analysis of the loci containing the endothelial gene *Pecam1* (*Cd31*) and the myogenic gene *Myod1* show no difference in chromatin conformation between the TCP and MCP tumors (Fig. 5b).

P3F largely binds at intergenic or intronic regions suggesting that P3F functions at enhancers located at cis-regulatory elements to regulate gene expression[47]. To identify differences in enhancer and super-enhancer activity driven in TCP and MCP tumors, we used cleavage under targets and releasing using nuclease (CUT&RUN) for H3K27ac. Signal for H3K27ac at statistically enriched peaks genome-wide are highly consistent between TCP and MCP ($r^2 = 0.899$) (Fig. 5c) (Supplementary Data 23). To examine potential effects of the variation in TCP and MCP enhancers and promoters identified by H3K27ac CUT&RUN, we located the nearest gene to each H3K27ac peak and used DESeq2 to identify genes enriched in either TCP or MCP tumors and their relationship to differentially active enhancers[48]. Additionally, we located peaks near genes expressed in RMS tumors identified by groups at the NIH, St. Jude Children's Research Hospital, and the Broad Institute[38,49]. This revealed 2702 TCP-specific peaks (log₂ FC (MCP/TCP) ≤ −1 and adj. $P < 0.05$) and 1772 MCP-specific peaks (log2 FC (MCP/TCP) ≥ 1 and adj. $P < 0.05$) total (Fig. 5d). In TCP tumors 53 of the TCP-specific peaks were associated with core RMS genes that showed increased expression while in MCP tumors only 33 MCP-specific peaks were associated with core RMS genes that showed increased expression (Fig. 5d) (Supplementary Data 24). However, through GSEA we found that genes located near peaks found uniquely in TCP tumors are not significantly overrepresented in relevant ontology terms (Supplementary Data 25). To refine our analysis of H3K27ac, we specifically compared H3K27ac coverage at super-enhancers, identified with the

ROSE algorithm, and again found few differences in all genes and in RMS genes (Fig. 5e) (Supplementary Data 26)[50–52]. Again, we used DESeq2 to identify genes associated with super-enhancers specific to TCP and MCP and found few differences between the tumor types (Fig. 5f) (Supplementary Data 27). No lineage-associated gene sets were significantly enriched in genes associated with TCP-specific or MCP-specific super-enhancers (Supplementary Data 28). To determine if lineage specific transcription factor motifs were enriched in H3K27ac peaks called in TCP and MCP tumors, we generated a list of endothelial transcription factors and performed DNA binding motif analysis. This list was generated by compiling transcription factors identified as top marker genes for endothelial subtypes and tissue specific endothelial phenotypes in EDL and soleus muscles by Kalucka et al.[53]. Additionally, we included H3K27ac enrichment at *Myod1* motifs as a positive control. In line with the expression data, this motif analysis showed that there were minimal differences in the H3K27ac status of endothelial transcription factor motifs (Fig. 5g). Together, these data are consistent with P3F activating the expression of RMS genes despite the cell of origin.

## P3F causes transdifferentiation of human umbilical vein cells to FP-RMS

We have determined that P3F expression in murine endothelial cells reprograms and transforms cells to FP-RMS. Next, we aimed to define the capacity of human endothelial cells to transform into FP-RMS. We began by expressing P3F in human umbilical vein endothelial cells (HUVECs). By 48 h after lenti-P3F transduction we observed *MYOD1* and *MYOGENIN* expression in HUVECs by real-time PCR (Supplementary Fig. 6a). However, by 12 days post-transduction *MYOD1* expression was no longer detectable despite maintaining P3F expression. These data suggest these cells could not be transdifferentiated to FP-RMS by P3F expression alone (Supplementary Fig. 6b). These results align with previous work in human primary myoblasts which demonstrated that P3F is insufficient for tumorigenesis and requires the expression of MYCN and TERT for transformation[54]. To determine if MYCN and hTERT expression permits tumorigenesis of HUVECs with P3F co-expression, we transduced primary HUVECs with lentivirus driving expression of large T-antigen (T-Ag), hTERT, MYCN, and P3F (HUVEC-THMP cells). We expanded HUVEC-THMP cells in culture and injected cells into immunocompromised mice which developed tumors with 100% penetrance and are consistent with FP-RMS expressing characteristic MYOD1, MYOGENIN, and DESMIN (Fig. 6a–e). Additionally, when compared by RNA-seq to HUVECs transduced with T-Ag, TERT and MYCN, HUVEC-THMP xenograft tumors and patient FP-RMS tumors are both enriched for ARMS gene sets by GSEA (Fig. 6f, g) This illustrates that P3F has potential to transdifferentiate human ECs into FP-RMS.

## P3F blocks endothelial differentiation and reprograms human iPSCs into FP-RMS

Our prior work illustrates that an endothelial cell (EC) progenitor, not the mature EC, is an origin of FN-RMS[22]. We leveraged directed differentiation of induced pluripotent stem cells (iPSCs) to direct P3F expression along endothelial differentiation (Fig. 7a). The original reports of the PAX3-FOXO1 mouse models eloquently demonstrated that only 1 out of 200 mice expressing *Myf6-Cre;Pax3^{P3Fm/P3Fm}* developed tumors with wild type *Trp53*[26]. Upon knockout of either *Trp53* or *Cdkn2a*, mice developed tumors with 100% penetrance[55]. Therefore, we generated *TP53* knockout (p53^{KO}) BJFF.6 iPSCs with CRISPR-Cas9 to limit cell apoptosis after P3F expression (Supplementary Fig. 7a–d). We confirmed our directed differentiation protocol differentiates iPSCs to the endothelial cells expressing (*CD31, VE-CADHERIN, CD34*) and form endothelial tubes in culture (Supplementary Fig. 8a–c). Enforced expression of P3F in p53^{KO} iPSCs directed to differentiate into ECs downregulates expression of endothelial genes (*CD31, VE-CADHERIN, CD34*) and increased muscle gene expression (*MYOD1*) measured by

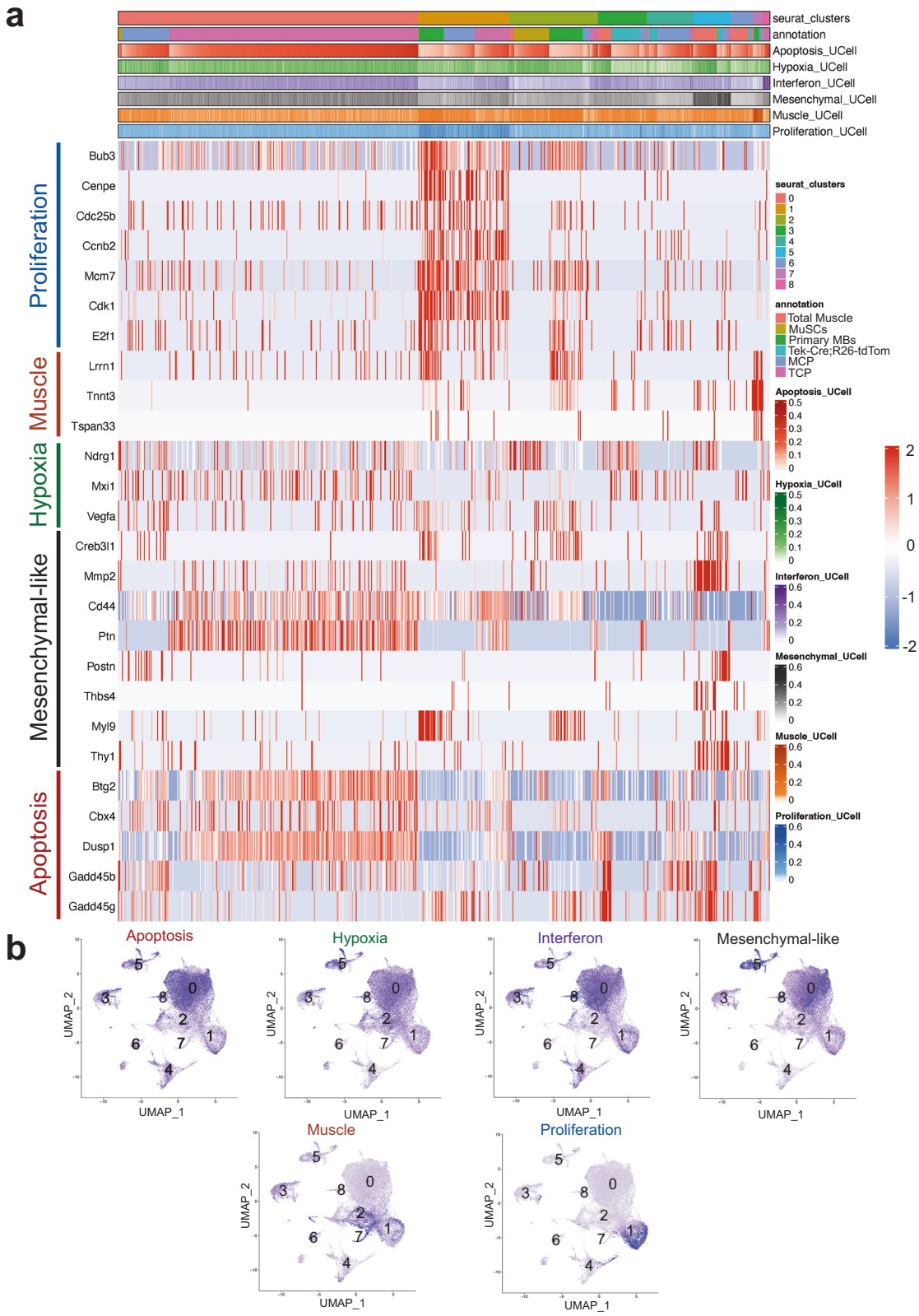

**Fig. 4 | TCP and MCP mouse FP-RMS tumors are comprised of cell states identified in human FP-RMS tumors. a** scRNAseq gene expression of representative genes from the cell state signature gene modules identified in human RMS PDXs (Proliferation, Muscle, Hypoxia, Mesenchymal-like and Apoptosis) derived from Wei et al. (Fig. 1C)[43] in mouse MCP and TCP tumors, primary myoblasts, muscle stem cells, total muscle, and Tom+ cells from *Tek-Cre;R26-tdTomato* muscle. Heatmap illustrates single cells from mouse samples (*x*-axis) and the normalized gene expression in cell state signature gene modules (y-axis). UCell signature score[103] is calculated for every module and visualized in the color bar above the heatmap. **b** Feature plot UMAPs illustrating expression of conserved cell state gene modules derived from Wei et al.[43] across integrated scRNAseq data from mouse MCP and TCP tumors, primary myoblasts, muscle stem cells, total muscle, and Tom+ cells from *Tek-Cre;R26-tdTomato* muscle, with cluster locations labeled.

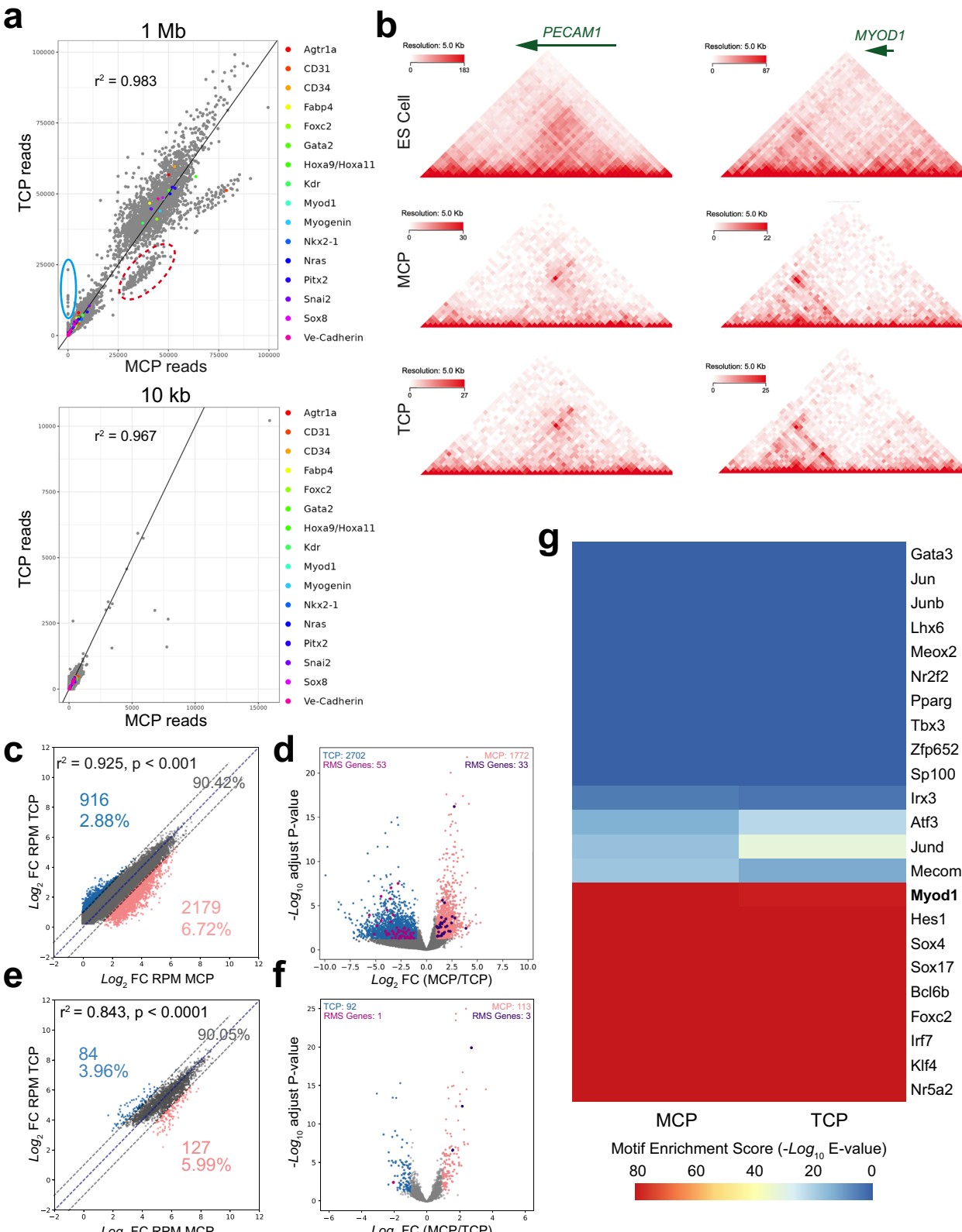

real-time PCR, flow cytometry, and IF (Fig. 7b–e) (Supplementary Fig. 8d, e). P3F expressing p53$^{KO}$ iPSCs but not control cells acquire the capacity to express MyHC and form myotubes, indicating cellular reprogramming into a myogenic cell (Fig. 7f). Additionally, P3F expressing p53$^{KO}$ iPSCs showed increased proliferation compared to control cells (Fig. 7g). We next aimed to determine if these cells form FP-RMS tumors in vivo. All ten immunocompromised mice injected

with four million P3F expressing iPSCs in the hindlimb formed tumors with histology and IHC positive for MYOD1, MYOGENIN, and DESMIN staining consistent with FP-RMS (Fig. 7h, k). Gene set enrichment analysis of xenograft tumors compared to parental p53$^{KO}$ iPSCs revealed strong enrichment of the ARMS gene set (REN_ALVEOLAR_-RHABDOMYOSARCOMA_UP) like that of FP-RMS tumors from the Pediatric Cancer Genome Project compared to p53$^{KO}$ iPSCs (Fig. 7i, j).

**Fig. 5 | FP-RMS tumors from different cells of origin have similar chromatin landscapes. a** Scatterplot of TCP ($n = 2$) and MCP ($n = 2$) HiC reads connecting pairs of bins at 1 Mb (top) and 10 kb (bottom) resolution. Representative scatterplot from TCP 79 vs MCP 76 shown. Read counts between bin-pairs where at least one bin overlaps a chromosomal location of a selected gene are highlighted. Bin-pairs within the red dotted line involve bins located on the X chromosome and bin-pairs located within the blue solid line involve bins located on the Y chromosome. **b** Heatmap of HiC connectivity read counts at the *Pecam1* and *Myod1* loci in embryonic stem cells (ES cells)[104], TCP, and MCP. Representative data from TCP 79 and MCP 43 shown. **c** Scatterplot comparison of differences in H3K27ac read coverage at H3K27ac peaks. Collapsed peaks showing biased coverage (TPM absolute linear average fold-change > 2) in TCP ($n = 3$) in blue and MCP ($n = 4$) in pink ($P = 4.62 \times 10^{-10}$, two-tailed Pearson's r correlation). **d** Volcano plot comparing coverage at collapsed peaks in TCP (blue) and MCP (pink). Points colored blue or

pink if statistically significantly differentially covered (DEseq2 $P < 0.05$ and absolute value log2 fold-change > 1; DEseq2 uses a Wald test and internal multiple testing amelioration). Location of RMS genes in dark pink and purple for TCP and MCP, respectively. **e** Scatterplot comparison of differences in H3K27ac read coverage at super-enhancers. Collapsed super-enhancers significantly selectively covered (TPM absolute log2 fold-change >2) in TCP in blue and MCP in pink ($P = 2.2 \times 10^{-16}$, two-tailed Pearson's r correlation). **f** Volcano plot comparing coverage at collapsed super-enhancers in TCP (blue) and MCP (pink). Points colored blue or pink if statistically significantly differentially covered (DEseq2 $P < 0.05$ and absolute value log2 fold-change > 1; DEseq2 uses a Wald test and internal multiple testing amelioration). Location of super-enhancers associated with RMS genes in dark pink and purple for TCP and MCP, respectively. **g** Heatmap displaying statistical enrichment of endothelial TF motif hits in H3K27ac CUT&RUN data. *Myod1* (not an endothelial gene) is included as a positive control for H3K27ac enrichment in bold.

This illustrates that EC derived tumors expressing P3F are FP-RMS with similar gene expression profiles to human ARMS tumors. To determine if we could identify residual expression of endothelial cell transcription factors in transformed P3F expressing p53^KO iPSCs and xenografts, we measured expression of endothelial specific transcription factor gene sets from a murine endothelial atlas and from PanglaoDB[53,56]. This analysis demonstrated that the P3F expressing p53^KO iPSCs and xenografts had little to no expression of either endothelial gene set, on par with the expression levels found in parental undifferentiated iPSCs and FP-RMS cell lines (Fig. 7l). Additionally, we compared the expression of these endothelial gene sets in HUVECs transduced with T antigen, hTERT, and MYCN to HUVEC-THMP Xenografts and observed that the HUVEC-THMP xenografts also no longer expressed either endothelial gene set (Fig. 7l). These findings illustrate that differentiating human endothelial cells possess the potential to transdifferentiate into FP-RMS through expression of P3F.

This malignant myogenic phenotype is unique to P3F and is not observed in cells transduced with PAX3 or FOXO1 alone (Supplementary Fig. 9). The p53^KO iPSCs transduced with lenti-*PAX3* show signs of myogenic reprogramming with expression of *MYOD1* and formation of myotubes with myogenic differentiation (Supplementary Fig. 9a–d) although markedly less than observed with P3F. However, no mice injected with p53^KO iPSCs cells transduced with lenti-*PAX3* developed tumors observed for up to 282 days post injection. Lenti-*FOXO1* transduced p53^KO iPSCs stop proliferating before the completion of the 14-day endothelial differentiation protocol abrogating further study. Given that MYOD1 alone is sufficient to reprogram murine fibroblasts, we also tested whether enforced expression of MYOD1 during the endothelial differentiation protocol would be sufficient to fully reprogram the iPSCs[19]. Unlike P3F, MYOD1 expression was not sufficient to block endothelial differentiation or to completely reprogram cells as evidenced by co-localization of MYOD1 and CD31 immunofluorescence in differentiated cells, increased total proportion of CD31+ cells, and by persistence of endothelial surface markers CD31, CD34, and VE-CADHERIN in lenti-*MYOD1* transduced cells (Supplementary Fig. 9e, f). Additionally, the MYOD1 expressing cells displayed decreased viability, proliferation, and focus formation compared to the P3F expressing cells, thus eliminating the ability to test tumorigenicity in a xenograft model (Supplementary Fig. 9g–i). These data demonstrate that unlike P3F, enforced expression of MYOD1, PAX3, or FOXO1 alone are not sufficient to drive reprogramming and transformation of p53^KO iPSCs into FP-RMS.

### P3F maintains myogenic gene expression and proliferative drive in transformed p53^KO iPSCs

We have illustrated that P3F expression is capable of transdifferentiating and transforming endothelial cells into FP-RMS; however, it is unclear if continued expression of P3F is required for FP-RMS tumor maintenance. To investigate how P3F depletion after transformation affects proliferation and gene expression, we leveraged our P3F

expressing p53^KO iPSC systems. We knocked down expression of P3F using lentivirus expressing a shRNA targeting the breakpoint of the *PAX3;FOXO1* fusion (shP3F) and a vector expressing a shRNA targeted against firefly luciferase (shLuc) as a control[57]. Transduction with pLKO-shP3F resulted in rapid depletion of P3F transcript and protein (Supplementary Fig 10a, b). This was accompanied by loss of expression of the muscle lineage transcription factors *MYOD1*, *MYF5*, and *MYF6* when compared to shLuc transduced cells (Supplementary Fig 10c, d). Furthermore, pLKO-shP3F transduced cells showed significantly reduced proliferation and were unable to form foci in a focus formation assay (Supplementary Fig. 10e, f). These findings suggest that continued P3F expression is required to maintain the increased proliferation and myogenic gene expression profile of the transformed P3F expressing p53^KO iPSCs.

### P3F transforms p53^KO iPSCs by promoting epigenetic modifications

The striking transformation of iPSCs undergoing directed differentiation into endothelial cells to FP-RMS upon the addition of P3F provided us with a unique opportunity to identify a prospective mechanism by which P3F mediates transformation. We determined P3F genome-wide occupancy through performing anti-FOXO1 CUT&RUN. In addition to P3F occupancy, we interrogated active enhancer and promoter regions with H3K27ac CUT&RUN and correlated it to transcriptional output with RNAseq on matched samples before (day 0) and after (day 15) transformation into FP-RMS, with untransduced endothelial differentiated iPSCs as a control. We observed decreased expression of the TF *ERG*, a critical regulator of endothelial lineage specification and homeostasis, in transformed P3F expressing p53^KO iPSCs (Fig. 8a)[58,59]. Interestingly, this was coupled with loss of H3K27ac signal at an internal *ERG* super-enhancer, but no FOXO1 binding (Fig. 8a). This suggests that P3F engages a chromatin regulatory program that reduces H3K27ac modifications at this locus and thus indirectly prevents expression of a crucial endothelial specification factor. This is consistent with previous flow cytometry and qPCR data demonstrating loss of VE-CADHERIN in transformed P3F expressing p53^KO iPSCs, as VE-CADHERIN is a direct target of ERG. Additionally, we observed FOXO1 binding and H3K27ac-marked chromatin at known proximal and distal enhancer elements of *MYOD1*, *MYF5*, and *MYF6* only in cells expressing P3F (Fig. 8b, c). RNAseq showed that FOXO1 binding at these loci was accompanied with a strong increase in expression of *MYOD1*, *MYF5*, and *MYF6* compared to untransduced cells (Fig. 8b, c). These data, coupled with the P3F depletion observations, suggest that direct binding of P3F at super-enhancers of master myogenic transcription factors promotes their expression and mediates reprogramming to a myogenic tumor cell.

## Discussion
Due to intrinsic tumor cell histology and transcriptional programs, the cellular origins of FP-RMS are inferred to be skeletal muscle progenitor

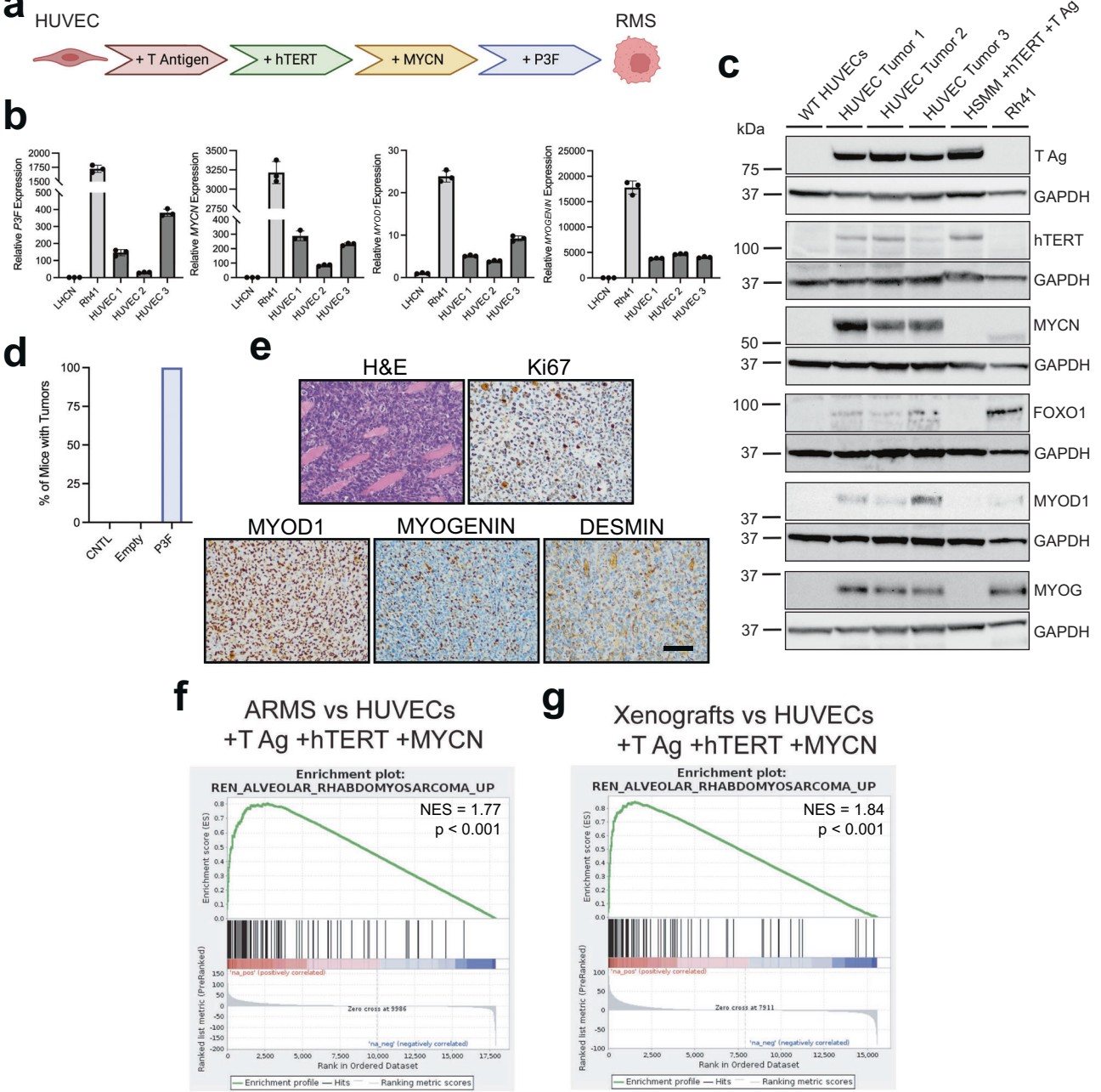

**Fig. 6 | Transdifferentiation of human umbilical vein cells to FP-RMS.**
**a** Schematic of HUVEC transduction generating HUVEC-THMP cells. Created with BioRender.com. **b** Gene expression by real-time PCR of *PAX3-FOXO1, MYOD1*, and *MYOGENIN* in three HUVEC-THMP tumors, human immortalized myoblasts (LHCNs), and human FP-RMS cell line (Rh41) (*n* = 3 technical replicates for each sample shown). **c** Immunoblots of T- Ag, hTERT, MYCN, FOXO1, MYOD1, MYO-GENIN, and corresponding GAPDH on WT HUVECs, three HUVEC-THMP tumors, and Rh41 cells. GAPDH loading controls were ran on the same blot as the corresponding antibody above it. Uncropped and unprocessed scans of immunoblots are included in the Source Data file. **d** Percent of SCID beige mice with tumors after 125 days post injection with four million untransduced HUVEC cells (CNTL), or HUVECs expressing T-Ag, hTERT and MYCN with lenti-Empty or lenti-P3F. **e** Representative histology of HUVEC-THMP tumors staining for H&E, MYOD1, MYOGENIN, and DESMIN. N = three separate tumors. Scale bar = 100 μm. **f** Enrichment plot of FP-RMS gene signatures (REN_ALVEOLAR_-RHABDOMYOSARCOMA_UP) for PCGP FP-RMS tumors compared to HUVECs expressing T-Ag, hTERT and MYCN. **g** Enrichment plot of FP-RMS gene signatures (REN_ALVEOLAR_RHABDOMYOSARCOMA_UP) for xenograft HUVEC-THMP tumors expressing T-Ag, hTERT and MYCN with lenti-P3F compared to HUVECs expressing T-Ag, hTERT and MYCN. *P* values determined by Student's t-test (unpaired, two-tailed); *$P < 0.05$. Data represented as mean+/- SEM. Source data are provided as a Source Data file.

cells arrested in myogenic development[8]. However, FP-RMS tumors can occur throughout the body in locations devoid of skeletal muscle, including a subset of very high-risk patients with widespread disease and bone marrow involvement without a primary tumor[9,11–13,60]. Additionally, the t(2;13)(q35:q14) chromosomal translocation brings the downstream regulatory elements of *FOXO1* that drive endothelial cell type expression of *FOXO1* in proximity to the *PAX3* promoter, thus providing the potential of P3F expression in endothelial cells[20]. This combination of regulatory elements upstream of *PAX3* and downstream of *FOXO1* provides a unique genomic architecture that allows dysregulated expression of lineage factors giving a non-myogenic endothelial cell the ability to transdifferentiate into FP-RMS driven by

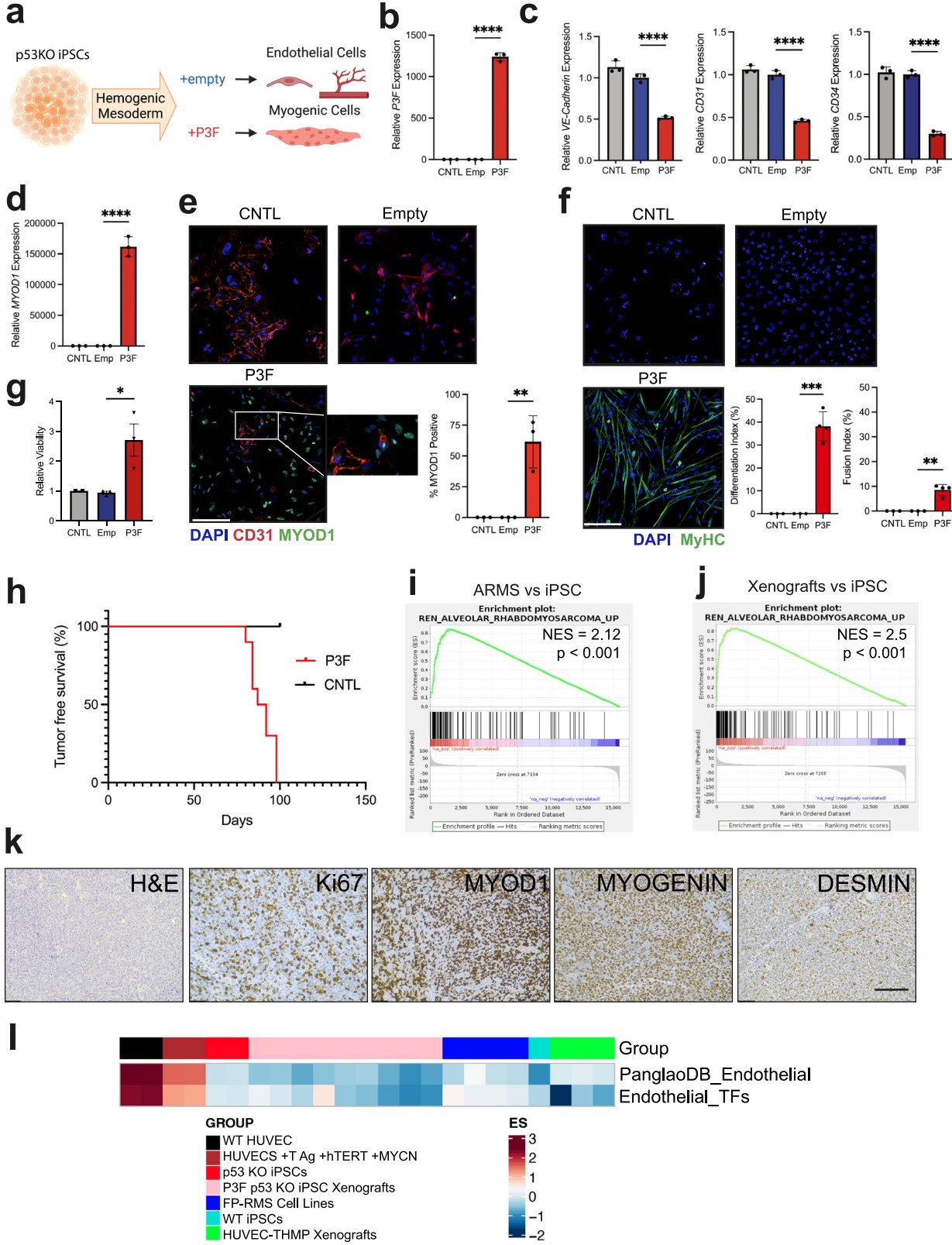

P3F expression. The data presented here suggest that FP-RMS can originate from an endothelial origin in addition to skeletal muscle origins. This could explain the origin of aggressive tumors without a known myogenic primary tumor.

In this study, we generate multiple mouse models driven by forced P3F expression in non-myogenic cells that accurately and consistently develop tumors with histologic and molecular features of

human FP-RMS. P3F expression in both *Myf6-Cre* expressing myogenic cells and *Tek-Cre* expressing non-myogenic cells creates a nearly identical chromatin landscape. In the mouse model used for this study, P3F expression is driven by the *Pax3* promoter and influenced by the regulatory elements of *Foxo1* knocked into the *Pax3* locus[26]. However, this model lacks downstream regulatory elements over 100 kb from the TSS required to drive FOXO1 expression in vascular endothelial

**Fig. 7 | P3F blocks endothelial differentiation and reprograms human iPSCs into FP-RMS. a** Schematic of iPSC experiment. Created with BioRender.com. **(b-d)** Representative technical replicates from one of three experimental replicates after 14-day endothelial differentiation protocol. Additional experimental replicate data provided in Source Data file. CNTL cells are untransduced p53[KO] iPSCs. Gene expression by real-time PCR for **(b)** *PAX3-FOXO1*, ($n = 3$, $P < 0.0001$) **c** *VE-Cadherin* (*KDR*) ($n = 3$, $P < 0.0001$), *CD31* (*PECAM1*) ($n = 3$, $P < 0.0001$), *CD34* ($n = 3$, $P < 0.0001$). **d** Gene expression by real-time PCR for *MYOD1* ($n = 3$, $P < 0.0001$). **e** Representative IF for MYOD1 (green), CD31 (red), DAPI (blue) ($n = 3$, $P = 0.0073$). Scale bar = 50 µm. **f** Representative IF for MyHC (green). Differentiation index = percent of MyHC[+]/DAPI[+] cells ($n = 3$ for CNTL and Emp, $n = 4$ for P3F, $P = 0.0002$), and fusion index = percent of MyHC[+]/DAPI[+] cells with two or more nuclei ($P = 0.0016$). DAPI (blue). Scale bar = 50 µm. **g** Cell viability by Cell Titer Glow assay comparing CNTL, empty, and P3F cells after 48 h (CNTL $n = 2$, Emp and P3F $n = 3$, $P = 0.0315$). **h** Kaplan-Meier tumor free survival curve for SCID beige mice injected with 4 million cells. (CNTL = 3, P3F = 10). Enrichment plot of FP-RMS gene signatures (REN_ALVEOLAR_RHABDOMYOSARCOMA_UP) for **(i)** PCGP ARMS, and **(j)** iPSC tumors, compared to BJFF.6 p53[KO] iPSCs. **k** Representative histology of iPSC tumors stained with H&E, DESMIN, Ki67, MYOD1 and MYOGENIN. $N = 10$. Scale bar = 100 µm. $P$ values determined by Student's *t*-test (unpaired, two-tailed); \*\*$P < 0.01$, \*\*\*$P < 0.001$, \*\*\*\*$P < 0.0001$. Data represented as mean+/- SEM. **l** heatmap comparing the gene expression of endothelial cells (WT HUVECs and HUVECs expressing T Ag, hTERT and MYCN), p53[KO] iPSCs, transformed P3F expressing p53[KO] iPSC derived xenografts (P3F p53 KO iPSC Xenografts), FP-RMS cell lines (RH28, RH18, RH30_1, RH30_2), WT BJFF.6 iPSCs (WT iPSCs), and HUVEC-THMP derived xenograft tumors. Single sample gene set enrichment analysis (ssGSEA) was used to score an endothelial cell transcription factor list generated by comparing the mutually expressed top 50 genes identified by Kalucka et al. 2020[53] and the PanglaoDB_Endothelial gene list downloaded from PanglaoDB[56]. Source data are provided as a Source Data file.

cells[20]. Forcing the expression of P3F using this genetically engineered mouse model in endothelial cells that arose from a PAX3+ progenitor in the dermomyotome circumvents the lack of distal *Foxo1* cis-regulatory elements in the *Pax3:Foxo1* allele to drive P3F expression robustly and completely in this cell type. As a newly added hallmark of cancer, more studies are showing the importance of phenotypic plasticity in tumorigenesis[61]. This work highlights transdifferentiation as a mechanism underlying FP-RMS transformation.

Intriguingly, we found P3F reprograms *aP2-Cre* expressing endothelial cells located in the skeletal muscle interstitium to functional muscle stem cells. Upon P3F expression, these cells express PAX7, a marker of muscle stem cells and contribute to muscle regeneration after injury. P3F impedes myogenic differentiation by maintaining the *MYOD1* super-enhancer that keeps cells in an undifferentiated state[17]. However, here we show these cells can differentiate and express the terminal muscle differentiation marker MyHC. We hypothesize that the cell type and context of P3F expression is important when determining its effect on differentiation. Although our lineage tracing shows *aP2-Cre, Fabp4-Cre*, and *Tek-Cre* share expression in the same cell types in adult tissue, P3F cooperates with *Cdkn2a* loss in each model to give rise to a unique phenotype. One potential explanation for the disparate phenotypes is the timing of Cre expression during endothelial development. Tomato expression is an indelible mark in this system, rendering it impossible to determine when in the lifespan of the cell Cre-mediated recombination occurred. Further investigation into the embryonic origins of *aP2-Cre, Fabp4-Cre*, and *Tek-Cre* may provide insight into the mechanism governing why different phenotypes arise in response to enforced P3F expression. Our work illustrates that P3F maintains the ability to reprogram *aP2-Cre* expressing cells without transforming them, highlighting the intricate balance between transformation and cell state change. Understanding the fundamental mechanisms that define the balance between transformation and lineage fidelity will also prove useful in the context of muscle stem cells and associated pathologic states, such as muscular dystrophies, multiple sclerosis, and muscular atrophy. Given P3F is a powerful oncoprotein, P3F will not be a viable candidate to treat these conditions, but the underlying process and mechanisms may be fruitful fields for therapeutic development.

P3F and P7F are comprised of individual pioneer factors, PAX3, or its family member PAX7, and FOXO1[62–65]. The t(2;13)(q35:q14) chromosomal translocation brings together tissue specific enhancers driving FOXO1 expression with the DNA binding domain of PAX3 to drive myogenic programs in endothelial cells[20]. PAX3 alone is unable to maintain transcriptional regulatory complexes seen when P3F is present, showing the importance of the oncofusion protein in regulating gene expression. P3F creates novel TADs and directly activates the myogenic super-enhancer regulating *MYOD1*. MYOD1 then initiates MYOG expression, another hallmark of FP-RMS[66]. P3F is only found in RMS which is explained by two hypotheses proposed by Vicente-

Garcia et al.: (1) P3F can be expressed in multiple cell types, but only those harboring the necessary regulatory factors are capable of transformation; or (2) only specific cell types can express P3F because of co-transcription of both loci involved[20]. Our experiments aim to address the first hypothesis and show that the effect of P3F varies depending on the context of expression. For example, forcing P3F expression in *aP2-Cre* expressing cells causes reprogramming to functional muscle stem cells but not RMS, while in *Tek-Cre* expressing cells, the same genetic perturbations cause transformation to FP-RMS. Additionally, Keller et al. demonstrates P3F exclusively transforms more differentiated *Myf6-Cre* expressing cells to but not muscle stem cells expressing *Pax7-Cre*[67]. Together, this shows that combination of cellular context and mutagenic drivers are important when studying the effect of oncogenes.

Here, we detail how genetically engineered mouse models and human cell lines allow us to systematically dissect P3F function in a non-muscle progenitor cell and explore the core dependencies of this oncogene in RMS oncogenesis in both species. FP-RMS tumors from TCP and MCP mice accurately recapitulate human disease through histologic profiling and gene expression. While TCP and MCP tumors originate from different cell types, they share nearly indistinguishable histology, gene expression, and chromatin architecture. This shows that it is potentially misleading to assume the cell of origin based on the characteristics of the tumor cell. Additionally, we show that HUVECs and human iPSCs with TP53 loss undergoing directed differentiation to endothelial cells can transdifferentiate to myogenic cells expressing MYOD1, MYOG, and DESMIN, and form FP-RMS tumors with gene expression signatures matching patient tumors. Furthermore, the iPSC model presented provides a system to identify prospective mechanisms of transformation by comparing gene expression and chromatin landscape changes in transformed FP-RMS cells to their corresponding endothelial differentiated counterparts which arose from the same cell of origin. This revealed that P3F indirectly promotes repression of endothelial specification factor expression and directly promotes the transcription of master myogenic transcription factors. Our data suggest that both roles are required for transformation into FP-RMS. Enforced expression of MYOD1 alone was not sufficient to block endothelial differentiation and did not lead to malignant transformation. Depletion of P3F after transformation lead to a complete loss of myogenic gene expression and significant reduction in proliferation/focus forming capacity consistent with prior work[49,57]. Further interrogation of gene expression and chromatin landscape dynamics through the time course of iPSC transformation could identify when key regulators of cell fate and oncogenicity are activated or repressed in P3F expressing cells. This will be crucial to deciphering the stepwise progression to transformation of this aggressive tumor and will be a valuable resource to the field when determining therapeutic vulnerabilities and developing targeted therapies to improve patient outcomes. The advent of

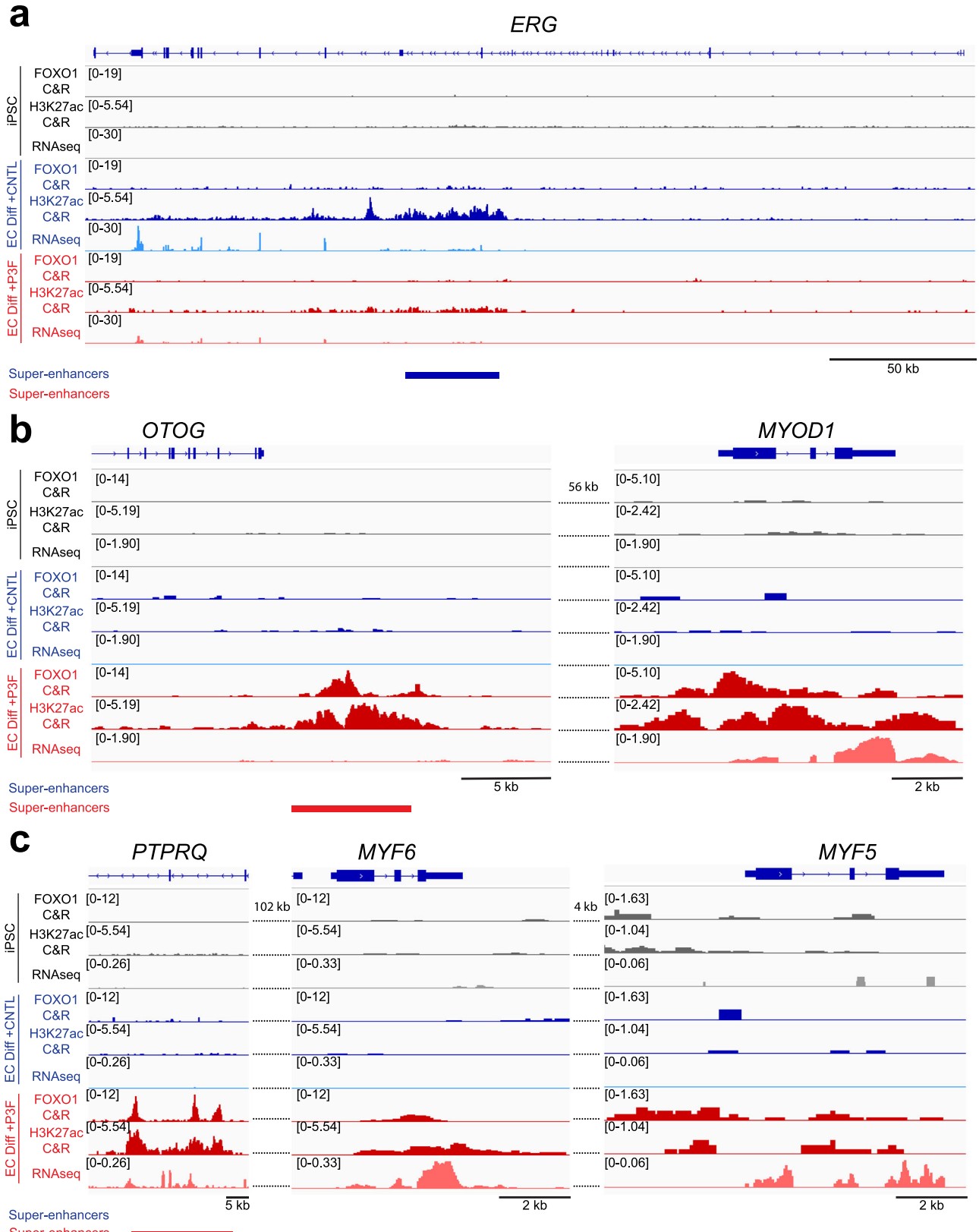

**Fig. 8 | P3F transforms p53^KO iPSCs by promoting epigenetic modifications.** Analysis from one experimental replicate after 14-day endothelial differentiation protocol. **a**–**c** Groupscaled CUT&RUN (C&R) tracks depicting genomic coverage of H3K27ac-marks and FOXO1 reads with matched RNAseq tracks at Day 0 in untransduced p53^KO iPSCs (grey), Day 15 in untransduced endothelial differentiated p53^KO iPSCs (blue), and Day 15 in transformed P3F expressing p53^KO iPSCs (red). Super-enhancers called by ROSE are shown for (**a**) *ERG*, (**b**) *MYOD1*, (**c**) *MYF5*, and *MYF6* loci.

proximity induced targeted protein degradation provides a potential avenue to target the P3F oncoprotein; however, further mechanistic understanding of P3F function and necessity for the malignant FP-RMS phenotype is needed to assist in the development and interpretation of P3F-depletion approaches.

FP-RMS tumors develop throughout the body, including tissues devoid of skeletal muscle, highlighting the potential for non-myogenic origins. Here, we identified endothelial cells as a cell of origin for FP-RMS. FP-RMS tumors from endothelial origins are nearly indistinguishable from FP-RMS tumors derived from myogenic cells. Differentiation therapy is an intriguing alternative treatment avenue which has been extensively studied in neuroblastoma, another pediatric solid tumor that can appear as an arrested state of normal development[68]. This study highlights that patient FP-RMS tumors may come from differing cells of origin and thus could exhibit unique responses to differentiation therapy. FP-RMS in the head and neck originating from endothelial cells may not respond to these therapies even though they resemble tumors from a myogenic cell of origin. Future work investigating therapeutic vulnerabilities of FP-RMS tumors from different cells of origin will help inform clinical trial design to better treat patients with FP-RMS.

This study identifies mouse and human endothelial cells as a cell of origin for FP-RMS. While these tumors are indistinguishable from FP-RMS tumors originating from myogenic cells by both gene expression and chromatin landscape, it is possible they possess differences not made evident by our analysis. Additionally, we cannot be certain of the cell of origin of patient tumors and can only show the similarity of the models analyzed in this study to human FP-RMS.

## Methods

### Genetically engineered mouse models

All mouse strains are reported: *aP2-Cre*[69], *R26-tdTomato* (#7914), The Jackson Laboratory (JAX)[70], *Pax3*$^{Pax3-Foxo1}$ [26], *Cdkn2a*$^{Flox}$ (Nabeel Bardeesy[71]), *Fabp4-Cre* (#5069, The Jackson Laboratory)[72], *Tek-Cre*[35], *Myf6-Cre*[73]. For Kaplan-Meier survival analysis, animals were observed from birth and sacrificed at humane endpoints, such as showing signs of obvious tumor burden or other distress. Maximal tumor size/burden permitted by the SJCRH Institutional Animal Care and Use Committee is 2 cm³, maximal tumor size/burden was not exceeded in this study. Full necropsies were performed. All mice had consistent access to food and water and were housed at ambient temperature (20–25 C) and humidity (40–60%) with 12-h light/12-h dark cycles. All animal experiments were reviewed and approved by the SJCRH Institutional Animal Care and Use Committee.

### Cell lines

293T (Martine Roussel, SJCRH) cells were maintained in DMEM (#SH32043, HyClone) supplemented with 10% fetal bovine serum (FBS, HyClone) and 1% antibiotic/antimycotic. HUVECs (LifeLine Cell Technologies, FC-0044) were maintained in VascuLife VEGF Endothelial Medium Complete Kit (LifeLine Cell Technologies, LL-0003). Primary HUVECs were immortalized by retroviral transduction with large T (genomic) antigen (#1778, Addgene, Bob Weinberg) and telomerase reverse transcriptase (pBABE-hygro-hTERT) (#1773, Addgene)[74]. BJFF.6 Human iPSCs were maintained in mTeSR Plus (Stemcell Technologies, 100–0276) on growth factor reduced Matrigel basement membrane matrix (Corning, 354230). All cell lines were maintained in a humidified incubator at constant 37 °C and 5% CO$_2$.

### Molecular cloning and viral transduction

pSIN-EF2-IRES-mCherry was generated by PCR amplifying mCherry from pmRi-mCherry (Clontech, #631119) using oligonucleotides adding 5′ *Nde*I (5′-GGGCATATGGTGAGCAA GGGCGAGG-3′) and 3′ *Cla*I (5′-GGGGATCGATCTAGTTTCCGGACTTGT-3′) restriction sites digesting

the PCR product with *Nde*I and *Cla*I and ligating to similarly digested pSIN-EF2-IRES-Blast[75]. pcDNA-PAX3-FOXO1 (Rene Galindo) was subcloned into the pSIN-EF2-IRES-mCherry lentiviral vector. Plasmids encoding shRNAs were generated by ligating annealed overlapping oligonucleotides targeting the breakpoint of PAX3;FOXO1 (shP3F) (5′-CCGGTCTCACCTCAGAATTCAATTCCTCGAGGAATTGAATTCTGAGG TGAGATTTTTG-3′ and 5′-AATTCAAAAATCTCACCTCAGAATTCAATT CCTCGAGGAATTGAATTCTGAGGTGAGA-3′) and Firefly Luciferase (shLuc) (5′- CCGGCTTACGCTGAGTACTTCGACTCGAGTCGAAGTACT-CAGCGTAAGTTTTTG-3′) and (5′- AATTCAAAAACTTACGCTGAGTAC TTCGACTCGAGTCGAAGTACTCAGCGTAAG-3′) to *Age*I and *EcoR*I digested pLKO.1-puro (#8453, Addgene) to generate pLKO.1-shP3F-puro and pLKO-shLuc-puro. The resulting plasmids were digested with *Kpn*I and *Spe*I to excise the puromyocin resistance gene and ligate to a similarly digested EGFP fragment from pLKO.3G (#14748, Addgene) generating the pLKO-shP3F-EGFP and pLKO-shLuc-EGFP plasmids. The pLKO-shP3F-EGFP and pLKO-shLuc-EGFP plasmids were co-transfected with packaging plasmids pMD2.G VSV-G envelope (#12259, Addgene) and psPAX2 packaging vector (#12260, Addgene) into 293 T cells with FuGENE6 (Promega #E2691) according to manufacturer's protocol. iPSCs were transduced with conditioned EGM Bullet Kit media containing viral particles after filtering (0.22 μM).

### Generation of p53$^{KO}$ BJFF.6 iPSCs

hTP53$^{-/-}$ BJFF.6 iPSCs were generated using CRISPR-Cas9 technology. Briefly, BJFF.6 iPSCs were pretreated for 1 h in StemFlex (Thermo Fisher Scientific) supplemented with 1X RevitaCell (Thermo Fisher Scientific). Approximately 1X10⁶ cells were nucleofected (Lonza, 4D-Nucleofector™ X-unit) with precomplexed ribonuclear proteins (RNPs) consisting of 150 pmol of chemically modified sgRNA (5′ UCCU-CAGCAUCUUAUCCGAG 3′; Synthego), 50 pmol of Cas9 protein (St. Jude Protein Production Core), and 500 ng of pMaxGFP (Lonza) in a small (20 μl) cuvette using solution P3 and program CA-137 according to the manufacturer's recommended protocol. Cells were sorted five days post nucleofection for single cells by FACS at the Flow Cytometry and Cell Sorting Shared Resource (St. Jude) for transfected (GFP+) cells and plated into prewarmed (37C) StemFlex media supplemented with 1X CloneR (Stem Cell Technologies) into Vitronectin XF (Stem Cell Technologies) coated 96-well plates. Clones were screened for out-of-frame indels via targeted deep sequencing on a MiSeq Illumina sequencer as previously described[76]. Briefly, gene specific primers (TP53.F-5′ CTACACGACGCTCTTCCGATCTGGC GCTGCCCCCACCAT-GAG 3′ and TP53.R- 5′ CAGACGTGTGCTCTTCCGATCTCTGGAGGG CCACTGACAACCACCCT 3′) with partial Ilumina adaptors (upper case) were used to amplify the region flanking the gRNA target site and then indexed by nested PCR with primers containing the remainder of the Ilumina adaptors (Indexing primer F- 5′ AATGATACGGCGA CCACCG AGATCTACACX$_6$ACACTCTTTCCCTACACGACGCTCTTC 3′ and Indexing primer R-5′ CAAGCAGAAGACGGCATACGAGATX$_{10}$GTGACTG GAGTTCAGACGTGTGCTC 3′; indexes shown as "X"s). Samples were demultiplexed using the index sequences, fastq files were generated, and NGS analysis of clones was performed using CRIS.py[77]. Knockout clones were identified, expanded, and sequence confirmed. Cell identity was authenticated using the PowerPlex® Fusion System (Promega) performed at the Hartwell Center (St. Jude) and tested negative for mycoplasma by the MycoAlert™Plus Mycoplasma Detection Kit (Lonza).

### Histology and IF

Dissected tissues were submerged in ice cold PBS. Frozen sections were prepared by fixing tissues 2 h in 4% paraformaldehyde (PFA) at 4 °C and cryoprotecting in 30% Sucrose, 2 mM MgCl$_2$. Muscle and tumors were snap frozen in 2-methyl-butane cooled in liquid nitrogen prior to sectioning using a conventional cryostat. Formalin-fixed paraffin-embedded (FFPE) sections were prepared by fixing tissues in 10%

neutral-buffered formaldehyde (NBF) prior to paraffin embedding. Hematoxylin and eosin (H&E) and immunohistochemistry staining were performed using standard procedures. Antibodies and antigen retrieval conditions are listed in Supplementary Data 1. Images captured on Leica DMi Thunder Imager inverted fluorescent microscope.

## Myogenic differentiation assay

Mononuclear cells were isolated from SCM of adult (>8-week-old) mice as previously described[22] and plated on 0.1% gelatin then grown until confluent in DMEM (#SH32043, HyClone) supplemented with 10% FBS (HyClone) and 1% PSA (#A5955, Sigma). Once confluent, cells were rinsed twice with PBS then cultured with DMEM (#SH32043, HyClone) supplemented with 2% horse serum and 1% PSA (#A5955, Sigma). Differentiated iPSCs were plated on 0.1% gelatin then grown until confluent in EGM BulletKit (CC-3124, Lonza) supplemented with 20 ng/mL bFGF (AF-100-18B, PeproTech), 1 μM CHIR-99021 (13122, Cayman Chemical), and 20 ng/mL VEGF (100-20, PeproTech) then transferred to LHCN diff media. Following differentiation, cells were fixed in 4% PFA for 15 min and immunofluorescence was performed according to standard techniques previously described[22] using antibodies listed in Supplementary Data 1. Images captured on Leica DMi Thunder Imager inverted fluorescent microscope.

## Flow cytometry and fluorescence-activated cell sorting

Muscle was dissected and digested as previously described[22], then stained for flow cytometry prior to analysis with a BD Biosciences Symphony analyzer for flow cytometry analysis and a BD Biosciences Aria cell sorter for cell sorting. In brief, muscle was mechanically and enzymatically digested in a solution of 2 U/mL Collagenase B (#11088831001, Roche), 2 U/mL Dispase II (#04942078001, Roche), 50 mM HEPES/KOH pH 7.4, and 150 mM NaCl for one hour at 37 °C, and cultured cells were lifted and dissociated with accutase (A1110501, ThermoFisher) for 5 min at 37 °C. Cells were then sterile filtered to single-cell suspensions and stained in 5%FBS/PBS on ice using antibody dilutions provided in Supplementary Data 1. DAPI was used as a live/dead cell marker. Flow cytometry analysis for endothelial differentiation markers shown in Supplementary Fig. 9f was performed on a Cytek Aurora spectral analyzer and was analyzed with SpectroFlo. All other flow cytometry analyses were collected using a BD Biosciences Symphony analyzer and analyzed with FlowJo.

## Cardiotoxin Muscle Injury

Gastrocnemius and SCM of 8-week-old mice was injured by injection with a 27 gauge needle with 100 μl and a 30 gauge needle with 50 μl of 10 μM cardiotoxin from *Naja mossambica mossambica* (C9759-5MG, Sigma), respectively. Muscle was dissected 28 days post injury then fixed and stained as described above. Antibody information listed in Supplementary Data 1.

## Immunoblotting

SDS-PAGE and subsequent immunoblotting were performed with standard procedures previously described in[30]. Briefly, protein lysates of cells and tissues were prepared using RIPA lysis buffer with 1X PPIC (#11697498001, Roche and #4906845001, Roche). Protein concentration was determined using a BCA (#23225, Pierce) and equally loaded lysates were resolved using SDS-PAGE. Blot were incubated overnight at 4 °C using primary antibodies listed in Supplementary Data 1. Membranes were then washed and proved with the species-specific secondary antibody conjugated to HRP. Protein bands were visualized using chemiluminescent luminol (sc-2048, Santa Cruz Biotechnology).

## RNA isolation and gene expression analysis

Total RNA was isolated from tissues or cells using miRNEasy mini kit (#217004, Qiagen) and from sorted cell populations using a miRNEasy

micro kit (#217084, Qiagen) according to the manufacturer's instructions. Superscript III First Strand Synthesis using random hexamer primers (#18080051, ThermoFisher) was used to synthesize first-strand cDNA. Real-time PCR was performed utilizing SYBR primers listed in Supplementary Table 1 and normalized to 18S ribosomal RNA control (#4308329, ThermoFisher Scientific). Relative quantity of genes of interest was determined using the ΔΔCT. RNA sequencing was performed on purified RNA using the Illumina NovaSeq6000 genome sequencing system. Paired end sequencing was used for 100 base pair reads.

## Bulk RNAseq data analysis

**(Sequence alignment, differential gene expression analysis, GSEA).** Mouse and human sequences were mapped to the mm10 and hg38 genomes, respectively, using the STAR aligner[78]. Gene level quantification was determined using RSEM[79] and based on GENCODE annotations M22 (for mouse samples) or Human Release 31 (for human samples), respectively. Non-coding and GENCODE level 3 genes were excluded. Differential gene expression was modeled using the voom method[80], available in the Limma R software package. Normalization factors were generated using the TMM method. Voom normalized counts were analyzed using the lmFit and eBayes functions in Limma. The false discovery rate (FDR) was estimated using the Benjamini-Hochberg method. Gene set enrichment analysis (GSEA) was performed using curated signatures from MSigDB. Ranking of genes was calculated using negative log10(*p*-value)*log2(fold change), and a *P*-value for each gene set was estimated by comparing the observed enrichment score to that obtained from a null distribution computed from 1000 permutations of genes within gene sets. FDR was estimated as previously described[81]. For gene ontology (GO) in *Tek-Cre;R26-tdTom* RNA-seq (Supplementary Fig. 2f), Enrichr was used to determine JENSEN tissue terms[82–84]. Functional annotation clustering by the Database for Annotation, Visualization, and Integrated Discovery (DAVID, 2021 version) was used for GO analyses. We used GOTERM_BP_FAT selection for functional annotation clustering and present the top ten unique GO terms visualized and represented as the -log₁₀Pvalue.

## Single-cell RNAseq

Mononuclear cells were dissociated from muscle or tumor and live tomato-positive cells collected by FACS in ice-cold 0.04% BSA (#A7159, Sigma-Aldrich) in PBS. Single cells were washed with 0.04% BSA in PBS and counted on a Countess II automated cell counter (ThermoFisher). 16,000 cells were loaded per lane on the 10x Chromium platform and processed using manufacturers protocol for cDNA synthesis and library preparation. Quality checks were performed using the Agilent 4150 TapeStation and Agilent Bioanalyzer before sequencing with NovaSeq6000 (Illumina).

## Single-cell RNAseq analysis

We used Cell Ranger v.4.0.1 pipeline (10x Genomics) with default parameters to aligned raw reads to the GRCm38 Ensembl 97 genome (mm 10) and to generate raw gene-barcode matrices. We use Seurat (version 4.0.6, R version 4.1.0) to perform merging, thresholding, normalization, principal component analysis, non-linear multi-dimensional reduction, clustering analysis, visualization and differential gene expression analysis[85]. Quality filtering was conducted by excluding low-quality or doublet outlier cells with values beyond 3x Median Absolute Deviation (MAD) from the median in the QC metrics of nFeature_RNA, nCount_RNA or mitochondria count per cell and removing genes expressed in less than 3 cells. To identify the integration of anchor genes among the samples, the FindIntegrationAnchors function was used with default parameters. Using Seurat's IntegrateData function, samples were combined into one object. Data were normalized using the SCTransform function and then principal

component analysis (PCA) was used to reduce the dimensionality of this dataset. The number of relevant dimensions was determined by ElbowPlot function, and these PCs were summarized further using Uniform Manifold Approximation and Projection (UMAP). For cell clustering, we used highly variable genes selected using FindVariableFeatures function with default parameters. In addition, nCount_RNA and percentage of mitochondrial genes were considered as the source of unwanted variability and were regressed out. Clustering was conducted using the FindNeighbors and FindClusters functions with original Louvain algorithm and a resolution parameter of 0.2. Differential gene expression analysis was performed by the FindAllMarkers function in Seurat. Clusters were annotated using markers genes found in the literature in combination with differentially expressed genes. We used PanglaoDB Augmented 2021 and CellMarker Augmented 2021 through Enrichr to assign cell cluster identity[82–84]. We use Seurat methods (DimPlot, FeaturePlot, VlnPlot, DoHeatmap etc.) or custom functions for data visualizations including UMAP, feature plot, violin plot and heatmap. Additional Note: Due to suboptimal QC metrics in some samples/analyses, performed additional manual cell filtering with arbitrary cutoffs.

Supplementary Fig. 1j: CAB-2829, retain cells with minimum 1000 detectable genes per cell and at least 2500 UMI counts per cell. Supplementary Fig. 1m: CAB-3347, retain cells with minimum 1000 detectable genes per cell per cell.

Gene lists used for conserved RMS cell states displayed in Fig. 4 and Supplementary Fig. 5 are mouse orthologs of the genes listed in Supplementary Table S2 of Wei et al.[43]. This list of murine genes is provided as Supplementary Data 22.

Cell dynamics were analyzed using scVelo (version 0.2.4)[41] package implementation in Scanpy (version 1.7.2)[86]. The spliced / unspliced count matrix was generated by Velocyto (version 0.17.17)[87] and was import to Scanpy_AnnData object. Then cell barcodes, clusters and UMAP embeddings were imported from Seurat object. Then the data was then processed using default parameters following Scanpy scVelo implementation. In short, the samples were pre-processed using scVelo functions for filtering and normalization using scv.pp.filter_and_normalise and followed by scv.pp.moments function. We used scv.tl.recover_dynamics function to recover the full splicing kinetics of specified genes. The gene specific velocities were then calculated using scv.tl.velocity with dynamic mode, and scv.tl.velocity_graph functions and visualized using scv.pl.velocity_graph function. For Partition-based graph abstraction (PAGA) analysis[88], we used Scanpy scVelo implementation function scv.tl.paga with default parameters and scv.pl.paga function for velocity-driven PAGA analysis and plotting PAGA graph with velocity-directed edges.

## Nuclei isolation and cleavage under targets and release using nuclease (CUT&RUN)

Live tumor cells were isolated from bulk TCP and MCP tumors by enzymatic digestion and FACS as described above. Nuclei were isolated by incubating cells in Nuclear Extraction Buffer (NEB) which consists of 20 mM HEPES KOH pH7.9, 10 mM KCl, 0.1% Triton X-100, 20% Glycerol, EDTA-free Protease Inhibitor (11836170001, Roche) and water for 10 min on ice. Aliquots of 250,000 nuclei were spun down then resuspended in nuclear extraction buffer and slowly frozen at −80 °C in a Mr. Frosty freezing container (C1562, Sigma-Aldrich).

CUT&RUN was performed using the CUTANA ChIC/CUT&RUN kit (14-1048, EpiCypher) and each reaction used 250,000 nuclei. All buffers except for the bead activation buffer, which was from the CUTANA ChIC/CUT&RUN kit, were made by our lab. In brief, concanavalin A beads (ConA, 14–1048, EpiCypher) were activated with bead activation buffer (14–1048, EpiCypher). Frozen nuclei were quickly thawed at 37 °C and bound to activated ConA beads then transferred to antibody buffer, which is comprised of 20 mM HEPES pH 7.5, 150 mM NaCl, 0.5 mM Spermidine, EDTA-free Protease Inhibitor (11836170001,

Roche) 0.0008% Digitonin (#16359, Cell Signaling), and 2 mM EDTA in water. Antibodies were added and incubated on nutator overnight at 4 °C. Next, nuclei were washed with cold digitonin buffer, 20 mM HEPES pH 7.5, 150 mM NaCl, 0.5 mM Spermidine, EDTA-free Protease Inhibitor (11836170001, Roche) 0.0008% Digitonin (#16359, Cell Signaling), and CUTANA pAG-MNase was added then washed with digitonin buffer. 100 mM CaCl₂ was added to activate MNase to cleave target chromatin. After two hours, stop buffer comprised of 340 mM NaCl, 20 mM EDTA, 4 mM EGTA, 50ug/ml RNAse A, and 50 ug/ml Glycogen in water, was added. CUT&RUN-enriched DNA was then purified using CUTANA ChIC/CUT&RUN kit (14–1048, EpiCypher). Libraries were prepared with the xGen ssDNA & Low-Input DNA Library Prep Kit (#10009859, IDT) and quantified using the Agilent 4150 TapeStation and Agilent Bioanalyzer before sequencing with NovaSeq6000 (Illumina).

## CUT&RUN analysis

Paired-end CUT&RUN-seq reads were aligned first to the *E. coli* reference genome using bowtie[89](version 1.2.2) in paired-end with parameters -k 1 –best and –un to retain unmapped reads. Because of nonbiological variability among samples, the numbers of *E. coli* reads were not used for cell-number-normalization hereafter. Retained non-*E. coli* reads were then mapped to the mouse reference genome (mm10) or human reference genome (hg38) using bowtie in paired-end mode with parameters -p 20 -k 2 -m 2 –best. A BAM file containing CUT&RUN-seq fragments was computationally created from each aligned read-pair using samtools sort -n, bedtools bamToBed -bedpe, a manual conversion from BEDPE to BED3, and bedtools bedToBam. For display, the reference genomes were binned into 50 bp windows using bedtools makewindows, and coverage of those bins by fragments was calculated using bedtools intersect -c. Fragment coverage was normalized per million mapped fragments, converted to bedGraph using bedGraphToBigWig, converted to wiggle using bigWigToWig, converted to TDF using igvtools toTDF, and then visualized in the IGV[90](version 2.16.2.) browser in hg38 for human samples.

MACS1.4[91] was used to identify regions significantly enriched in CUT&RUN-seq reads. Prior to peak-calling, individual reads from each paired-end alignment were discarded if they overlapped the ENCODE-defined Problematic Regions list[92] corresponding to the reference genome in use, and further filtered in mouse if they overlapped the region chr1:24611436-24616256, because of an observed genomically amplified region in all samples near the *Col19a1* gene. Retained reads were used for peak-calling against each sample's corresponding, identically processed, control IgG. In mouse, parameters –keep-dup = auto and –p 0.001 were used; in human, parameters –keep-dup=auto and -p 1e-9 were used. To compare TCP vs. MCP coverage at regions of interest, we collapsed peaks called separately in each sample ($N = 3$ TCP and $N = 4$ MCP in mouse) using bedtools merge. Bedtools closest -t first was used to annotate each peak with the name of the gene whose transcript has the nearest transcription start site from mm10_refGene.gtf downloaded from the UCSC Genome Browser. Individual read coverage in each of the seven samples was separately quantified using bedtools intersect -c. For comparative analyses (Fig. 5c), read coverage in each sample was normalized to the millions of mapped reads (RPM), and the average RPMs from each condition were calculated. For statistical analysis (Fig. 5d) of differential coverage, raw reads from the three TCP samples and three representative MCP samples (MCP20_1, MCP20_2, and MCP_38) were used as input for DEseq2. Peaks were considered differentially covered if they had DEseq2 adjusted log2 fold-change >1 or <−1 and adjusted *p* value of 0.05.

We used ROSE[51] (https://bitbucket.org/young_computation/rose) to separately identify super-enhancers from each sample of MCP, TCP, Day 15 Untransduced p53^KO iPSC, and Day 15 Lenti-*P3F* transduced p53^KO iPSC CUT&RUN-seq data targeting H3K27ac, as previously

described[93]. Each sample was separately processed with its corresponding IgG. For each sample, we used MACS1.4 to identify two sets of peaks from each read set filtered as above. In mouse, we used parameter sets –keep-dup=auto –p 0.001 and –keep-dup=all –p 0.001; in human, we used parameter sets –keep-dup=auto –p 1e-9 and –keep-dup = all -p 1e-9. These two peak sets from each sample were collapsed into a per-sample set of peaks using bedtools merge; these collapsed peaks were used as input for ROSE with parameters -s 12500 -t 1000.

Regions displayed in Fig. 5e and f were created from the collapsed union of super-enhancers for MCP ($n = 4$) and TCP ($n = 3$). Coverage of these collapsed regions in each condition was separately quantified using bedtools intersect -c. For comparative analyses, we followed the same strategy described in peak-based comparative analyses to normalize read coverage in each sample to the millions of mapped reads (RPM), and the average RPMs from each condition were calculated. The DESeq2 package[48] and its standard parameters was used to identify statistically differential coverage of super-enhancers between TCP and MCP replicates, by using three TCP samples and three representative MCP samples (MCP20_1, MCP20_2, and MCP_38). Significantly differentially covered super-enhancers met two thresholds: adjusted $P$-value = 0.05 and absolute $\log_2$Fold change (MCP/TCP) > 1. After collapsing, super-enhancers were assigned to the single gene with a transcript whose start site from the mm10_refGene.gtf gene list is nearest the center of the collapsed super-enhancer using bedtools closest.

To determine if enhancers or super-enhancers were specifically activated to drive specific gene sets of known function, we performed Gene Set Enrichment Analysis (GSEA). DESeq2 $\log_2$FC(MCP/TCP) tables associated with collapsed enhancers (Fig. 5d) or collapsed super-enhancers (Fig. 5f) were used as raw table for preparing inputs for GSEA Preranked (version 4.3.2)[94]. Each enhancer/super-enhancer is associated with a gene, so a single gene could have multiple ranked loci associated with it on the raw DESeq2 $\log_2$FC(MCP/TCP) tables mentioned above. The single representative enhancer/super-enhancer for each gene was selected as the one with the highest absolute DESeq2 $\log_2$FC(MCP/TCP). These ranked genes were analyzed for their enrichment using the Gene Ontology hallmarks collection in MSigDB[95]. We used parameters minSize=15, maxSize = 500, nperm=1000 for collapsed peaks GSEA Preranked analysis, and minSize=1, maxSize = 2500, nperm=1000 for pre-ranked $\log_2$FC(MCP/TCP) from collapsed Super-Enhancers GSEA Preranked analysis.

## Motif enrichment analysis

We performed a motif enrichment analysis in putative enhancers using AME (Analysis of Motif Enrichment)[96]. A list of transcription factors associated with the endothelial lineage was built from literature analysis and is available as (Supplementary Data 29). We selected the *Mus musculus*-sourced endothelial TF motifs PWMs (Position-Weight Matrix) from the CIS-BP 2.00 database[97] corresponding to these TFs. Coordinates of putative enhancers were defined as above using CUT&RUN-Seq from MCP or TCP cells with only the parameter set -p 0.001 –keep-dup=auto. For each transcript from mm10_refGene.gtf, its putative promoter region was defined as +/− 2 kb from the TSS (Transcription Start Site); we used bedtools subtract to remove sections of H3K27ac peaks defined as above that overlap these promoters because H3K27ac marks both active promoters and enhancers. The sequences of these remaining regions were acquired using bedtools getfasta and the mm10 reference genome. AME was then used to identify the statistically enriched TF motif hits within the sequences of these putative enhancer regions, with the command 'ame --control --shuffle-- --method fisher'.

## HiC

Live tumor cells were isolated from bulk TCP and MCP tumors by enzymatic digestion and FACS as described above. HiC and library preparation was performed on 500,000 cells per replicate using

reagents and protocol from the Arima-HiC Kit (#A510008, Arima). Libraries were quantified and quality check was performed using the Agilent 4150 TapeStation and Agilent Bioanalyzer before sequencing with NovaSeq6000 (Illumina).

## HiC Analysis

For display of HiC interaction frequencies, paired-end reads of 150 bp were processed by Juicer (v1.5, default parameters)[98] based on mm10 and Arima fragmentation (cut sites GATC or GANTC). Each replicate was processed separately using default parameters (every replicate got >500 M pairs as >360 M contacts). We required MAPQ score greater than 30 for generating.hic file and downstream analysis. Reproducibility have been confirmed by visual inspection.

For genome-wide comparisons of contact frequencies, we applied HiC-Pro (v 2.11.1)[99] to perform read alignment, read filtering, quality check, and contact matrix building as described with default parameters unless otherwise specified below. The two replicate paired-end FASTQ files from each condition were separately pooled, and to achieve improved performance, we split these pooled FASTQ files into multiple subsets, each with a maximum of 10 million read pairs, which were used as input for HiC-Pro in parallel mode. HiC-Pro was configured so that each end of the read pair was mapped independently to the mm10 reference genome using bowtie2 (v2.3.5.1)[100]. For global and local alignment, we used the options "bowtie2 --very-sensitive -L 30 --score-min L,−0.6,−0.2 --end-to-end −reorder" and "bowtie2 --very-sensitive -L 20 --score-min L,−0.6,−0.2 --end-to-end −reorder" respectively. For read mapping, we used MAPQ threshold=10 and removed multi-mapped reads (RM_MULTI = 1) and duplicated read pairs (RM_DUP = 1). Taking the final valid read pairs generated from HiC-Pro as input, we used the "hicpro2juicebox.sh" utility script to create a HiChIP contact map in ".hic" format. The bin intervals files fastq.matrix from HiC-Pro outputs were combined as RelevantBins matrices between TCP and MCP at 5Kb, 10Kb, 100Kb, and 1 Mb bin sizes. The TCP and MCP connectivity RelevantBins combined matrices were used for generating the scatterplot of TCP and MCP in Fig. 5a.

Scatterplots comparing MCP vs. TCP connectivity were assembled from aligned HiC data utilizing the ggplot2 package in R[101]. All bin-pairs with at least one read connecting them were used as an input for a matrix that describes connectivity across experiments. Reads connecting pairs of bins (bin-pairs) were plotted. A subset of points was highlighted if either of the bins comprising a bin-pair overlapped a preselected gene of interest by at least 1 bp. *Hoxa9* and *Hoxa11* are within both the same 1MB bin and the same 10 kb bin, so they are assigned the same color.

## Endothelial Differentiation Assay

We generated high-purity endothelial cells from patterned mesoderm from p53^{KO} BJFF.6 human induced pluripotent stem cells following the protocol described by the Murry laboratory[102]. Here, iPSCs were directed to differentiate to hemogenic mesoderm with the addition of RPMI based media supplemented with 50 ng/mL activin A (#338-AC, R&D Systems) and 1xB-27 without insulin (#A18956-01, Life Technologies). The next day, cells were transferred to RPMI media with 40 ng/mL BMP4 (#314-BP, R&D Systems), 1 μM CHIR-99021 (#13122, Cayman Chemical), and 1xB-27 without insulin (#A18956-01, Life Technologies). To differentiate the cells to the endothelial lineage, media was transferred to StemPro-34 (#10639-011, Life Technologies) supplemented with 4x10⁻⁴ M 1-Thioglycerol (MTG, #M6145-25 mL, Sigma-Aldrich), 2mM L-glutamine (#25030-081, Life Technologies), 50ug/mL L-Ascorbic acid 2-phosphate sesquimagnesium salt hydrate (#A8960-5G, Sigma-Aldrich), 10 ng/mL BMP4 (#314-BP, R&D Systems), 5 ng/mL bFGF (AF-100-18B, PeproTech), and 300 ng/mL VEGF (#100-20, PeproTech) and left for 72 h. The end of this incubation is designated as day 5, and cells were transduced with a lentiviral vector with P3F or empty in EGM BulletKit (#CC-3124, Lonza) supplemented with 20 ng/

mL VEGF (#100-20, PeproTech), 20 ng/mL bFGF (AF-100-18B, Pepro-Tech), and 1 μM CHIR-99021 (#13122, Cayman Chemical) containing 8ug/mL polybrene (#TR-1003, Sigma-Aldrich) and 10 μM Y27632 (#1254, Tocris Bioscience). Lentiviral media was changed on trans-duced cells 18 h after. Non-transduced cells are referred to as control (CNTL). Cells were maintained on 0.1% porcine gelatin (#G1890-500G, Sigma) coated plates in supplemented EGM from day 5 to day 14. A modification to the published protocol, we used accutase (#A1110501, ThermoFisher Scientific) instead of 2.5% (vol/vol) trypsin, to improve cell viability after plating.

### Endothelial tube formation assay

12-well cell culture plates (#3513, Corning) were coated with 250 μl matrigel basement membrane matrix (#354234, Corning) and har-dened in incubator at 37 °C for 30 min. Endothelial differentiated p53$^{KO}$ iPSCs were dissociated with accutase (#A1110501, ThermoFisher Sci-entific) and resuspended in supplemented EGM 100,000 cells were plated per well and monitored for tube formation from 4 h to 8 h after plating.

### Viability assay

Viability of endothelial differentiated CNTL, +empty and +P3F iPSCs was assessed with CellTiterGlo assays (#G7570, Promega) performed according to the manufacturer's protocol. Luminescence was mea-sured on a BioTek Synergy 2 with BioTek's Gen51.11 software.

### Focus Formation

Transformed P3F expressing p53$^{KO}$ iPSCs were plated at a density of 1 x 10$^5$ cells per well of a gelatin coated six-well dish 17 h after transduction with shP3F or shLuc expressing lentivirus. Cells were grown for 10 days in supplemented EGM and then fixed in methanol prior to staining with 0.02% crystal violet staining solution (0.2 g of crystal violet dissolved in 100 mL of a 2% ethanol solution) for 10 min at room temperature (#C581-25, Fisher Scientific). Wells were washed in deionized water 3 times and allowed to air dry overnight before imaging. Foci were counted by hand.

### Teratoma Assay

p53$^{WT}$ and p53$^{KO}$ iPSCs were detached from plates and resuspended in mTeSR Plus (Stemcell Technologies, 100-0276) then counted and spun down. Cells were resuspended in matrigel basement membrane matrix (#354234, Corning) at a concentration of 4x10$^6$ cells per 100 μl and kept on ice. SCID beige mice were injected with a 25-gauge needle with 100 μl iPSCs in Matrigel into their gastrocnemius.

### Statistical Analyses

Data analyses were performed using Microsoft Excel, GraphPad Prism (version 9.3.1), and the R software packages. Sample size and replicates are listed in the text or Figure legends. All error bars are reported as mean ± SEM. Unless otherwise mentioned, statistical significance was determined using Students t-test for pairwise comparisons (*$P < 0.05$, **$P < 0.01$, ***$P < 0.001$, ****$P < 0.0001$). Flow cytometry analyses were performed on a BD FACSAria sorter and FlowJo 10.8.1.

### Reporting summary

Further information on research design is available in the Nature Portfolio Reporting Summary linked to this article.

## Data availability

The transcriptomics and genomics data have been deposited to the Gene Expression Omnibus (GEO) as series GSE218274 at the following: St. Jude ProteinPaint RNA sequencing data is publicly available and can accessed through their open resource page (https://pecan.stjude.cloud). Source data are provided with this paper. All remaining data is available in the Article, Supplementary and Source Data files. Any materials generated by this study will be made available by request directed to M. E. Hatley. Source data are provided with this paper.

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

## Acknowledgements

We thank St. Jude Children's Research Hospital Shared Resources including Flow Cytometry and Cell Sorting (Richard Ashmun), Comparative Pathology Core (Peter Vogel, Pam Johnson), the Hartwell Center and Functional Genomics Core (Geoff Neale, Melanie Loyd), the Center for Applied Bioinformatics (Gang Wu), the Animal Resources Center (Harshan Pisharath), the Center for Advanced Genome Editing (Shondra Pruett-Miller), the Center for In Vivo Imaging and Therapeutics (Walter Akers). We are grateful for expertise provided by the St. Jude Collaborative Research Consortium on the 3D Regulatory Nuclear Landscape of Pediatric Cancer Cells. We thank Lawryn Kasper and Dr. Suzanne Baker's laboratory for their technical support. K.E.G. is supported by a Damon Runyon-Sohn Pediatric Cancer Fellowship (DRSG-30-19). Research reported in this publication was supported by the National Cancer Institute of the National Institutes of Health under Award Numbers R01CA216344 and R01CA251436 (MEH) and F31CA250398 (MB). The content is solely the responsibility of the authors and does not necessarily represent the official views of the National Institutes of Health. The Hatley laboratory is also supported by grants from the V Foundation for Cancer Research, the Rally Foundation for Childhood Cancer Research and Open Hands Overflowing Hearts award number 20IC23 (MEH), St. Jude Cancer Center Support Grant (P30 CA21765), American Lebanese Syrian Associated Charities of St. Jude Children's Research Hospital, and the St. Jude Graduate School of Biomedical Sciences.

## Author contributions

M.S. Conceptualization, formal analysis, funding acquisition, validation, investigation, methodology, writing-original draft, writing-review and editing. R.L. Formal analysis, investigation, methodology, writing-review and editing. B.S. Investigation, data curation, methodology, writing-review and editing. Y.Z. Data curation, formal analysis, investigation, methodology, writing-review and editing. H.J. Data curation, formal analysis, investigation, methodology, writing-review and editing. C.D. Conceptualization, investigation, methodology, writing-review and editing. C.L. Investigation, writing-review and editing. K.G. Investigation, funding acquisition, writing-review and editing. K.V. Investigation, writing-review and editing. K.R. Investigation, writing-review and editing. M.G. Investigation, writing-review and editing. B.X. Data curation, formal analysis, investigation, methodology, writing-review and editing. D.K. Investigation, writing-review and editing. G.A. Investigation, writing-review and editing. N.D. Data curation, investigation, formal analysis, methodology, writing-review and editing. S.P. Investigation, methodology, writing-review and editing. P.S. Data curation, investigation, formal analysis, writing-review and editing. S.P.-M. Methodology, writing-review and editing. B.A. Formal analysis, investigation, methodology, writing-review and editing. J.R. Formal analysis, writing-review and editing, M.H. Conceptualization, resources, formal analysis, supervision, funding acquisition, writing-original draft, writing-review and editing.

## Funding

National Cancer Institute of the National Institutes of Health under Award Numbers R01CA216344 and R01CA251436 (MEH) and F31CA250398 (MB). Damon Runyon-Sohn Pediatric Cancer Fellowship DRSG-30-19 (KEG).

## Competing interests

MEH has served on advisory board for Servier. The remaining authors declare no competing interests.
