## [Peer Review File · Nature Communications]

Reviewers' Comments:

Reviewer #1:

Remarks to the Author:

The authors provide compelling evidence that endothelial cells may give rise to FP-RMS, and that non-myogenic cells of origin may produce gene expression profiles quite similar to myogenic derived FP-RMS models. This is impactful work.

The authors hypothesize that endothelial cells may be cell of origin for PAX-fusion positive RMS. The hypothesis is motivated from the novel combination of regulatory elements that is derived from t(2;13) translocations in RMS tissues, which enable expression of the PAX3-FOXO1 fusion in cells that would not normally express wild-type PAX3.

The authors generated aP2-Cre;Cdkn2aFlox/Flox;Pax3P3Fm/P3Fm;R26-tdTom to conditionally delete Cdkn2a in combination with induced expression of PAX3-FOXO1. The CRE-expressing cells induce fusion oncogene expression in endothelial progenitors, and are not associated with skeletal muscle development. Interestingly, a substantial percentage of PAX3-FOXO1 expressing cells also expressed PAX7. Further results suggest that the PAX3-FOXO1 inducible expression reprograms early endothelial cells into cells that have a capacity to contribute to myogenesis.

The authors perform scRNAseq in P2-Cre;R26-tdTom and Fabp4-Cre;R26-tdTom systems to compare populations of cell types associated with the conditional CRE expression. However, the endothelial progenitor populations are fairly consistent across the different systems. Next the authors compare a TEK-driven CRE with the aP2-driven CRE to develop endothelial specific induction of PAX3-FOXO1. Tek-Cre;Pax3P3Fm/P3Fm;Cdkn2aFlox/Flox;R26-tdTom were shown to be distinct from muscle cell progenitors, and thus in the TEK-CRE system PAX3-FOXO1 does not reprogram the cells into myogenic progenitor cells. The authors then compare a Myf6-CRE driven PAX3-FOXO1 RMS model with the TEK-CRE PAX3-FOXO1 RMS model, and find that CDKN2A loss alone is insufficient to produce alveolar RMS tumors, while in both Myf6- and TEK-driven systems alveolar RMS tumors are observed. Each system (Myf6- and TEK-driven) produces RMS with MYOD1 induction as well as SOX8 and other core regulatory circuitry genes. Moreover, each class of FP-RMS system (Myf6- and TEK- driven) are associated with transcriptional heterogeneity, as has been seen in recent impactful studies from the Dyer and Langenau groups (Dev Cell 2022, Nat Cancer 2022).

Interestingly, at the population level, based on transcriptional patterns, there are many similarities between endothelial derived and myogenic derived FP-RMS tumors. Epigenetically there are also many similarities in FP-RMS models, regardless of myogenic or endothelial cell origins, based on H3K27ac profiles in the cells.

The authors also investigated the ability of PAX3-FOXO1, in combination with MYCN to transform human endothelial cells. With lentiviral infections, PAX3-FOXO1, human endothelial cells begin to express MYOD1 and Myogenin. Moreover, with co-transduction with TERT and MYCN, PAX3-FOXO1 is able to produce tumors from human endothelial cells that histologically and transcriptionally resemble alveolar RMS. Interestingly, human iPSCs expressing PAX3-FOXO1 during endothelial differentiation conditions were found to express myogenic cell markers and form RMS tumors with alveolar histology when injected into immunocompromised mice

Major comments:

It would strengthen the work to meta-analyze the scRNAseq clusters with the recently published scRNAseq from Wei Nat Cancer 2022 and Patel Dev Cell 2022 to identify further similarities in the gene expression patterns and heterogeneity with human derived FP-RMS.

Are there any endothelial signatures present in human FP-RMS cell lines, PDXs, or primary patient samples? Would it be possible to compare RNA-seq signatures in the MCP, TCP, and human model systems in the manuscript to identify any endothelial regulatory TFs that are expressed? Master regulatory GATA-family TFs, and FOXF1 are each implicated in endothelial cell circuitry.

The epigenetics analysis would be strengthened from identifying high confidence DNA binding motifs within the H3K27ac data the authors already have. Are cell type specific TFs implicated in the most highly enriched motifs?

It would strengthen the study to determine if PAX3-FOXO1 is essential for tumor maintenance. If there is PAX3-FOXO1 depletion in the TCP mice after transformation, how does this affect proliferation or gene expression? Similarly, if the HUVECs or iPSCs are (1) transduced PAX3-FOXO1 earlier or later than TERT/MYCN does this alter oncogenic transformation or gene expression, and (2) if PAX3-FOXO1 is depleted after tumor formation in the human iPSCs or HUVECs how does this alter proliferation and gene expression?

Minor comments:

The font is quite small especially in Figs. 3,4,5. It would be helpful if the authors could increase font size for readers, and make sure labels are clear and consistent.

What are the therapeutic implications of the findings? The cell of origin in RMS has been widely debated, and work is significant. However it is unclear how identification of endothelial cells as potential cells of origin could affect clinical decision making or potential pre-clinical testing of agents. It would strengthen the manuscript if the authors could comment on this.

Reviewer #2:

Remarks to the Author:

In this manuscript, the authors provide evidence that expression of the PAX3-FOXO1 (P3F) can reprogram mouse and human endothelial progenitors to muscle stem cells that can generate fusion-positive rhabdomyosarcoma (FP-RMS). They used a variety of mouse models of endothelial-specific expression of P3F to show the ability to reprogram endothelial progenitors into myogenic stem cells that retain the capability to support regeneration of injured muscle fibers. They also show that within the context of a mouse model of Cdkn2a loss in endothelial cells P3F expression promoted FP-RMS and that P3F expression in TP53-null human iPSCs prevents their differentiation into the endothelial lineage while promoting conversion into myogenic cells that can form FP RMS tumors in immuno-compromised mice.

The data shown in this manuscript further support an emerging concept that RMS can originate from aberrant development of non-myogenic cells – in this case FP-RMS driven by P3F. At the same time, these findings suggest that P3F can reprogram somatic and stem cell types into myogenic cells that, when deficient for key regulators cell cycle checkpoints (Cdkn2 and TP53), can undergo transformation into FP-RMS.

The manuscript is of interest, but some issues need to be addressed, as outlined below

- 1) Overall, while this work fairly supports the hypothesis that FP-RMS can derive from endothelial cells upon ectopic expression of P3F in experimental mouse models, this evidence does not mean FP-RMS invariably originate from endothelial cells. This limitation should be acknowledged by the authors
- 2) While it is interesting that P3F reprograms endothelial cells into muscles, the manuscript does not provide insights into the mechanism by which P3F promotes this reprogramming. Is this a direct or indirect action of P3F? Or it is mediated by the well-known reprogramming factor MyoD? This should be addressed by performing genome-wide chromatin binding (CUT and RUN or ChIPseq) of P3F and MyoD, with data integration with H3K27Ac genome-wide signal and RNAseq. Moreover, experiments within the context of MyoD null mice could also address this issue
- 3) Why TP53 deficiency is required to promote selective conversion of hiPSCs into myogenic cells (with repression of endothelial cell lineage) that can form FP RMS tumors in immuno-compromised mice?
- 4) There are concerns with significance of HiC datasets and their analysis. What is the resolution of the analysis? It looks like the resolution is not high enough to detect other structures than TADs. Figure 4b and c compare TADs at specific loci in MPC and TCP, but no control cell of origin is shown

(necessary)

Reviewer #3:

Remarks to the Author:

In the manuscript entitled PAX3-FOXO1 dictates myogenic reprogramming and rhabdomyosarcoma identity in endothelial progenitors, Searcy et al show that endothelial cells can be a cell of origin for fusion positive rhabdomyosarcoma (FP-RMS). The same group has previously shown that fusion negative RMS (FN-RMS) can have a non-myogenic origin. Further, based on the finding by Vicente-Garcia et al, showing that cis-regulatory elements driving endothelial expression of FOXO1 in humans and mice regulate PAX3-FOXO1 (P3F) expression, the authors hypothesise that endothelial specific expression of P3F could cause transdifferentiation of vasculature giving rise to FP-RMS. To test this hypothesis the authors, induce P3F expression and Cdkn2a deletion using various Cre mouse lines: aP2-Cre (ACP), Tek-Cre (TCP) and the previously described Myf6-Cre (MCP). While in ACP mice P3F reprograms endothelial progenitors to functional myogenic stem cells capable of regenerating injured muscle fibers, in TCP mice P3F accurately recapitulates human FP-RMS through histologic profiling and gene expression. Importantly, while TCP and MCP tumors originate from different cell types, they share nearly indistinguishable histology, gene expression, and chromatin architecture. In TP53-null human iPSCs P3F blocks endothelial-directed differentiation and guides cells to become myogenic cells that form FP-RMS tumors in immunocompromised mice. Overall, this work demonstrate that P3F has the potential to reprogram mouse and human endothelial progenitors to FP-RMS an could explain why in humans FP-RMS can occur in several sites in the body devoid of skeletal muscle.

This is a very interesting study that highlights how transdifferentiation plays an important role in cancer and in FP-RMS in particular. The study will be of interested for those studying cells-of-origin in cancer (which in sarcoma remain elusive for most sarcoma subtypes), and sarcomagenesis in general, but it also has implications in the field of muscle regeneration and cell plasticity/transdiferentiation. The following points should be clarified:

1. FP-RMS tumors from TCP and MCP mice accurately recapitulate human disease through histologic profiling, gene expression and enhancer state. Together with the data shown in human HUVECs and iPSCs the authors clearly show that P3F is capable of reprogramming endothelial cells to FP-RMS. However, the results from all the Cre lines (aP2-Cre and Fapbb vs Tek-cre and Myf6-Cre) are not very easy to follow. One of the main issues in terms of lack of clarity given the several models described is exactly which endothelial cell population can serve as cell of origin for FP-RMS. In ACP mice P3F reprograms endothelial cells to functional muscle stem cells but does not give rise to FP-RMS. In Fapbb4-Cre which exhibit higher Cre-recombination efficiency 3 out of 20 mice develop FP-RMS. In Tek-Cre mice P3F is not able to reprogram endothelial cells to functional muscle stem cells but gives rise to FP-RMS with much higher efficiency. So, the question becomes: is there a common endothelial cell population in aP2-Cre (or Fapbb4-Cre) and Tek-Cre mice that can give rise to FP-RMS upon P3F expression? Why does P3F reprogram endothelial cells to muscle in aP2-Cre mice but not in Tek-Cre mice if they recombine the P3F allele in similar endothelial cell populations? Indeed, the authors use scRNAseq to show that both aP2-Cre and Tek-Cre mediate recombination in the same cell types in the SCM including endothelial and hematopoietic cells and the authors state that Tek-Cre is expressed by endothelial cells labelled by aP2-Cre. This is confusing and not clearly discussed in the manuscript. The authors should consider the option of comparing the different Cre lines within a figure instead of scattering them through the main figures and supplementary material.

2. It should be clear in the text why the authors compare P3F activation in hindlimb and neck sternocleidomastoid muscle (SCM) of ACP mice and why those are expected to show different results. On one hand muscles in the head and neck have distinct embryonic origins from somite-derived trunk and limb muscles. On the other hand, previous studies from the same lab shown that aP2-Cre-labeled interstitial endothelial cells adjacent to the myofibers in the SCM are the ones giving rise to FN-RMS in response to SmoM2 expression (Drummond et al 2018). Such cells are not present in hindlimb muscle and FN-RMS develops exclusively in head and neck (correct?). This should be briefly mentioned as it is not totally clear why in Fig.1 SCM and hindlimb muscle are

compared or if the disparate results are expected.

3. Line 160: It is unclear why if only 10% of isolated muscle stem cells from SCM of ACP mice are tdTomato positive (compared to about 80% in PAX7CreERT2) (Fig 1d-e) how almost 100% of the myofibers are tdTomato+ following injury (Fig1f). Does this mean that existent muscle stem cells (which should be negative for tdTomato) have very little to no contribution to muscle regeneration in contrast to tdTomato + P3F +cells?

4. Line 146: "Surprisingly, of 940 Tom+ cells counted in the SCM of ACP mice, 30% co-localized with PAX7. None of the 66 PAX7+ muscle stem cells counted beneath the basal lamina were Tom+. Tom+ cells in the hindlimb of ACP mice did not co-localize with PAX7 (Fig. 1b)." It is unclear if this co-localization is driven by P3F expression. The same staining in aP2-Cre, tdTomato mice in the absence of P3F should be shown.

5. The authors mention in the Introduction (line 110) that P3F has been shown to block endothelial differentiation by decreasing FOXC2 expression (ref 24,25). However, this is never again mentioned in the manuscript even though the authors use the appropriate models to test it. Is FOXC2 expression down regulated in differentiating iPSCs expressing P3F for example?

6. Although the authors most times include the number of replicates in the figure legends, some experiments only show representative images and no statistics (eg standard deviation)(Fig1b, c for example). In the text results are expressed as an overall % but are lacking statistics. Example, Line 155: "We found that dissociated ACP cells from the SCM but not from the hindlimb formed myotubes evident by the co-localization of Tomato and myosin heavy chain (MyHC) staining in 51% of Tom+ cells (Fig. 1c)". the authors should represent these quantifications in graphs so that error bars corresponding to replicates are shown.

7. In the discussion the authors mention that: "In the mouse model used for this study, P3F expression is driven by the Pax3 promoter and influenced by the regulatory elements of Foxo1 knocked into the Pax3 locus. However, this model lacks downstream regulatory elements over 100 kb from the TSS required to drive FOXO1 expression in vascular endothelial cells. Forcing the expression of P3F using this genetically engineered mouse model in endothelial cells circumvents the lack of distal Foxo1 cis-regulatory elements in the Pax3:Foxo1 allele to drive P3F expression robustly and completely in this cell type"

Maybe I'm missing something but if P3F expression is driven by the Pax3 promoter and the model lacks downstream regulatory elements over 100 kb from the TSS required to drive FOXO1 expression in vascular endothelial cells, how is the expression of P3F "forced" in these cells? Pax3 should not be expressed in endothelial cells if I understood properly?

Minor points

- Figure 1A should show also a control muscle condition that was not treated with cardiotoxin, and if possible, a marker of muscle regeneration to show that the cardiotoxin treatment is effective.

- Line 186: "Preformed" instead of performed.

- Line 200: The authors optimize the "liberal gating" to exclude the probability of contamination by Tomato- cells. However, td-tomato signal is quite bright and two independent populations should in principle be identified without the need to gate for the top 50% positive cells. The original gating should be shown in supplementary figure so that it is clear to what extent tdTomato positive and negative cells can be distinguished. Or do the author think cell duplets could explain the intermedium contaminating signals?

- Line 219: To test this (...) mice did not co-localize" this phrase is not well constructed.

Response to Reviewer's Comments

We thank the editor and the reviewers for the insightful comments on our manuscript and the opportunity to improve it by responding to their suggestions, questions, and critiques. We hope that you agree that these changes resulted in a more clear and rigorous manuscript. Included are references to changes in the manuscript text or changes to the Figures (main or supplemental). The manuscript changes are in red in the final manuscript document for re-submission. We have also elevated Randolph K. Larsen IV to co-first author, added Kristin B. Reed and Grace E. Adkins as authors to the manuscript, and included additional acknowledgements based on additional data and effort added in this revised manuscript. A point-by-point response to the reviewer inquiries are discussed below.

Reviewer #1 – RMS, epigenomics, scRNA-seq (Remarks to the Author):

The authors provide compelling evidence that endothelial cells may give rise to FP-RMS, and that non-myogenic cells of origin may produce gene expression profiles quite similar to myogenic derived FP-RMS models. This is impactful work.

The authors hypothesize that endothelial cells may be cell of origin for PAX-fusion positive RMS. The hypothesis is motivated from the novel combination of regulatory elements that is derived from t(2;13) translocations in RMS tissues, which enable expression of the PAX3-FOXO1 fusion in cells that would not normally express wild-type PAX3.

The authors generated aP2-Cre;Cdkn2aFlox/Flox;Pax3P3Fm/P3Fm;R26-tdTom to conditionally delete Cdkn2a in combination with induced expression of PAX3-FOXO1. The CRE-expressing cells induce fusion oncogene expression in endothelial progenitors, and are not associated with skeletal muscle development. Interestingly, a substantial percentage of PAX3-FOXO1 expressing cells also expressed PAX7. Further results suggest that the PAX3-FOXO1 inducible expression reprograms early endothelial cells into cells that have a capacity to contribute to myogenesis.

The authors perform scRNAseq in P2-Cre;R26-tdTom and Fabp4-Cre;R26-tdTom systems to compare populations of cell types associated with the conditional CRE expression. However, the endothelial progenitor populations are fairly consistent across the different systems. Next the authors compare a TEK-driven CRE with the aP2-driven CRE to develop endothelial specific induction of PAX3-FOXO1. Tek-Cre;Pax3P3Fm/P3Fm;Cdkn2aFlox/Flox;R26-tdTom were shown to be distinct from muscle cell progenitors, and thus in the TEK-CRE system PAX3-FOXO1 does not reprogram the cells into myogenic progenitor cells. The authors then compare a Myf6-CRE driven PAX3-FOXO1 RMS model with the TEK-CRE PAX3-FOXO1 RMS model, and find that CDKN2A loss alone is insufficient to produce alveolar RMS tumors, while in both Myf6- and TEK-driven systems alveolar RMS tumors are observed. Each system (Myf6- and TEK-driven) produces RMS with MYOD1 induction as well as SOX8 and other core regulatory circuitry genes. Moreover, each class of FP-RMS system (Myf6- and TEK- driven) are associated with transcriptional heterogeneity, as has been seen in recent impactful studies from the Dyer and Langenau groups (Dev Cell 2022, Nat Cancer 2022).

Interestingly, at the population level, based on transcriptional patterns, there are many similarities between endothelial derived and myogenic derived FP-RMS tumors. Epigenetically there are also many similarities in FP-RMS models, regardless of myogenic or endothelial cell origins, based on H3K27ac profiles in the cells.

The authors also investigated the ability of PAX3-FOXO1, in combination with MYCN to transform human endothelial cells. With lentiviral infections, PAX3-FOXO1, human endothelial cells begin to express MYOD1 and Myogenin. Moreover, with co-transduction with TERT and MYCN, PAX3-FOXO1 is able to produce tumors from human endothelial cells that histologically and transcriptionally resemble alveolar RMS. Interestingly, human iPSCs expressing PAX3-FOXO1 during endothelial differentiation conditions were found to express myogenic cell markers and form RMS tumors with alveolar histology when injected into immunocompromised mice

Major comments:

It would strengthen the work to meta-analyze the scRNAseq clusters with the recently published scRNAseq from Wei Nat Cancer 2022 and Patel Dev Cell 2022 to identify further similarities in the gene expression patterns and heterogeneity with human derived FP-RMS.

We thank the reviewers for this suggestion and have included a comparison of our mouse scRNAseq data to published human scRNAseq data sets. We performed a comparison of the TCP and MCP mouse tumor scRNAseq data to human scRNAseq carried out by Wei et al. published in Nature Cancer in 2022¹. Wei et al. defined common cell states present in patient tumors including proliferation, muscle, interferon, hypoxia, mesenchymal-like and apoptosis cell states¹. While this analysis was successful at comparing the gene expression profiles of human and mouse FP-RMS tumors, there are some caveats to this analysis. First, the cell states identified by Wei et al. were initially found in FN-RMS tumors but were confirmed to be universally present across a spectrum of fusion negative and fusion positive tumors with some variability between samples. Additionally, not all genes driving the segregation of cell states in the human RMS samples have mouse orthologs and were not included in the analysis. To overcome this caveat, we compared the expression of cell state driving genes in addition to larger lists of genes expressed by each cell state to determine if the cell states present in the human data corresponded with the mouse FP-RMS models.

In the new Figure 4, we show a heatmap and feature plots highlighting the expression of cell state specific driver genes from the human tumors in our mouse scRNAseq data. These are the genes enriched by the different cell types used by Wei et al. to identify the RMS cell states in Fig 1c of their manuscript¹. Additionally, in Supplemental Figure 5 we took a larger list of genes expressed in each cell state (Supplemental Table 10) and include a heatmap showing the expression of these genes in our mouse scRNAseq data. This analysis shows the expression of cell state specific gene sets in discrete, agnostically generated clusters identified in our scRNAseq data from mouse tumors. The proliferative cell state is contained to cluster 1 which we identified as tumor stem cells, the mesenchymal-like cell state correlates with cluster 5 which we identified as fibroblasts/smooth muscle cells and cluster 1 which we identified as undifferentiated tumor cells, and the muscle cell state genes are expressed by cluster 7 which was identified as the most myogenic population of cells within the tumors. The expression of genes associated with these cell states was consistent between TCP and MCP tumors highlighting the similarity of TCP and MCP tumors and their accurate recapitulation of human FP-RMS. These findings demonstrate the fidelity of TCP tumors as an accurate model of FP-RMS that will be useful in developing novel therapeutic strategies to target these aggressive tumors, but also shows the reprogramming of endothelial cells expressing *Tek-Cre* to myogenic FP-RMS tumors upon P3F expression. The figures generated to address this comment are included below for ease of review.

Figure 4 Searcy and Larsen et al.

a

Cell State Signature	# of Genes	
	Mouse	Human
Apoptosis	56	80
Hypoxia	42	42
Interferon	38	46
Mesenchymal	125	133
Muscle	68	75
Proliferation	112	114

b
Supplemental Figure 5 Searcy and Larsen et al.

Are there any endothelial signatures present in human FP-RMS cell lines, PDXs, or primary patient samples? Would it be possible to compare RNA-seq signatures in the MCP, TCP, and human model systems in the manuscript to identify any endothelial regulatory TFs that are expressed? Master regulatory GATA-family TFs, and FOXF1 are each implicated in endothelial cell circuitry.

We appreciate the reviewer’s suggestion and agree that it is important to thoroughly investigate the RNAseq signatures of our FP-RMS models from an endothelial cell of origin for signatures of their original endothelial cell identity. To investigate the presence of endothelial cell gene signatures in our data we used two endothelial genes sets as comparators. First, we generated a list of endothelial specific transcription factors, including GATA-family transcription factors and FOXF1, from published endothelial cell datasets (Supplementary Table 11)². Next, we used a list of endothelial cell genes used by the PanglaoDB algorithm³ to identify endothelial cell clusters in scRNAseq data. We generated a heatmap to illustrate the expression of these endothelial genes in WT HUVECs and HUVECs with T Ag, hTERT, and MYCN to validate the fidelity of these endothelial cell gene lists to endothelial cells. Next, we determined the endothelial cell gene expression of WT iPSCs, p53^{KO} iPSCs, and xenografts from both endothelial differentiated p53^{KO} iPSCs transduced with P3F (P3F iPSC Xenografts) and HUVECs with T Ag, hTERT, and MYCN transduced with P3F (HUVEC-THMP Xenografts) and found little to no expression of genes associated with endothelial cells in any of these cell types. We also included human FP-RMS cell lines (RH28, RH18, RH30_1, RH30_2) and found that human RMS cell lines do not express endothelial gene signatures (Figure 7I). This analysis showed little expression of endothelial genes in human FP-RMS cell lines or xenograft tumors further demonstrating the complete reprogramming of endothelial cells to FP-RMS.

Additionally, we compared the expression of endothelial cell genes in the RNAseq data from MCP and TCP tumors. This analysis showed similar levels of endothelial cell gene expression in both tumor genotypes and further highlighted the similarity between MCP and TCP gene expression profiles. This data is not included in the revised manuscript but is included below for the reviewers.

The epigenetics analysis would be strengthened from identifying high confidence DNA binding motifs within the H3K27ac data the authors already have. Are cell type specific TFs implicated in the most highly enriched motifs?

Thank you for suggesting this additional analysis of the H3K27ac data. We used a targeted approach to determine if cell type specific TFs motifs are in fact highly enriched in the H3K27ac data of MCP and TCP tumors and if there are differences among the tumors. We first generated a list of endothelial transcription factors (Supplementary Table 11) and performed DNA binding motif analysis on the H3K27ac data specifically looking for endothelial transcription factor binding motifs at non-promoter H3K27ac peaks. We then plotted these results as heatmaps to compare the enrichment of active endothelial cell transcription factor motifs in the H3K27ac CUT&RUN data in MCP and TCP data (Figure 5g). In line with the RNA expression data in Figure 7, this motif analysis shows that there is virtually no difference in the presence of endothelial cell type specific transcription factor motifs within the H3K27ac data. To validate this method of identifying transcription factor motifs with the H3K27ac

CUT&RUN data, we specifically looked for the transcription factor MYOD1 which is a known myogenic/FP-RMS transcription factor that has large amounts of H3K27ac activation at the super-enhancer, enhancer, and promoter of this gene. We found increased levels of the MYOD1 binding motif in the H3K27ac CUT&RUN data from both TCP and MCP which is what we expected to see when looking at FP-RMS which highly expresses *Myod1*. Together, this analysis further reinforces that the MCP and TCP FP-RMS tumors are nearly indistinguishable despite originating from different progenitors and that TCP tumors have been completely reprogrammed to FP-RMS by P3F expression.

It would strengthen the study to determine if PAX3-FOXO1 is essential for tumor maintenance. If there is PAX3-FOXO1 depletion in the TCP mice after transformation, how does this affect proliferation or gene expression? Similarly, if the HUVECs or iPSCs are (1) transduced PAX3-FOXO1 earlier or later than TERT/MYCN does this alter oncogenic transformation or gene expression, and (2) if PAX3-FOXO1 is depleted after tumor formation in the human iPSCs or HUVECs how does this alter proliferation and gene expression?

These are very intriguing comments, and the question about P3F's requirement in maintenance remains a relatively unexplored topic in the field. First, we sought to address the question about the order of P3F, hTERT, and MYCN expression affecting proliferation and gene expression. We added

additional data showing that HUVECs transduced with P3F without T antigen, hTERT and MYCN maintain P3F expression in culture over time; however, expression of *MYOD1* (a key diagnostic mark of RMS) is lost after 12 days in culture (Supp Fig 6). This demonstrates that transduction with viruses expressing T Antigen, hTERT, or MYCN prior to P3F expression in HUVECs is required for stable MYOD1 expression and RMS transformation. A similar phenomenon was demonstrated in previous work by Naini and Etheridge et al. which showed that the acquisition of P3F, MYCN, and hTERT had to occur in a specific order for human skeletal muscle myoblasts to transform into FP-RMS⁴.

Determining the role of P3F in tumor maintenance is a very interesting avenue of research and important. We considered addressing this in our mouse models and in the newly generated *in vitro* systems. Trying to determine whether P3F depletion in TCP tumors after transformation affects proliferation or gene expression poses significant experimental challenges. We do not currently have genetic tools to stably and consistently deplete P3F in TCP tumors *in vivo*. Additionally, TCP the that develop must be euthanized for humane endpoints around 7 days after tumors become visible. We are currently attempting but have been unsuccessful to establish adherent culture cell lines and allografted transplant TCP tumors models to date. These barriers prevent us from using any depletion tools designed for *in vitro* tissues and cells without generating new genetically engineered mice.

Thus, to investigate how P3F participates in tumor maintenance and how depletion after transformation affects proliferation and gene expression, we leveraged our p53^{KO} endothelial differentiated +P3F iPSC system. Previously, we used lentiviral systems to express shRNA targeting the PAX3-FOXO1 breakpoint, Tet-pLKO-shPF1, for inducible knockdown of the oncofusion gene in human FP-RMS cell lines⁵. We redesigned the lentiviral system for constitutive expression of this validated shPF1 and instead of doxycycline-inducible expression. Leveraging our p53^{KO} iPSCs after endothelial cell directed differentiation and P3F transduction we used lentiviral transduction to express shPF1 to target the PAX3;FOXO1 breakpoint and a vector encoding an shRNA targeting firefly luciferase (shLuc) as a non-silencing control. The shPF1 transduced cells showed reduced expression of the skeletal muscle lineage factors *MYOD1*, *MYF5*, and *MYF6* compared the shLuc control transduced cells. Additionally, the shPF1 transduced cells showed significant loss of proliferation and reduction in focus forming capacity compared to the shLuc transduced controls. These P3F knock down data were included as in Supplementary Figure S10 and demonstrate that P3F is needed for tumor maintenance of the continued proliferation and myogenic gene expression profile of these cells.

Minor comments:

The font is quite small especially in Figs. 3,4,5. It would be helpful if the authors could increase font size for readers, and make sure labels are clear and consistent.

Thank you for bringing this to our attention. We have made the suggested corrections and increased the font size across these figures. Additionally, we have updated figures to ensure that labels are clear and consistent.

What are the therapeutic implications of the findings? The cell of origin in RMS has been widely debated, and work is significant. However it is unclear how identification of endothelial cells as potential cells of origin could affect clinical decision making or potential pre-clinical testing of agents. It would strengthen the manuscript if the authors could comment on this.

Thank you for this question. We agree that this is an important point and have added our comments to the discussion.

“FP-RMS tumors develop throughout the body, including tissues devoid of skeletal muscle, highlighting the potential for non-myogenic origins. Here, we identified endothelial cells as a cell of origin for FP-RMS. FP-RMS tumors from endothelial origins are nearly indistinguishable from FP-RMS tumors from myogenic cells. The possibility of forced differentiation of pediatric embryonal tumors that appear as an arrested state of normal development is conceptually intriguing. Differentiation therapy is an intriguing alternative treatment avenue which has been extensively studied in neuroblastoma, another pediatric solid tumor that can appear as an arrested state of normal development⁶. This study highlights that patient FP-RMS tumors may come from differing cells of origin and thus exhibit unique responses to differentiation therapy. FP-RMS in the head and neck originating from endothelial cells may not respond to these therapies even though they resemble tumors from a myogenic cell of origin. Future work investigating therapeutic vulnerabilities of FP-RMS tumors from different cells of origin will help inform clinical trial design in this space to better treat patients with FP-RMS.”

Reviewer #2 - Myogenic stem cells, iPSCs (Remarks to the Author):

In this manuscript, the authors provide evidence that expression of the PAX3-FOXO1 (P3F) can reprogram mouse and human endothelial progenitors to muscle stem cells that can generate fusion-positive rhabdomyosarcoma (FP-RMS). They used a variety of mouse models of endothelial-specific expression of P3F to show the ability to reprogram endothelial progenitors into myogenic stem cells that retain the capability to support regeneration of injured muscle fibers. They also show that within the context of a mouse model of Cdkn2a loss in endothelial cells P3F expression promoted FP-RMS and that P3F expression in TP53-null human iPSCs prevents their differentiation into the endothelial lineage while promoting conversion into myogenic cells that can form FP RMS tumors in immuno-compromised mice.

The data shown in this manuscript further support an emerging concept that RMS can originate from aberrant development of non-myogenic cells – in this case FP-RMS driven by P3F. At the same time, these findings suggest that P3F can reprogram somatic and stem cell types into myogenic cells that, when deficient for key regulators cell cycle checkpoints (Cdkn2 and TP53), can undergo transformation into FP-RMS.

The manuscript is of interest, but some issues need to be addressed, as outlined below

1) Overall, while this work fairly supports the hypothesis that FP-RMS can derive from endothelial cells upon ectopic expression of P3F in experimental mouse models, this evidence does not mean FP-RMS invariably originate from endothelial cells. This limitation should be acknowledged by the authors.

Thank you for this comment and we wholeheartedly agree and apologize if the work was presented as to suggest FP-RMS invariably originates from endothelial cells. We acknowledge that myogenic progenitors are a cell of origin for rhabdomyosarcoma. This has been clearly established in the literature by multiple groups showing that activation of various oncogenes in multiple myogenic progenitor cell types gives rise to fusion-positive (FP-RMS) and fusion-negative (FN-RMS) rhabdomyosarcoma^{4,7-10}. Additionally, we generated a mouse model of FP-RMS originating from myogenic cells expressing *Myf6-Cre* which was originally established by Charles Keller and Mario Capecchi^{10,11}. Here, we show that FP-RMS from a non-myogenic cell of origin in cells expressing *Tek-Cre* look nearly indistinguishable from FP-RMS tumors originating from myogenic cells expressing *Myf6-Cre*. Furthermore, we show that human endothelial cells can give rise to FP-RMS tumors. Together, these findings highlight the likelihood of patient FP-RMS tumors originating from both non-myogenic endothelial cells and myogenic cells which is contradictory to the previously held beliefs that all FP-RMS tumors derived from myogenic progenitors that fail to differentiate or satellite cells.

We modified the end of the first paragraph of the discussion as follows:

“The data presented here suggest that FP-RMS can originate from an endothelial origin in addition to skeletal muscle origins. This could explain potential origins of aggressive tumors without a known myogenic primary tumor.”

2) While it is interesting that P3F reprograms endothelial cells into muscles, the manuscript does not provide insights into the mechanism by which P3F promotes this reprogramming. Is this a direct or indirect action of P3F? Or it is mediated by the well-known reprogramming factor MyoD? This should be addressed by performing genome-wide chromatin binding (CUT and RUN or ChIPseq) of P3F and MyoD, with data integration with H3K27Ac genome-wide signal and RNAseq. Moreover, experiments within the context of MyoD null mice could also address this issue.

Thank you for this question. We agree that identifying a prospective mechanism by which P3F promotes reprogramming to FP-RMS would strengthen this manuscript. Our p53^{KO} iPSCs+P3F system provided the best opportunity to study the direct role of P3F in transformation as 1) it can be scaled to generate the amount of biomaterial necessary to perform multi-omic analyses on matched, homogenous samples and 2) it provides insight into the P3F mediated reprogramming of a human system. To define direct actions of P3F, we interrogated P3F occupancy and active enhancer and promoter regions with FOXO1 and H3K27ac CUT&RUN, respectively, and correlated these data with transcriptional output using RNAseq data on matched samples before (day 0) and after (day 15) transformation of our p53^{KO} iPSCs transduced with P3F during endothelial directed differentiation. We observed decreased expression of the TF *ERG*, a critical regulator of endothelial lineage specification and homeostasis, in transformed P3F expressing p53^{KO} iPSCs^{12,13}. This was coupled with loss of H3K27ac signal at an internal *ERG* super-enhancer, but no FOXO1 binding. This suggests that P3F engages a chromatin regulatory program that reduces H3K27ac modifications at this locus and thus indirectly prevents expression of a crucial endothelial specification factor. This is consistent with previous flow cytometry and qPCR data demonstrating loss of VE-CADHERIN in transformed P3F expressing p53^{KO} iPSCs, as VE-CADHERIN is a direct target of ERG. Additionally, we observed FOXO1 and H3K27ac binding at the known proximal and distal enhancer elements of *MYOD1*, *MYF5*, and *MYF6* only in cells expressing P3F. RNAseq data revealed that FOXO1 binding at these loci was accompanied with a strong increase in the expression of *MYOD1*, *MYF5*, and *MYF6*. These data suggest that direct binding of P3F at super-enhancers of master myogenic transcription factors promotes their expression and mediates reprogramming to a myogenic tumor cell. Taken together, this experiment demonstrates that P3F indirectly promotes repression of endothelial specification factor expression and directly promotes the transcription of master myogenic transcription factors to drive the transformation of the endothelial directed iPSCs into RMS. We have included these data in a new figure (Figure 8) at the conclusion of the results section.

Furthermore, we appreciate the astute inquiry as to whether the reprogramming to FP-RMS was simply the consequence of MYOD1 expression, as MYOD1 alone is sufficient to reprogram murine fibroblasts to myoblasts¹⁴. Indeed, the data added in Figure 7 demonstrate P3F-driven MYOD1 expression is observed in transformed cells, suggesting that it is one of many factors involved in P3F-mediated transformation. To address whether MYOD1 alone was sufficient to transform endothelial progenitor cells into FP-RMS, we repeated the endothelial differentiation experiment in p53^{KO} iPSCs with enforced MYOD1 expression. Unlike P3F, MYOD1 expression was not sufficient to block endothelial differentiation and completely reprogram cells as evidenced by co-localization of MYOD1 and CD31 in differentiated cells and by persistence of endothelial markers CD31 and CD34 by flow cytometry. Additionally, the MYOD1 expressing cells displayed decreased viability, proliferation, and focus formation compared to the P3F expressing cells. Finally, the reduced proliferation of the

MYOD1 expressing p53^{KO} iPSCs eliminated the ability to test tumorigenicity in a xenograft model due to the lack of sufficient cell numbers. These experiments illustrate that MYOD1 is not sufficient to drive reprogramming and transformation of p53^{KO} iPSCs into FP-RMS. We have added these data to Supplementary Figure S9 in the manuscript.

3) Why TP53 deficiency is required to promote selective conversion of hiPSCs into myogenic cells (with repression of endothelial cell lineage) that can form FP RMS tumors in immunocompromised mice?

Thank you for pointing out this lack of clarity. We do not know if TP53 deficiency is required for the conversion of the hiPSCs into myogenic cells following PAX3-FOXO1 expression as we did not perform the directed differentiation experiment in TP53-WT hiPSCs. The rationale guiding our use of *TP53* deletion in our iPSC experiments included prior observations in both GEMMs and in human primary myoblast models. First, the original reports from Charles Keller and Mario Capecchi of the PAX3-FOXO1 mouse models eloquently demonstrated that only 1 out of 200 mice expressing *Myf6-Cre;Pax3^{P3Fm/P3Fm}* developed tumors with wild type *Trp53*¹⁰. Upon knockout of either *Trp53* or *Cdkn2a*, mice developed tumors with 100% penetrance¹⁵. Additionally, Corinne Linardic's group determined that primary human myoblasts required sequential enforced expression of P3F, hTERT, and MYCN for tumorigenesis to occur⁴. Linardic's work nicely illustrated that PAX3-FOXO1 expression in primary human myoblasts led to hypermethylation and attenuated expression of *CDKN2A* (both *INK4A* and *ARF*) and loss of *CDKN2A* was required for tumorigenicity of the modified cells^{4,8}. Given similar results with both deletion of *Trp53* or *Cdkn2a* in the GEMMs, we speculated that the deletion of *Trp53* and not *Rb1* was driving the phenotype. As we postulated the iPSC endothelial directed differentiation experiments, we chose to delete *TP53* using gene editing as we did not know if PAX3-FOXO1 expression in this system would result in the epigenetic silencing that Linardic observed in primary human myoblasts. As well, the background of *TP53* loss would provide a robust background to observe PAX3-FOXO1-mediated transformation. The *TP53*^{KO}-iPSC system provided us an initial proof-of-principle of human endothelial progenitors transdifferentiation into FP-RMS. It is interesting to note that the *TP53* deletion did not limit the ability of the iPSC to undergo endothelial directed differentiation, nor did *TP53* deletion alone result in transformation and tumor development in the iPSCs. Although in children FP-RMS tumors with *TP53* loss have been shown to occur at a low frequency, they are universally fatal showing the significance of the interplay between *TP53* loss and P3F expression in a clinically relevant group of patients with a particularly poor survival¹⁶. Whether *TP53* loss is required is currently unknown. Now that we have developed an *in vitro* system that is easily genetically manipulated, we can carefully address mechanistic questions such as the requirement for *TP53* loss.

To address this more succinctly, we added the following statement in the results:

We leveraged directed differentiation of induced pluripotent stem cells (iPSCs) to direct P3F expression along endothelial differentiation (Fig. 7a). **The original reports of the PAX3-FOXO1 mouse models eloquently demonstrated that only 1 out of 200 mice expressing *Myf6-Cre;Pax3^{P3Fm/P3Fm}* developed tumors with wild type *Trp53*.¹⁰ Upon knockout of either *Trp53* or *Cdkn2a*, mice developed tumors with 100% penetrance.¹⁵**

4) There are concerns with significance of HiC datasets and their analysis. What is the resolution of the analysis? It looks like the resolution is not high enough to detect other structures than TADs. Figure 4b and c compare TADs at specific loci in MPC and TCP, but no control cell of origin is shown (necessary)

Thank you to the reviewers for these comments. We apologize for the lack of clarity on the HiC analysis. We performed analysis of the HiC data at multiple resolutions from 1Mb to 5kb and chose to only visualize the 1Mb analysis because it showed the largest differences between MCP and TCP

tumors. We have now included visualizations of the HiC contact frequencies at finer resolutions with the location of myogenic and endothelial genes to show the similarity of MCP and TCP chromatin architecture at higher resolutions (Figure 5a). Similar to the 1 Mb analysis, the HiC reads at 10 kb resolution also illustrate that chromatin surrounding both skeletal muscle and endothelial genes is organized similarly in TCP and MCP. Additionally, we examined embryonic stem cell HiC data at a similar resolution to the MCP and TCP HiC data as a control cell, which shows distinct structures from our RMS tumors (Figure 5b). These findings lend further support to our initial observation that genome structure at important RMS genes is not materially different between TCP and MCP tumors. The scatterplot of TCP and MCP HiC reads at 10 kb resolution and HiC interaction heatmap with embryonic stem cell included are shown below for convenience.

Reviewer #3 - RMS mouse models (Remarks to the Author):

In the manuscript entitled PAX3-FOXO1 dictates myogenic reprogramming and rhabdomyosarcoma identity in endothelial progenitors, Searcy et al show that endothelial cells can be a cell of origin for fusion positive rhabdomyosarcoma (FP-RMS). The same group has previously shown that fusion negative RMS (FN-RMS) can have a non-myogenic origin. Further, based on the finding by Vicente-Garcia et al, showing that cis-regulatory elements driving endothelial expression of FOXO1 in humans and mice regulate PAX3-FOXO1 (P3F) expression, the authors hypothesise that endothelial specific expression of P3F could cause transdifferentiation of vasculature giving rise to FP-RMS. To test this hypothesis the authors, induce P3F expression and Cdkn2a deletion using various Cre mouse lines: aP2-Cre (ACP), Tek-Cre (TCP) and the previously described Myf6-Cre (MCP). While in ACP mice P3F reprograms endothelial progenitors to functional myogenic stem cells capable of regenerating injured muscle fibers, in TCP mice P3F accurately recapitulates human FP-RMS through histologic profiling and gene expression. Importantly, while TCP and MCP tumors originate from different cell types, they share nearly indistinguishable histology, gene expression, and chromatin architecture. In TP53-null human iPSCs P3F blocks endothelial-directed differentiation and guides cells to become myogenic cells that form FP-RMS tumors in immunocompromised mice. Overall, this work demonstrate that P3F has the potential to reprogram mouse and human endothelial progenitors to FP-RMS an could explain why in humans FP-RMS can occur in several sites in the body devoid of skeletal muscle.

This is a very interesting study that highlights how transdifferentiation plays an important role in cancer and in FP-RMS in particular. The study will be of interested for those studying cells-of-origin in cancer (which in sarcoma remain elusive for most sarcoma subtypes), and sarcomagenesis in general, but it also has implications in the field of muscle regeneration and cell plasticity/transdiferentiation. The following points should be clarified:

1. FP-RMS tumors from TCP and MCP mice accurately recapitulate human disease through histologic profiling, gene expression and enhancer state. Together with the data shown in human Huvecs and iPSCs the authors clearly show that P3F is capable of reprogramming endothelial cells to FP-RMS. However, the results from all the Cre lines (aP2-Cre and Fapbb vs Tek-cre and Myf6-Cre) are not very easy to follow. One of the main issues in terms of lack of clarity given the several models described is exactly which endothelial cell population can serve as cell of origin for FP-RMS. In ACP mice P3F reprograms endothelial cells to functional muscle stem cells but does not give rise to FP-RMS. In Fabp4-Cre which exhibit higher Cre-recombination efficiency 3 out of 20 mice develop FP-RMS. In Tek-Cre mice P3F is not able to reprogram endothelial cells to functional muscle stem cells but gives rise to FP-RMS with much higher efficiency. So, the question becomes: is there a common endothelial cell population in aP2-Cre (or Fabp4-Cre) and Tek-Cre mice that can give rise to FP-RMS upon P3F expression?

Why does PF3 reprogram endothelial cells to muscle in aP2-Cre mice but not in Tek-Cre mice if they recombine the P3F allele in similar endothelial cell populations? Indeed, the authors use scRNAseq to show that both aP2-Cre and Tek-Cre mediate recombination in the same cell types in the SCM including endothelial and hematopoietic cells and the authors state that Tek-Cre is expressed by endothelial cells labelled by aP2-Cre. This is confusing and not clearly discussed in the manuscript. The authors should consider the option of comparing the different Cre lines within a figure instead of scattering them through the main figures and supplementary material.

Thank you for this comment and apologize for any confusion regarding the different Cre lines used in this manuscript. We struggled with the best way to present our observations without getting much too granular and given the space constraints. The work of the lab has largely leverage the *aP2-Cre* mouse that uses the immediate 5.4 kb cis-regulatory region upstream of the transcriptional start of the *adipose protein 2 (aP2)* gene. We have reported that *aP2-Cre* mediates Cre recombination in endothelial cells within the interstitium of skeletal muscle and constitutive activation of the sonic hedgehog pathway in these cells results in cell reprogramming and transdifferentiation of these cells into fusion-negative RMS¹⁷⁻¹⁹. First, we sought to determine if expression of the PAX3-FOXO1 oncofusion protein would result in transformation of these endothelial cells into FP-RMS similar to SHH pathway activation led to FN-RMS. P3F expression and *Cdkn2a* loss in *aP2-Cre* expressing cells initiates myogenic reprogramming to functional muscle stem cells capable of regenerating injured muscle, however, these cells are not transformed. *aP2* is also known as FABP4. Ron Evans generated an independent transgenic mouse, *Fabp4-Cre*, using the same *aP2* 5.4 kb cis-regulatory region to dictate Cre recombinase expression. In our prior work in SHH-driven FN-RMS, we have noted differences in the magnitude of phenotypes between the *aP2-Cre* and *Fabp4-Cre*. Specifically, FN-RMS models derived from *Fabp4-Cre* were embryonic lethal with RMS while *aP2-Cre* mice are born with RMS. Therefore, we sought to see if the *Fabp4-Cre* would similarly have an enhanced phenotype in our P3F studies. We confirmed that *Fabp4-Cre;Cdkn2a^{Flox/Flox};Pax3^{P3F/P3F}* mice develop FP-RMS. This is likely due to an increased Cre expression in *Fabp4-Cre* mice resulting in increased Cre-mediated recombination. *aP2-Cre* and *Fabp4-Cre* have both been widely utilized to study adipose biology and not well-recognized as mediated recombination in endothelial cells. It is now clear that both *aP2-Cre* and *Fabp4-Cre* mediate Cre recombination in a multitude of cell types including brown and white adipose cells, endothelial cells, dorsal root ganglia, cartilage, and macrophages. We were interested to study the effects of P3F expression in another more endothelial-restricted Cre line so we generated *Tek-Cre;Pax3^{P3F/P3F};Cdkn2a^{Flox/Flox}* mice. We determined that these cells are not reprogrammed to functional muscle stem cells; however, *Tek-Cre;Cdkn2a^{Flox/Flox} Pax3^{P3F/P3F}* mice develop FP-RMS. The exact mechanism behind the differences in phenotypes from *aP2/Fabp4/Tek-Cre* lines is unknown.

To determine if *aP2-Cre*, *Fabp4-Cre*, and *Tek-Cre* are expressed by the distinct or similar endothelial populations that may be the cells of origin for FP-RMS, we compared the single cell RNA sequencing profiles of Tomato positive cells from the sternocleidomastoid (SCM) neck skeletal muscle of adult *aP2-Cre;R26-tdTom*, *Fabp4-Cre;R26-tdTom*, and *Tek-Cre;R26-tdTom* mice and determined that they all share expression of the same cell type clusters in adult tissue. These data are shown below and illustrate that while there are subtle differences the magnitude of representation each Cre-driver is represented in each cluster. It is important to consider that Tomato labelling of cells from these models is an indelible mark and it is impossible to know when in the lifespan of the cell Cre-mediated recombination occurred. One potential explanation for the disparate phenotypes could result from the timing of Cre expression during the endothelial cell development. Further investigation into the embryonic origins of expression may provide further insight into the differences between these three lines.

We've added to the discussion to clarify the points made above:

“We hypothesize that the cell type and context of P3F expression is important when determining its effect on differentiation. Although our lineage tracing shows *aP2-Cre*, *Fabp4-Cre*, and *Tek-Cre* share expression in the same cell types in adult tissue, P3F cooperates with *Cdkn2a* loss in each model to give rise to a unique phenotype. One potential explanation for the disparate phenotypes is the timing of Cre expression during endothelial development. Tomato expression is an indelible mark in this system, rendering it impossible to determine when in the lifespan of the cell Cre-mediated recombination occurred. Further investigation into the embryonic origins of *aP2-Cre*, *Fabp4-Cre*,

and *Tek-Cre* may provide insight into the mechanism governing why different phenotypes arise in response to enforced P3F expression. Our work illustrating that P3F maintains the ability to reprogram *aP2-Cre* expressing cells without transforming them highlights the intricate balance between transformation and cell state change.”

Determining the exact cell of origin is an ongoing extensive effort that is well beyond the scope of this project that illustrates the capacity of endothelial cells as an origin of FP-RMS. We hope that further interrogation of both our mouse models and the human iPSC system will allow us to define the timepoints and exact mechanism of reprogramming giving rise to FP-RMS from a non-myogenic progenitor.

2. It should be clear in the text why the authors compare P3F activation in hindlimb and neck sternocleidomastoid muscle (SCM) of ACP mice and why those are expected to show different results. On one hand muscles in the head and neck have distinct embryonic origins from somite-derived trunk and limb muscles. On the other hand, previous studies from the same lab shown that aP2-Cre-labeled interstitial endothelial cells adjacent to the myofibers in the SCM are the ones giving rise to FN-RMS in response to SmoM2 expression (Drummond et al 2018). Such cells are not present in hindlimb muscle and FN-RMS develops exclusively in head and neck (correct?). This should be briefly mentioned as it is not totally clear why in Fig.1 SCM and hindlimb muscle are compared or if the disparate results are expected.

Thank you for bringing this lack of clarity to our attention. You are absolutely correct in your assessment of why we compared ACP hindlimb and SCM skeletal muscle. We have added a more detailed explanation of why these tissues were compared to give further rational behind Fig.1.

“Similar to *aP2-Cre;R26-tdTom* mice, Tom⁺ cells in ACP mice were located between skeletal muscle fibers in the interstitium in muscle tissue throughout the body. We compared skeletal muscle from the hindlimb and the SCM given the locational specificity of FN-RMS tumors originating from *aP2-Cre* expressing cells in the SCM^{17,18}.”

3. Line 160: It is unclear why if only 10% of isolated muscle stem cells from SCM of ACP mice are tdTomato positive (compared to about 80% in PAX7CreERT2) (Fig 1d-e) how almost 100% of the myofibers are tdTomato+ following injury (Fig1f). Does this mean that existent muscle stem cells (which should be negative for tdTomato) have very little to no contribution to muscle regeneration in contrast to tdTomato + P3F +cells?

This is an excellent point. The impact of the Tomato positive or Cre-expressing muscle stem cells on the regenerating muscle fiber is difficult to discern. If one tdTomato⁺ cell or a cell expressing Cre recombinase fuses to a skeletal muscle fiber, Cre recombinase will recombine the reporter in the muscle fiber and thus result in Tomato expression in the across the entire muscle fiber, instead of only what is carried over from the fused muscle stem cell. This explains how fewer reprogrammed muscle stem cells from ACP mice (10%) can result in almost 100% of the myofibers expressing tdTomato.

4. Line 146: “Surprisingly, of 940 Tom+ cells counted in the SCM of ACP mice, 30% co-localized with PAX7. None of the 66 PAX7+ muscle stem cells counted beneath the basal lamina were Tom+. Tom+ cells in the hindlimb of ACP mice did not co-localize with PAX7 (Fig. 1b).” It is unclear if this co-localization is driven by P3F expression. The same staining in aP2-Cre, tdTomato mice in the absence of P3F should be shown.

Thank you for this comment. We have previously shown that tdTomato⁺ cells in the SCM are distinct from PAX7⁺ skeletal muscle stem cells in *aP2-Cre;R26-tdTom* compound mutant mice using confocal images with 3D-reconstruction. This is equivalent to the staining described (staining for tdTomato and PAX7 in the absence of P3F), so we did not include it in the manuscript¹⁷.

5. The authors mention in the Introduction (line 110) that P3F has been shown to block endothelial differentiation by decreasing FOXC2 expression (ref 24,25). However, this is never again mentioned in the manuscript even though the authors use the appropriate models to test it. Is FOXC2 expression down regulated in differentiating iPSCs expressing P3F for example?

Thank you for bringing this to our attention. By bringing up FOXC2, we intended to highlight Margaret Buckingham's prior work on a common mesodermal progenitor in the somite in which cell fate decisions to either skeletal muscle or endothelial cells are governed by opposing transcriptional

equilibrium from transcription factors PAX3 and FOXC2 respectively. Currently, the level of FOXC2 expression in transformed P3F expressing iPSCs is unclear. RNAseq data shows strong signal at the 3'-UTR of FOXC2 with minimal coverage of the main exon, likely the result of 3' bias often observed in mRNAseq. This data is shown below but has not been added to the manuscript.

We predict that in this system FOXC2 expression increases at an intermediate point during the endothelial differentiation and then is lost prior to forming a mature endothelial cell. It is possible that P3F binding we observe at the FOXC2 promoter could have repressed FOXC2 expression at that intermediate time point and permitted the transformation of these cells to FP-RMS. Alternatively, there are TFs other than FOXC2 that function in the dynamic equilibrium with PAX3 dictating endothelial or myogenic cell fate, respectively. Identifying when individual TFs associated with endothelial differentiation, such as FOXC2, are expressed and if they are associated with P3F binding during the process of endothelial differentiation is an extensive ongoing effort which is beyond the scope of our intention in this manuscript.

6. Although the authors most times include the number of replicates in the figure legends, some experiments only show representative images and no statistics (eg standard deviation)(Fig1b, c for example). In the text results are expressed as an overall % but are lacking statistics. Example, Line 155: “We found that dissociated ACP cells from the SCM but not from the hindlimb formed myotubes evident by the co-localization of Tomato and myosin heavy chain (MyHC) staining in 51% of Tom+ cells (Fig. 1c)”. the authors should represent these quantifications in graphs so that error bars corresponding to replicates are shown.

Thank you for pointing out this inconsistency. We have checked all figure legends and made sure to include all information about replicates and statistics. Regarding the example given (Fig1b and c), we were unable to perform statistics because zero Tom+ ACP cells from the hindlimb formed MyHC+ myotubes which prevented us from performing statistics on this data. We have included the exact numbers of cells counted in the text to provide more clarity to the reader.

7. In the discussion the authors mention that: “In the mouse model used for this study, P3F expression is driven by the Pax3 promoter and influenced by the regulatory elements of Foxo1 knocked into the Pax3 locus. However, this model lacks downstream regulatory elements over 100 kb from the TSS required to drive FOXO1 expression in vascular endothelial cells. Forcing the expression of P3F using this genetically engineered mouse model in endothelial cells

circumvents the lack of distal *Foxo1* cis-regulatory elements in the *Pax3:Foxo1* allele to drive P3F expression robustly and completely in this cell type”

Maybe I’m missing something but if P3F expression is driven by the *Pax3* promoter and the model lacks downstream regulatory elements over 100 kb from the TSS required to drive *FOXO1* expression in vascular endothelial cells, how is the expression of P3F “forced” in these cells? *Pax3* should not be expressed in endothelial cells if I understood properly?

Thank you for this great point, we have provided more clarity. You are correct that PAX3 should not be expressed in mature, differentiated endothelial cells. However, there is a PAX3 positive progenitor in the developing dermomyotome gives rise to both endothelial and myogenic cells²⁰⁻²². We predict that the expression of P3F is permitted in that embryonic progenitor which then begins to transform into an FP-RMS cell. This is a current area of investigation in our laboratory, and we are working to define the early genetic and epigenetic changes of these progenitor cells as they differentiate to RMS. We have added to the discussion to clarify the point above:

“Forcing the expression of P3F using this genetically engineered mouse model in endothelial cells that arose from a PAX3+ progenitor in the dermomyotome circumvents the lack of distal *Foxo1* cis-regulatory elements in the *Pax3:Foxo1* allele to drive P3F expression robustly and completely in this cell type.”

Minor points

- Figure 1A should show also a control muscle condition that was not treated with cardiotoxin, and if possible, a marker of muscle regeneration to show that the cardiotoxin treatment is effective.

Thank you for this comment. Both images in Figure 1A show myotubes with centrally located DAPI+ nuclei, a classic hallmark of injured and regenerating muscle²³. This can also be observed in the injured vs. uninjured images shown in Figure 1F. This demonstrates that the cardiotoxin treatment was effective in causing muscle injury. Additionally, we have previously shown that Tom+ cells from *aP2-Cre;R26-tdTom* compound mutant mice do not contribute to skeletal muscle, which is equivalent to the uninjured muscle condition described, so we did not include it in this manuscript¹⁷.

C

C)
Immunostaining of
AT SCM cross-
sections showing
Tomato (red),
MHC (green),
LAMININ
(magenta), and
DAPI (blue)(n=3).
Scalebar, 50 μ m.

- Line 186: “Preformed” instead of performed.

Thank you, we have made the correction.

- Line 200: The authors optimize the “liberal gating” to exclude the probability of contamination by Tomato- cells. However, td-tomato signal is quite bright and two independent populations should in principle be identified without the need to gate for the top 50% positive cells. The original gating should be shown in supplementary figure so that it is clear to what extent tdTomato positive and negative cells can be distinguished. Or do the author think cell duplets could explain the intermedium contaminating signals?

This is a great point and we do believe that cell duplets likely explain the contaminating signals seen in the single cell RNAseq. Additionally, when the cells go through the flow cytometer, there is always some level of contamination observed in any experiment. This is not a large problem when you are performing bulk RNAseq on sorted populations, however it begins to impact the results when single analysis is done. Here, we separated the top 50% of tdTomato expressing cells, re-sorted the cells through the flow cytometer on a very slow speed, and confirmed the quality with some of the sample being re-sorted to ensure purity of the tomato positive population.

- Line 219: To test this (...) mice did not co-localize” this phrase is not well constructed.

Thank you for bringing this issue to our attention. To provide more clarity, we have changed those lines to:

To test this **in our system, we interrogated whether Tom⁺ cells from the SCM and hindlimb of *Tek-Cre; R26-tdTom* expressed muscle stem cell markers or could function as a muscle stem cell.** Tom⁺ cells from the SCM and hindlimb of *Tek-Cre;R26-tdTom* mice did not co-localize with PAX7, were not identified by a muscle stem cell FACS method, did not fuse with myoblasts in an *in vitro* myogenic differentiation assay, and did not participate in muscle injury repair *in vivo*, **demonstrating that *Tek-Cre* did not drive recombination in muscle stem cells in our mouse models** (Supplementary Fig. 2a-d).

References

- 1 Wei, Y. *et al.* Single-cell analysis and functional characterization uncover the stem cell hierarchies and developmental origins of rhabdomyosarcoma. *Nat Cancer* **3**, 961-975 (2022). <https://doi.org/10.1038/s43018-022-00414-w>
- 2 Kalucka, J. *et al.* Single-Cell Transcriptome Atlas of Murine Endothelial Cells. *Cell* **180**, 764-779 e720 (2020). <https://doi.org/10.1016/j.cell.2020.01.015>
- 3 Li, J., Yu, C., Ma, L., Wang, J. & Guo, G. Comparison of Scanpy-based algorithms to remove the batch effect from single-cell RNA-seq data. *Cell Regeneration* **9** (2020).
- 4 Naini, S. *et al.* Defining the cooperative genetic changes that temporally drive alveolar rhabdomyosarcoma. *Cancer research* **68**, 9583-9588 (2008). <https://doi.org/10.1158/0008-5472.CAN-07-6178>
- 5 Hanna, J. A. *et al.* PAX3-FOXO1 drives miR-486-5p and represses miR-221 contributing to pathogenesis of alveolar rhabdomyosarcoma. *Oncogene* **37**, 1991-2007 (2018). <https://doi.org/10.1038/s41388-017-0081-3>
- 6 Matthay, K. K. *et al.* Treatment of high-risk neuroblastoma with intensive chemotherapy, radiotherapy, autologous bone marrow transplantation, and 13-cis-retinoic acid. Children's Cancer Group. *N Engl J Med* **341**, 1165-1173 (1999). <https://doi.org/10.1056/NEJM199910143411601>
- 7 Linardic, C. M., Downie, D. L., Qualman, S., Bentley, R. C. & Counter, C. M. Genetic modeling of human rhabdomyosarcoma. *Cancer research* **65**, 4490-4495 (2005). <https://doi.org/10.1158/0008-5472.CAN-04-3194>
- 8 Hettmer, S. *et al.* Sarcomas induced in discrete subsets of prospectively isolated skeletal muscle cells. *Proceedings of the National Academy of Sciences of the United States of America* **108**, 20002-20007 (2011). <https://doi.org/10.1073/pnas.1111733108>
- 9 Linardic, C. M. *et al.* The PAX3-FKHR fusion gene of rhabdomyosarcoma cooperates with loss of p16INK4A to promote bypass of cellular senescence. *Cancer research* **67**, 6691-6699 (2007). <https://doi.org/10.1158/0008-5472.CAN-06-3210>
- 10 Keller, C. *et al.* Alveolar rhabdomyosarcomas in conditional Pax3:Fkhr mice: cooperativity of Ink4a/ARF and Trp53 loss of function. *Genes and Development* **18**, 2614-2626 (2004).
- 11 Keller, C., Hansen, M. S., Coffin, C. M. & Capecchi, M. R. Pax3:Fkhr interferes with embryonic Pax3 and Pax7 function: implications for alveolar rhabdomyosarcoma cell of origin. *Genes Dev* **18**, 2608-2613 (2004). <https://doi.org/10.1101/gad.1243904>
- 12 Birdsey, G. M. *et al.* The endothelial transcription factor ERG promotes vascular stability and growth through Wnt/ β -catenin signaling. *Developmental cell* **32**, 82-96 (2015).
- 13 Kalna, V. *et al.* The transcription factor ERG regulates super-enhancers associated with an endothelial-specific gene expression program. *Circulation research* **124**, 1337-1349 (2019).
- 14 Davis, L. R., Weintraub, H. & Lassar, A. B. Expression of a single transfected cDNA converts fibroblasts to myoblasts. *Cell* **51**, 987-1000 (1987).
- 15 Nishijo, K. *et al.* Credentialing a preclinical mouse model of alveolar rhabdomyosarcoma. *Cancer Res* **69**, 2902-2911 (2009). <https://doi.org/10.1158/0008-5472.CAN-08-3723>
- 16 Shern, J. F. *et al.* Genomic Classification and Clinical Outcome in Rhabdomyosarcoma: A Report From an International Consortium. *J Clin Oncol*, Jco2003060 (2021). <https://doi.org/10.1200/jco.20.03060>
- 17 Drummond, C. J. *et al.* Hedgehog Pathway Drives Fusion-Negative Rhabdomyosarcoma Initiated From Non-myogenic Endothelial Progenitors. *Cancer Cell* **33**, 108-124 e105 (2018). <https://doi.org/10.1016/j.ccell.2017.12.001>
- 18 Hatley, M. E. *et al.* A mouse model of rhabdomyosarcoma originating from the adipocyte lineage. *Cancer Cell* **22**, 536-546 (2012). <https://doi.org/10.1016/j.ccr.2012.09.004>

- 19 Langdon, C. G. *et al.* Synthetic essentiality between PTEN and core dependency factor PAX7 dictates rhabdomyosarcoma identity. *Nat Commun* **12**, 5520 (2021).
<https://doi.org/10.1038/s41467-021-25829-4>
- 20 Kardon, G., Campbell, J. K. & Tabin, C. J. Local extrinsic signals determine muscle and endothelial cell fate and patterning in the vertebrate limb. *Dev Cell* **3**, 533-545 (2002).
- 21 Lagna, M. *et al.* Pax3:Foxc2 reciprocal repression in the somite modulates muscular versus vascular cell fate choice in multipotent progenitors. *Dev Cell* **17**, 892-899 (2009).
<https://doi.org/10.1016/j.devcel.2009.10.021>
- 22 Mayeux-Louchart, A. *et al.* Notch regulation of myogenic versus endothelial fates of cells that migrate from the somite to the limb. *Proc Natl Acad Sci U S A* **111**, 8844-8849 (2014).
<https://doi.org/10.1073/pnas.1407606111>
- 23 Schmalbruch, H. The morphology of regeneration of skeletal muscles in the rat. *Tissue cell* **4**, 673-692 (1976). [https://doi.org/10.1016/0040-8166\(76\)90039-2](https://doi.org/10.1016/0040-8166(76)90039-2)

Reviewers' Comments:

Reviewer #1:

Remarks to the Author:

Major comments –

Overall the authors have addressed all of my comments and suggestions, and I support publication. This will be a great contribution to the literature. I have included minor comments below to improve clarity.

Minor Comments-

Line 286-287- "Both TCP and MCP express Pax3-Foxo1 however both groups exhibit heterogeneity in Myod1 and Myogenin expression (Fig. 2e)" but it seems like same trend was observed for P3F as well?

Line 317 "MCP 2 entirely lacks cluster 0, a population like cluster 0 in TCP that lacks an obvious unifying identity" Is this MCP2 referring to MCP 395 in Supplementary Figure S4? Same issue with MCP 4 and 5, discrepancies between figure and text, please clarify.

Lines 500-503- "However, no mice injected with p53KO iPSCs cells transduced with lenti-PAX3 developed tumors observed for more than 282 days post injection. Lenti-FOXO1 transduced p53KO iPSCs stop proliferating before the completion of the 14-day endothelial differentiation protocol". Have they quantified these results in Supplementary Figure S9?

Line 512-513- "These data demonstrate that unlike P3F, enforced expression of MYOD1, PAX3, or FOXO1 alone are not sufficient to drive reprogramming and transformation of p53KO iPSCs into FP-RMS". Have the authors shown each of these factors with data in Supplementary Figure S9?

Figures-

Overall, Figure captions can be more descriptive of which cells and mice were used. More consistency in how statistical significance is denoted could be made more standardized across Figures as well.

Figure 1a, Specifying that after cardiotoxin (CTX) injury on the figure itself might be helpful.

Figure 3b, It would be nice to define MuSCs and Primary MB in the Figure legend.

Figure 6b is missing the statistical significance denotations across all four charts.

Figure 8 and associated text - as the authors intended, their anti-FOXO1 ChIP-seq is measuring the fusion PAX3-FOXO1, so they should avoid any confusion by labeling the FOXO1 ChIP condition either as "anti-FOXO1 ChIP" or as "(PAX3)FOXO1 ChIP" which would avoid any ambiguity.

Line 1523 through 1525, can the authors clarify "AT" and "FT" that are on Supplementary Figure S1h and S1j? It seems that aP2-Cre;R26-tdTom is AP and Fabp4-Cre;R26-tdTom as FP. But better to have it in the caption in brackets

Figure S1s is missing statistical significance denotation.

Figure S6a, the Figure does not state what is been increased on X- axis even though it is mentioned in the caption. Would improve clarity to define units on the x-axis itself. Further, on MyoD1 expression (middle) figure, the statistical significance denotations are missing.

Figure S7a, It would improve clarity to mention what expression is measured on the y-axis itself even though it is in the caption.

Formatting-

Lines- 391-393-Typos, seems like "TCP" is misspelled as "TPC".

Line 1395- Caption is missing the denotation "d" that represents flow cytometry data.

Line 1397- There is a bracket around (e) but not on the rest of sub-Figures in the legend.

Reviewer #2:

Remarks to the Author:

The authors have largely addressed my comments (as well as those from other reviewers) and I have nothing else to do than recommending this interesting interesting article for publication and complimenting with the authors for the excellent work

Reviewer #3:

Remarks to the Author:

The authors have address all questions and I don't have additional comments. Thanks

We greatly appreciate the reviewers' thoughtful comments and suggestions to increase the clarity and rigor of our study. Below are our replies to reviewer #1's additional comments.

Reviewer #1 (Remarks to the Author):

Major comments –

Overall the authors have addressed all of my comments and suggestions, and I support publication. This will be a great contribution to the literature. I have included minor comments below to improve clarity.

Minor Comments-

Line 286-287- “Both TCP and MCP express Pax3-Foxo1 however both groups exhibit heterogeneity in Myod1 and Myogenin expression (Fig. 2e)” but it seems like same trend was observed for P3F as well?

Thank you for asking for more clarity. We agree. The sentence has been changed to the following:

Both TCP and MCP display heterogenous expression of *Pax3-Foxo1*, *Myod1*, and *Myogenin* (Fig. 2e).

Line 317 “MCP 2 entirely lacks cluster 0, a population like cluster 0 in TCP that lacks an obvious unifying identity” Is this MCP2 referring to MCP 395 in Supplementary Figure S4? Same issue with MCP 4 and 5, discrepancies between figure and text, please clarify.

We greatly appreciate you catching this. In the original submission, we had labeled them 1,2,3,4 and 5, but changed in the resubmission to reflect exactly the nomenclature in the GEO submission. It now reads as follows:

MCP 395 entirely lacks cluster 0, a population like cluster 0 in TCP that lacks an obvious unifying identity. Myogenic tumor clusters 1 and 3 have varying representation across MCP samples. Additionally, MCP 46 and 48 have little to no expression of cluster 8, a myocyte population that expresses high levels of *Myod1* and *Myog* (Supplementary Fig. 4i-l).

Lines 500-503- “However, no mice injected with p53KO iPSCs cells transduced with lenti-PAX3 developed tumors observed for more than 282 days post injection. Lenti-FOXO1 transduced p53KO iPSCs stop proliferating before the completion of the 14-day endothelial differentiation protocol”. Have they quantified these results in Supplementary Figure S9?

We apologize for the lack of clarity. The lenti-PAX3 transduced p53KO iPSC are markedly less proliferative and there were few cells available for injection after completion of the endothelial differentiation protocol. In the lenti-P3F transformed p53KO iPSCs, we injected 4×10^6 cells/mouse. We were only able to collect enough cells to inject 3 mice. None of these mice developed tumors and were sacrificed at days 87, 172, and 282 after injection for issues secondary to immunocompromise. None of these mice had tumors on necropsy. The p53KO iPSCs transduced with lenti-FOXO1 stop proliferating during the 14-day endothelial differentiation protocol and there were no cells remaining to perform downstream assays or

inject for xenografts.

Lenti-FOXO1 transduced p53^{KO} iPSCs stop proliferating before the completion of the 14-day endothelial differentiation protocol abrogating further study.

Line 512-513- "These data demonstrate that unlike P3F, enforced expression of MYOD1, PAX3, or FOXO1 alone are not sufficient to drive reprogramming and transformation of p53KO iPSCs into FP-RMS". Have the authors shown each of these factors with data in Supplementary Figure S9?

See above comment regarding the lack of ability to further interrogate lenti-FOXO1 transduced p53KO iPSC secondary to lack of cell proliferation. There were simply no cells to study at the end of the endothelial differentiation protocol. Please see comment above for lenti-PAX3 transduced cells.

Figures-

Overall, Figure captions can be more descriptive of which cells and mice were used. More consistency in how statistical significance is denoted could be made more standardized across Figures as well.

Figure 1a, Specifying that after cardiotoxin (CTX) injury on the figure itself might be helpful. Thank you for the suggestions. We have added "injured" to the panel to provide clarity.

Figure 3b, It would be nice to define MuSCs and Primary MB in the Figure legend. Thank you for the suggestions. We have defined MuSC and primary MB in the legend as below: (MuSCs = tissue-resident skeletal muscle stem cells, Primary MBs = committed primary myoblasts).

Figure 6b is missing the statistical significance denotations across all four charts. We appreciate your point. In panel, we were simply illustrating the relative expression of *P3F*, *MYCN*, *MYOD1*, and *MYOG* in the transformed HUVECs and include the LHCN myoblasts and human Rh41 FP-RMS cell lines as points of reference. Our intention was not to directly compare the expression of each to the other, but simply to illustrate that these transcripts were being expressed, not the extent to which they are similar or deviate from either LHCN, Rh41 or each other. We agree that we could provide statistical significance, but it would not provide additional insight and could cause unintended confusion to the point of the panel.

Figure 8 and associated text - as the authors intended, their anti-FOXO1 ChIP-seq is measuring the fusion PAX3-FOXO1, so they should avoid any confusion by labeling the FOXO1 ChIP condition either as "anti-FOXO1 ChIP" or as "(PAX3)FOXO1 ChIP" which would avoid any ambiguity.

Thank you for pointing this out. We have changed the labels to read "C&R" on the figure and added CUT & RUN (C&R) in the legend.

Line 1523 through 1525, can the authors clarify “AT” and “FT” that are on Supplementary Figure S1h and S1j? It seems that aP2-Cre;R26-tdTom is AP and Fabp4-Cre;R26-tdTom as FP. But better to have it in the caption in brackets

Thank you. We edited the supplementary figure legend as suggested.

Figure S1s is missing statistical significance denotation.

We appreciate you bringing this to our attention. We added missing statistical significance.

Figure S6a, the Figure does not state what is been increased on X- axis even though it is mentioned in the caption. Would improve clarity to define units on the x-axis itself. Further, on MyoD1 expression (middle) figure, the statistical significance denotations are missing.

Thank you. We added units on the x-axis as suggested. In all Figure S6a panels, the significant differences compared to control are noted and differences that were not statistically different were not noted as “NS”. We have adjusted the figure legend to make this more clear.

Figure S7a, It would improve clarity to mention what expression is measured on the y-axis itself even though it is in the caption.

We added “Relative p53 expression” to the y-axis.

Formatting-

Lines- 391-393-Typos, seems like “TCP” is misspelled as “TPC”.

We greatly appreciate you pointing this out. This prompted us to look and we found others too. Thank you.

Line 1395- Caption is missing the denotation “d” that represents flow cytometry data.

Line 1397- There is a bracket around (e) but not on the rest of sub-Figures in the legend.

Thank you we made both changes to the figure legend.

Reviewer #2 (Remarks to the Author):

The authors have largely addressed my comments (as well as those from other reviewers) and I have nothing else to do than recommending this interesting interesting article for publication and complimenting with the authors for the excellent work

Thank you.

Reviewer #3 (Remarks to the Author):

The authors have address all questions and I don't have additional comments. Thanks

Thank you.